# Mössbauer Spectroscopy with a High Velocity Resolution in the Studies of Nanomaterials

**DOI:** 10.3390/nano12213748

**Published:** 2022-10-25

**Authors:** Irina V. Alenkina, Michael V. Ushakov, Paulo C. Morais, Ramakrishan Kalai Selvan, Ernő Kuzmann, Zoltán Klencsár, Israel Felner, Zoltán Homonnay, Michael I. Oshtrakh

**Affiliations:** 1Department of Experimental Physics, Institute of Physics and Technology, Ural Federal University, Ekaterinburg 620002, Russia; 2Genomic Sciences and Biotechnology, Catholic University of Brasilia, Brasilia 71966-700, DF, Brazil; 3Institute of Physics, University of Brasilia, Brasilia 70910-900, DF, Brazil; 4Department of Physics, Bharathiar University, Coimbatore 641046, India; 5Laboratory of Nuclear Chemistry, Institute of Chemistry, Eötvös Loránd University, 1117 Budapest, Hungary; 6Nuclear Analysis and Radiography Department, Centre for Energy Research, 1121 Budapest, Hungary; 7Racah Institute of Physics, The Hebrew University, Jerusalem 91904, Israel

**Keywords:** Mössbauer spectroscopy, nanoparticles in biology and medicine, nanoparticles in pharmacy, nanoparticles in technology, nanostructured alloys

## Abstract

The present review describes our long experience in the application of Mössbauer spectroscopy with a high velocity resolution (a high discretization of the velocity reference signal) in the studies of various nanosized and nanostructured iron-containing materials. The results reviewed discuss investigations of: (I) nanosized iron cores in: (i) extracted ferritin, (ii) ferritin in liver and spleen tissues in normal and pathological cases, (iii) ferritin in bacteria, (iv) pharmaceutical ferritin analogues; (II) nanoparticles developed for magnetic fluids for medical purposes; (III) nanoparticles and nanostructured FINEMET alloys developed for technical purposes. The results obtained demonstrate that the high velocity resolution Mössbauer spectroscopy permits to excavate more information and to extract more spectral components in the complex Mössbauer spectra with overlapped components, in comparison with those obtained by using conventional Mössbauer spectroscopy. This review also shows the advances of Mössbauer spectroscopy with a high velocity resolution in the study of various iron-based nanosized and nanostructured materials since 2005.

## 1. Introduction

Nanosized and nanostructured materials are intensively studied, developed and used in various fields of science and technology (see, e.g., [1,2]). A large part of these materials demonstrates various magnetic features that are significantly different from their bulk materials [3]. The majority of magnetic nanoparticles and nanostructures are based on iron and iron oxides, which are used in various fields: From electronics to medicine and pharmacy (see, e.g., reviews [4,5,6,7,8,9,10,11,12,13,14,15,16,17,18,19]). The presence of iron in these nanosized and nanostructured materials permits one to use ^57^Fe Mössbauer spectroscopy for their studies. This technique is the most sensitive to the ^57^Fe hyperfine interactions, dynamic properties, iron valence/spin state, iron magnetic state, phase composition of iron-bearing samples, etc. (e.g., [20,21,22,23]). Therefore, various iron-based nano-systems were studied by Mössbauer spectroscopy (some examples are, e.g., in [18,24,25,26,27]). These studies demonstrated a complex character of the Mössbauer spectra of various nanoparticles and nanostructured materials. Therefore, Mössbauer spectroscopy with a high velocity resolution was applied to study these materials extensively. The latter method demonstrates significant advantages over the conventional Mössbauer spectroscopy in the studies of nanosized and nanostructured materials in biology, medicine, pharmacy and technology, such as: (i) nanosized iron cores in: Extracted ferritin, ferritin in liver and spleen tissues in normal and pathological cases, ferritin in bacteria, pharmaceutical ferritin analogues; (ii) nanoparticles developed for magnetic fluids for medical purposes; (iii) nanoparticles and nanostructured FINEMET alloys developed for technical aims. In addition, our Mössbauer spectroscopy results are supported by scanning and transmission electron microscopy (SEM and TEM), X-ray diffraction (XRD) and magnetization measurements. Their data are also included.

## 2. Mössbauer Spectroscopy with a High Velocity Resolution

The basic principles of the Mössbauer effect and spectroscopy, the Mössbauer various parameters and the main methodological features were described in various books and reviews, some of which are in [20,21,22,23]. Technical and methodological features of so-called Mössbauer spectroscopy with a high velocity resolution were considered in [28,29,30,31]. The term “velocity resolution” in Mössbauer spectroscopy was suggested for evaluation and characterization of the quality of both: (i) the spectrometer velocity driving system and (ii) the measured Mössbauer spectrum in addition to a number of other characteristics (e.g., differential and integral nonlinearity of the velocity reference signal, differential and integral nonlinearity of the velocity signal, time and temperature instability of the velocity signal amplitude, drift of the zero velocity point, noise of the velocity signal, signal-to-noise ratio in the spectrum, and so on). 

The velocity reference signal formation (in the constant acceleration mode) by the digital–analog converter with discretization using 2*^n^* steps is shown as a scheme in Figure 1. The velocity resolution in the spectrometer velocity driving system is defined as the smallest velocity step 2*V*/2*^n^*, where 2*V* is the velocity range (from −*V* till +*V*) and 2*^n^* is the discretization of the digital–analog converter used to form the velocity reference signal. The limitation in increasing the velocity resolution (i.e., in decreasing of 2*V*/2*^n^*) is determined by the *n* value with the requirement that the integral error ±Δ*V**_int_* for the velocity step should not exceed half of the velocity step (the maximal error ±Δ*V**_v_* = *V*/2*^n^* for the velocity step formation). If ±Δ*V**_int_* exceeds ±Δ*V**_v_*, it means that the Doppler energy shift for the given γ-rays will overlap with those shifts of the next or previous velocity steps, and so on, as shown in the scheme in Figure 1. In this case, a smoothing of the Mössbauer spectrum may be a result of this error excess. Therefore, it is desirable to decrease the velocity resolution at least twice (2*V*/2*^n^*^−1^), and so on, to reach the requirement of ±Δ*V**_int_* ≤ ±Δ*V**_v_*. The obtained discretization of the velocity reference signal will determine the appropriate velocity resolution for the Mössbauer spectrometer.

Velocity resolution in the Mössbauer spectrum is a velocity step per one spectrum point (one channel of multichannel analyzer), i.e., the same as the calibration constant for the spectrum. The velocity resolution coefficient 1/2*^n^* in the velocity driving system in the chosen spectrometer is generally constant, and the velocity resolution value mainly depends on the velocity range (2*V*) (in some spectrometers, it is possible to change *n*). In contrast, the velocity resolution in the Mössbauer spectrum may be the same or less (for instance, 2*^n^*^−1^, 2*^n^*^−2^, and so on) due to the decrease in the number of channels in the multichannel analyzer used for registration of corresponding pulse counts. This may be performed by increasing the time window *t**_w_* for each memory cell (channel in multichannel analyzer). If discretization of the velocity reference signal and channels number are the same, the time window *t_i_* of the velocity step formed by the digital–analog converter should correspond to *t_w_* plus the dead time *t**_d_* required for channel switching. The start of the velocity reference signal is synchronized with opening the first memory cell (the first channel of the multichannel analyzer) for registration of pulse counts corresponding to resonant γ-rays with energy modulation by the first velocity step. Then, the next velocity step starts after *t_i_* with simultaneous switching the memory cell after the same time (*t_i_* = *t_w_* + *t_d_*), and so on. To decrease the velocity resolution in the spectrum by the factor of 2, it is necessary to: (i) double *t_w_* with registration of pulse counts during a time 2*t_w_* + *t_d_* = 2*t_i_* for two velocity steps or (ii) sum up the two neighboring channels after the spectrum measurement with the same velocity resolution as that in the spectrometer driving system, and so on (the latter is a much better way).

Thus, if one has two spectrometers with different velocity reference signal discretization, for example, *n* = 12 (discretization is 2^12^ = 4096) and *n* = 10 (discretization is 2^10^ = 1024), the measured spectra contain 4096 and 1024 experimental points, respectively. The difference in the velocity steps for these two cases is shown in the scheme in Figure 1, demonstrating four times higher velocity step in the latter case (dashed bold lines), and consequently, the number of spectral points is four times smaller. Generally speaking, conventional spectrometers usually use discretization of 2^8^ = 256 or 2^9^ = 512 for the measurements of spectra. In the case of spectrometers with different qualities of the velocity driving system, these differences may be the reason of smoothing of the absorption line shape features indicating the presence of several overlapping components (an example of visible comparison for the 128-channel and 4096-channel spectral line shapes taken from [31] is shown in Figure 2).

Thus, all Mössbauer measurements described below were carried out using an automated precision Mössbauer spectrometric system based on two SM-2201 spectrometers and a temperature variable liquid nitrogen cryostat (295–90 K) with moving absorber (see [28,29,30,31]). The saw-tooth velocity reference signals in the spectrometers were formed by digital–analog converters with discretization of 2^12^. All measurements were performed in transmission geometry with moving absorbers and spectra registration in 4096 channels. Then, for the spectra fit, some of them with poor signal-to-noise ratios were converted into 2048- or 1024-channel spectra by consequent summation of two or four neighboring channels, respectively (“signal” is the difference in transmitted intensity at resonance and off-resonance, “noise” is the square root of off-resonance intensity; this ratio depends on the off-resonance intensity (statistics of count rates) and absorption effect in the sample and does not depend on discretization of the velocity reference signal; however, conversion, e.g., of the 4096-channel spectrum into the 2048-channel spectrum leads to an intensity (statistics) increase of two times in the latter spectrum and, therefore, to an increase of 2^1/2^ for the value of the signal-to-noise ratio in comparison with the initial 4096-channel spectrum). The ^57^Co(Cr) in earlier studies and then the ^57^Co(Rh) sources (Ritverc GmbH, St. Petersburg, Germany) with activities in the range (1.8−0.3) × 10^9^ Bq were at room temperature. The velocity calibration and the control of the velocity driving systems in both spectrometers were achieved using the thin reference absorbers: (i) α-Fe foils with thicknesses of 10 and 7 μm (for the large velocity range) and (ii) sodium nitroprusside (SNP) with thicknesses of 10 and 5 mg Fe/cm^2^ (for the small velocity range). The line shapes of the Mössbauer spectra of these reference absorbers were pure Lorentzian. The examples of their line widths are the following: The line widths of the 1st and the 6th, the 2nd and the 5th, and the 3rd and the 4th peaks in the α-Fe sextet are 0.222 ± 0.010, 0.223 ± 0.010 and 0.217 ± 0.010 mm/s, respectively (7 μm, 4096 channels), whereas the line width of two peaks in the SNP doublet is 0.229 ± 0.002 mm/s (5 mg Fe/cm^2^, 4096 channels). The thicknesses of all measured samples usually were ≤10 mg Fe/cm^2^. Some details of the velocity driving system control using the reference absorbers were described in [32]. The Mössbauer spectra were fitted using the UNIVEM-MS program and in some cases using the MossWinn code [33]. Spectral parameters such as isomer shift, δ, quadrupole splitting, Δ*E*_Q_, quadrupole shift for magnetically split spectra, ε, magnetic hyperfine field, *H*_eff_, line width, Γ, relative subspectrum (component) area, *A*, and normalized statistical quality of the fit, χ^2^, were determined. Criteria of the best fits were the differential spectrum (the difference between experimental and calculated spectral points), χ^2^ and the physical meaning of the parameters. The instrumental (systematic) error for each spectral point was equivalent to ±0.5 channel (in the velocity scale), while that for the hyperfine parameters was equivalent to ±1 channel (in mm/s or kOe). If an error calculated with the fitting procedure (fitting error) for these parameters exceeded the instrumental (systematic) error, the larger error was used instead. The values of *A* are given in table as calculated in the fit with two decimal digits to keep the total relative area equal to 100%. The estimated relative error for *A* usually did not exceed 10%. Values of δ are given relative to α-Fe at 295 K.

## 3. Nanoparticles in Biology, Medicine and Pharmacy

The main biological molecule containing the iron-bearing nanosized core is iron storage protein ferritin, which can be found in mammals in various tissues and in bacteria. In the case of mammals, ferritin can be found in liver and spleen tissues, for example. Pharmaceutical ferritin analogues (iron–polysaccharide complexes) are used for the treatment of iron deficiency anemia. Furthermore, various iron-based nanoparticles are used in medicine as drug deliverers, contrast agents for MRI, hyperthermia for cancer treatment, magnetic fluids, etc. Some of these objects were studied using Mössbauer spectroscopy with a high velocity resolution.

### 3.1. Nanosized Iron Cores in Human Liver Ferritin

Ferritin, the iron storage protein, consists of a hollow protein shell with 24 protein subunits with a cavity of 8 nm, which contains the iron core in the form of ferrihydrite 5Fe_2_O_3_ × 9H_2_O or that with inorganic phosphates (FeOOH)_8_(FeO:OPO_3_H_2_), as shown schematically in Figure 3 using the information taken from the Protein Data Bank. The maximal number of iron atoms in a cavity can be up to ~4500, while the usual number of iron atoms in ferritin in a healthy human is ~2000–2500. Hemosiderin is an insoluble iron storage protein that is considered as agglomerations of denatured ferritin molecules. There are many reviews on the structure and functions of various ferritins (e.g., [34,35,36,37]). However, there is no exact information about the structure of the iron cores in ferritin. Therefore, extracted and lyophilized normal human liver ferritin was studied using Mössbauer spectroscopy with a high velocity resolution.

The first Mössbauer spectra of human liver ferritin (200 mg) were measured at 295 K and presented in 1024 and 4096 channels in [38,39]; the latter spectrum is shown in Figure 4. The fits of both spectra demonstrated that at least four quadrupole doublets with free variation of all parameters were required to obtain the best fit as demonstrated by (i) the absence of misfits at the differential spectra and (ii) the low χ^2^ values. This was different from the core-shell model for the iron core in ferritin developed in [40] and applied to fit the ferritin Mössbauer spectra using two spectral components only (see for review [25]). Then, the Mössbauer spectra of ferritin were measured in the temperature range 295–90 K [41,42,43,44]. However, the fits with free variations of all parameters demonstrated no consistency between data obtained at 295 and 90 K. Therefore, two approaches to fit the Mössbauer spectra of ferritin were suggested: (i) the rough fit using one quadrupole doublet within the homogeneous iron core model and (ii) the fit based on the heterogeneous iron core model, which should demonstrate some consistency with decreasing temperature. The first attempt to reach a consistency of the 295 and 90 K spectra was based on the assumption that the relative areas of the corresponding spectral components are approximately the same [43,44]. However, no consistency was achieved for the Mössbauer spectra measured at intermediate temperatures.

Nevertheless, application of the first rough model in [45,46,47] permitted us to observe the features in the temperature dependencies of some parameters and to estimate the Mössbauer temperature Θ_M_ (see [23]) for ferritin within the Debye model by measuring the second order Doppler shift δ_SOD_ and using equation:
(1)δSODT=−3kT2mc3ΘM8T+3TΘM3∫0ΘMTx3ex−1dx, where *m* is the mass of Fe atom, *k* is the Boltzmann constant, *c* is the speed of light, and *x* = *ħ*ω/*kT* (ω being frequency of vibrations).

The observed anomalies of both the line width and the total spectra area normalized to the value at 295 K as well as the temperature dependence of δ(*T*) = δ(0) + δ_SOD_ are shown in Figure 5. The value of Θ_M_ = 461 ± 16 K was deduced for ferritin. These anomalies did not result from: (i) cryostat vibrations, (ii) vibrations of the sample particles in the moving absorber, (iii) increasing in the Mössbauer effect probability with temperature decrease leading to line broadening and (iv) slowing down of the magnetic relaxation [46]. Therefore, a new heterogeneous iron core model for ferritin Mössbauer spectra fit was developed in [48,49]. This model considers different core layers/regions (or crystallites, etc.) with the same ^57^Fe microenvironment within considered layers or regions. However, the ^57^Fe microenvironments in different layers/regions are different. The contribution of interlayer/interregional effects can be neglected. In this case, the line width for each spectral component related to the corresponding core layer/region should be the same. Thus, the line widths for various components in the Mössbauer spectrum were taken the same and varied simultaneously during the fit. This model permitted us to obtain consistent fits for the Mössbauer spectra of ferritin (100 mg) in the entire temperature range 295–90 K where five quadrupole doublets were used. Selected spectra are shown in Figure 6.

The obtained parameters for the five quadrupole doublets for the nanosized iron cores in ferritin demonstrate anomalous behavior with temperature decrease. The temperature dependencies of the Γ, δ and Δ*E*_Q_ values for components 1–5 are shown in Figure 7. These quadrupole doublets 1–5 were related to the layers/regions with more or less FeOOH close packing, namely, doublet 1 with the smallest Δ*E*_Q_ value was assigned to the most close-packed layers/regions while doublet 5 with the largest Δ*E*_Q_ value was associated with the less close-packed layers/regions (see [48,49,50,51]). Figure 7 shows an anomalous increase in the Γ values below ~150 K, so-called critical temperature *T*_0_, which is accompanied by an anomalous decrease in the δ values for component 5 as well as different increases in the δ values for components 1–4 (the latter may be related to the different contributions of the second order Doppler shifts for the ^57^Fe in the corresponding layers/regions of the nanosized iron cores with different FeOOH packing). The Δ*E*_Q_ values increase with temperature decrease for all components, although below *T*_0_, an increase in ΔE_Q_ for doublet 5 is more pronounced. The relative areas of components 1–5 (Figure 8a) also demonstrate anomalous temperature dependencies below *T*_0_, indicating some changes in the considered layers/regions. In addition, the total spectral areas normalized to that at 295 K also showed anomalous decreases below *T*_0_ (Figure 8b). The insignificant effect of slowing down of the magnetic relaxation on these anomalies in the paramagnetic nanosized ferritin iron cores with temperature decreases in the range around *T*_0_–90 K was confirmed by magnetization measurements as shown in Figure 9: The zero-field-cooled (ZFC) and field-cooled (FC) magnetization measurements below *T*_0_ show two blocking temperatures for the largest iron cores (~22 K) and for the mean blocking temperature (~12 K).

Taking into account the possible low temperature structural disorder in ferritin iron cores frozen down to 80 K, which was determined by the temperature-dependent extended X-ray absorption fine structure (EXAFS) measurements [52], an assumption about the low temperature structural rearrangement between more and less close-packed FeOOH layers/regions was considered [48,49]. Furthermore, the presence of more and less close-packed layers/regions in the ferritin iron cores was confirmed by the room temperature high resolution TEM (HRTEM) in [51], indicating the presence of ordered domains with different lattice periodicities as demonstrated in Figure 10.

### 3.2. Nanosized Iron Cores in Ferritin in Liver and Spleen Tissues

Liver and spleen tissues contain the largest amount of ferritin and hemosiderin molecules. Therefore, studies of these tissues in normal and pathological cases are of interest for analysis of the iron core structures. However, the iron content in tissues is substantially smaller than that in extracted ferritin samples. Therefore, the Mössbauer spectra measurements of spleen and liver tissues with a high velocity resolution require longer measurement time, up to one month, while the signal-to-noise ratio may remain very poor. The first Mössbauer spectra of lyophilized chicken liver and spleen measured at 295 K in 4096 channels and converted into the 1024-channel spectra were published in [38,42,53]. The Mössbauer spectra of the nanosized ferritin-like iron cores in normal chicken liver and spleen and in lymphoid leukemia chicken spleen are shown in Figure 11a–c. These spectra demonstrate a very small absorption effect, especially for spleen from chicken with lymphoid leukemia. Therefore, the spectra of normal chicken tissues were fitted using two quadrupole doublets related to the ferritin-like iron (Figure 11a,b), whereas the spectrum of lymphoid leukemia chicken spleen was fitted with one quadrupole doublet only (Figure 11c). Taken into account the later heterogeneous model for the ferritin iron cores discussed above, new fits with equal line widths were applied as the simplified heterogeneous model (this is not the core-shell model). The obtained Mössbauer parameters were compared for normal liver and spleen in the plots in Figure 11d–f. Different Δ*E*_Q_ and δ values for doublet 1 may be considered as slightly different local ^57^Fe microenvironments in the more closed-packed layers/regions in the ferritin-like iron cores in normal chicken liver and spleen. However, the relative areas of doublets 1 and 2 in these spectra were respectively very close for chicken liver and spleen. A significantly smaller absorption effect in the spectrum of lymphoid leukemia chicken spleen (Figure 11c) indicates that in this case there is an iron deficiency.

Next, the nanosized ferritin-like iron cores in normal human liver and spleen and liver and spleen tissues from patients with hematological malignancies were studied by Mössbauer spectroscopy with a high velocity resolution at 295 K in [54,55,56,57,58,59,60]. Tissue samples of healthy human (NOR) spleen from donor “1” and liver from donor “2” were obtained post mortem and were used as the reference samples. Spleen and liver tissues from patients “Ch” with acute myeloid leukemia (AML) and “N” with non-Hodgkin B-cells mantle cell lymphoma (MCL) were obtained post mortem, whereas spleen tissues from patients “G” with MCL, “P”, “De” and “Do” with marginal zone B-cell lymphoma (MZL) and “B” with primary myelofibrosis (PMF) were obtained after splenectomy. The lyophilized powdered samples with weights of spleen samples of 900 mg NOR (“1”), 1335 mg MCL (“N”), 1400 mg MCL (“G”), 938 mg MZL (“De”), 1530 mg MZL (“Do”), 700 mg MZL (“P”), 1300 mg AML (“Ch”), and 1100 mg PMF (“B”), and liver samples of 1390 mg NOR (“2”), 1450 mg MCL (“N”), and 1370 mg AML (“Ch”) were studied. The 295 K Mössbauer spectra of normal and patients’ spleen and liver tissues are shown in Figure 12 and Figure 13. These spectra were fitted using the simple heterogeneous iron core model for tissues using two quadrupole doublets 1 and 2 with the same line widths varied during the fit. In some tissues, the third quadrupole doublet 3 was detected and related to residual hemoglobin oxidized to the methemoglobin form. The absorption effect varied for different spectra indicating different amounts of ferritin-like iron in the studied tissues.

Considering an assignment of the smaller Δ*E*_Q_ values (ferritin-like component 1) to the most close-packed layers/regions and association of the larger Δ*E*_Q_ values (ferritin-like component 2) with the less close-packed layers/regions in the nanosized iron cores, a comparison of the contributions of more and less close-packed FeOOH layers/regions for ferritin-like iron in spleen and liver tissues can be roughly performed as shown in Figure 14. The relative part of more close-packed layers/regions in the nanosized ferritin-like iron cores in normal spleen is slightly smaller than that for spleen from patients with hematological malignancies, whereas in the studied liver tissues, the relative parts of more and less close-packed layers/regions are the same. The ^57^Fe hyperfine parameters for these layers/regions (for ferritin-like components 1 and 2) demonstrate some small variations in some samples as shown in the plots of Δ*E*_Q_ vs. δ in Figure 15.

ZFC/FC magnetization measurements of spleen tissues from healthy human and patients with hematological malignancies indicate the presence of different phases: Paramagnetic and ferrimagnetic embedded in the diamagnetic matrices in various proportions (see Figure 16). However, magnetization measurements of liver tissues demonstrated some anomalies when ZFC curves lie above FC curves in some temperature ranges (see Figure 17). This observation was discussed in [59,60].

An interesting result was obtained for the ferritin-like iron cores in spleen tissue taken from a patient with primary myelofibrosis “B”. Comparison of the total spectral area normalized by sample weight that demonstrated the largest iron content in spleen tissue from patient “B” (see Figure 18a). The content of ferritin-like iron in the patient’s “B” spleen is about eight times higher than that in healthy human “1” spleen. If this is a result of the increase in the nanosized ferritin-like iron core, the low temperature Mössbauer spectrum of patient’s “B” spleen may demonstrate an additional magnetic sextet similar to those observed earlier for thalassemia patients’ spleen tissues at 87 K (see, e.g., [61,62]). The latter indicated the iron overloading and more crystalline iron cores in the case of thalassemia. However, the Mössbauer spectrum of spleen from patient “B” measured at 90 K (Figure 18b) demonstrates no statistically reasonable magnetic component. This is confirmed by TEM images of spleen tissues from healthy human “1” and patient “B” (Figure 18c,d), which demonstrate similar iron core sizes. These results may indicate that the nanosized iron cores in spleen tissues in the studied case of primary myelofibrosis contain approximately the same quantity of iron as in the normal tissue, which is in the labile form, whereas the number of ferritin molecules substantially increases in the patient’s spleen with splenomegaly. These processes permit the body to substitute the bone marrow, which is replaced by fibrosis and starts extramedullary hematopoiesis for hemoglobin biosynthesis in the spleen (see [60]).

### 3.3. Nanosized Iron Cores in Ferritin in Bacteria

Bacteria contain three types of ferritin-like proteins, namely, (1) bacterial ferritin, which is similar to that usually found in eukaryotes including mammals, (2) bacterioferritin and (3) dodecameric ferritin [63]. Bacterial ferritin consists of a 24-subunit shell with a cavity of 7.9 nm, which can contain the nanosized iron core with about 2500 iron atoms. Mössbauer spectroscopy with a high velocity resolution was applied for the study of bacterial ferritin in the rhizobacterium *Azospirillum brasilense* (strain Sp245) prepared in two different ways [64]. The bacteria were grown in media containing ^57^Fe, and then, one part (sample 1) was immediately frozen and lyophilized while the other part (sample 2) was additionally kept for 3 more days at room temperature in media free from nutrients and then quickly frozen and lyophilized. The Mössbauer spectra of *Azospirillum brasilense* (strain Sp245) samples 1 and 2 measured at 295 K and presented in 2048 channels are shown in Figure 19 with the TEM image of bacteria. The spectrum of sample 1 contains an additional quadrupole doublet 5 that corresponds to a ferrous compound while the other components 1–4 in both spectra are related to the ferritin-like iron similar to the human liver ferritin spectrum exhibited in Figure 6a. It is possible that the smaller iron core size in the bacterial ferritin leads to the smaller number of layers/regions in the core as compared to human liver ferritin. However, more detailed studies are required. A comparison of the ^57^Fe hyperfine parameters for corresponding quadrupole doublets for samples 1 and 2 is shown in Figure 20.

These components 1–4 can also be assigned to more and less close-packed FeOOH layers/regions in the nanosized iron core in order to increase the Δ*E*_Q_ value. However, there are small variations in the corresponding iron core layers/regions in samples 1 and 2 due to small variations in the iron local microenvironments reflected by the ^57^Fe hyperfine parameters (see Figure 20). The relative parts of more and less close-packed layers/regions in the iron cores in bacterial ferritin in samples 1 and 2 are also different, as shown in Figure 21. It is interesting to observe that the part of the most close-packed layers/regions in the iron cores in bacteria sample 1 is smaller than that for sample 2 and *vice versa*. It is possible that during the additional 3 days of bacteria growth in the media free from iron, bacteria use a part of iron from their bacterial ferritin that leads to some rearrangement of the layers/regions in the iron cores with small modifications of the iron local microenvironments.

### 3.4. Nanosized Iron Cores in Pharmaceutical Ferritin Analogues

Iron–polysaccharide complexes, which are ferritin analogues, are used and developed for iron deficiency anemia treatment. These complexes are nanosized iron cores in the form of ferric hydrous oxides (usually akaganéite β-FeOOH or ferrihydrite depending on the preparation) surrounded with a polysaccharide shell (dextran, dextrin, polymaltose, etc.). The size of the iron core does not usually exceed 10 nm.

Samples of Imferon (Fisons, UK), an injectable iron-dextran complex, was studied in lyophilized form and in frozen solution by Mössbauer spectroscopy with a high velocity resolution in [41]. After the first measurements in 1998 using the 512-channel spectra published in [65], one ampule of Imferon was kept for 10 years at room temperature while the lyophilized sample was kept at −20 °C. The measured Mössbauer spectra of lyophilized Imferon (at 295 K) and Imferon frozen solution kept for 10 years (at 90 K) were presented in 2048 channels while the spectrum of initial Imferon frozen solution was taken from [65] as measured (in 512 channels at 87 K) for comparison. These spectra are shown in Figure 22. The spectra of Imferon in lyophilized form and the initial one in frozen solution were refitted using the new heterogeneous iron core model while the rest of the spectrum was fitted using the previous heterogeneous iron core model with free variations of parameters. Mössbauer parameters are shown in Table 1. The spectra of Imferon in lyophilized form and in frozen solution (initial) were better fitted with three quadrupole doublets assigned to the different layers/regions in the nanosized iron cores while the Mössbauer spectra of human liver ferritin were decomposed using five quadrupole doublets in the range of 295–90 K (see Figure 6). This may be a result of some differences in the iron core structure in Imferon. Quadrupole doublet 4 observed in the spectrum of Imferon frozen solution has ^57^Fe hyperfine parameters similar to FeCl_2_ × 4H_2_O obtained in [66,67]. It is possible that FeCl_2_ was used for iron–dextran complexes preparation and the residual part of FeCl_2_ has remained in the injectable solution of Imferon. In the lyophilized sample FeCl_2_ can bind dextran and has a negligible Mössbauer effect probability at 295 K. A quite different spectrum was observed for Imferon frozen solution after 10 years of storage at room temperature. This spectrum (Figure 22c) shows three paramagnetic doublets and three magnetic sextets associated with the nanosized iron cores. This means that Imferon aging leads to (i) destruction of some dextran shells and agglomeration of nanosized cores and/or (ii) partial crystallization of the poor crystallized iron cores. The values of *H*_eff_ are similar to akaganéite, which Néel temperature is *T*_N_ = 299 K [68]; therefore, in the case of agglomeration, there are at least three ranges of agglomerated nanoparticle sizes of about 10 nm and larger that demonstrate magnetic hyperfine splitting at 90 K. Remaining quadrupole doublet 7 can be attributed to residual FeCl_2_ in the frozen solution of Imferon. Unfortunately, at the time of these studies [41,65] no other techniques were applied to Imferon characterization.

Samples of iron–polymaltose complexes Ferrum Lek, Maltofer^®^ and Ferrifol^®^ prepared by manufacturers as tablets were studied by Mössbauer spectroscopy with a high velocity resolution in the temperature range 295–90 K [41,42,43,44,45,46,47,48,49,50,51]. These spectra show the paramagnetic two-peak shape with non-Lorentzian lines. Therefore, all Mössbauer spectra of Ferrum Lek, Maltofer^®^ and Ferrifol^®^ were fitted within several heterogeneous iron core models described for ferritin in Section 3.1. The new model with equal line widths for spectral components varied during the fit was accepted for the spectra of ferritin analogues, and these Mössbauer spectra were well decomposed using five quadrupole doublets as shown in Figure 23, similar to the ferritin Mössbauer spectra. XRD study of Ferrum Lek, Maltofer^®^ and Ferrifol^®^ shows that the iron cores in Ferrum Lek and Maltofer^®^ are in the form of β-FeOOH, while those in Ferrifol^®^ are in the form close to ferrihydrite [44,51].

Application of the new heterogeneous iron core model to fit the Mössbauer spectra of Ferrum Lek, Maltofer^®^ and Ferrifol^®^ measured in the temperature range 295–90 K revealed unusual temperature dependencies of the line widths similar to that observed for the spectra of ferritin as demonstrated in Figure 24 (see also Figure 7a). This anomalous line broadening was not related to the slowing down of the magnetic relaxation and other factors as the main reasons of line broadening [46]. Moreover, the magnetization measurements of the ZFC/FC curves show mainly the paramagnetic state of the iron cores similar to ferritin (see Figure 9) and higher blocking temperatures for the largest iron cores (*T*_B_) (see Figure 25) in comparison with that for ferritin. Taking into account (i) an appearance of the magnetic sextets in the Mössbauer spectra of Ferrum Lek, Maltofer^®^ and Ferrifol^®^ at around 70–60 K while the ferritin spectrum is almost paramagnetic at 20 K [46,50] and (ii) the higher *T*_B_ values for analogues in comparison with that for ferritin, while *T*_0_ is higher for ferritin iron cores than that for Ferrum Lek, Maltofer^®^ and Ferrifol^®^, the main effect of the slowing down of the magnetic relaxation on the anomalous line broadening can be excluded.

Further, unusual temperature dependencies for other Mössbauer parameters such as isomer shift, quadrupole splitting, relative and total spectral areas for the iron cores in Ferrum Lek, Maltofer^®^ and Ferrifol^®^ were observed in the range 295–90 K as shown in Figure 26, Figure 27 and Figure 28. The values of δ demonstrate a decrease in temperatures below *T*_0_ (*T*_0_ is in the range ~125–145 K) for spectral components 3, 4 and 5 after an increase in the range 295–*T*_0_. The values of Δ*E*_Q_ show rapid increase with decreasing temperatures below *T*_0_, especially for spectral components 3, 4 and 5. The relative areas of spectral components demonstrate different behaviors below *T*_0_: The values of *A* for component 1 are rapidly increased, the values of *A* for component 2 have a very small increase while the values of *A* for components 3, 4 and 5 are rapidly decreased. The total spectra areas normalized to the values at 295 K also show an unusual decrease below *T*_0_. Within the new heterogeneous iron core model, spectral components 1 with the smallest Δ*E*_Q_ values are assigned to the iron core layers/regions with the most close-packed FeOOH while spectral components 5 with the largest Δ*E*_Q_ values are attributed to the iron core layers/regions with the less close-packed FeOOH (other spectral components are associated with the iron core layers/regions with intermediate packing degree of FeOOH in order to increase the Δ*E*_Q_ values). The differences in the packing of the iron cores observed in the HRTEM image of Ferrifol^®^ iron cores showed lattice periodicity variation between 2.0 and 2.7 Å (see Figure 29). Therefore, similar to ferritin, the nanosized iron cores in its pharmaceutical analogues can consist of at least five different layers/regions with different close-packing of FeOOH that may be reflected by the ^57^Fe hyperfine parameters resulting from decomposition of the Mössbauer spectra using five quadrupole doublets. With the temperature decrease down to so-called critical temperature *T*_0_, some structural rearrangements are started in the iron cores resulting in a decrease in the less close-packed FeOOH layers/regions with a corresponding increase in the most close-packed FeOOH layers/regions. This rearrangement, accompanied by some FeOOH dynamics in the cores, can lead to unusual line broadening and a decrease in the total spectral area (absorption effect) in the Mössbauer spectra of Ferrum Lek, Maltofer^®^ and Ferrifol^®^ in the temperature range *T*_0_–90 K.

### 3.5. Nanoparticles for Magnetic Fluids and Other Medical Aims

Various iron oxide nanoparticles were developed or used for medical purposes (see, e.g., [4,5,6,7,8,9]). Some iron oxide nanoparticles were studied by Mössbauer spectroscopy with a high velocity resolution. The first example is magnetite Fe_3_O_4_ nanoparticles as-prepared (Fe_3_O_4_) and dispersed in Copaiba oil (Fe_3_O_4_-CO). Copaiba oil was collected from the Brazilian rainforest Copaifera tree (*Copaifera Langsdorffii Desf.*) and used for preparation of magnetic fluids [69,70,71,72,73]. TEM and HRTEM images of as-prepared Fe_3_O_4_ nanoparticles are shown in Figure 30. The average size was estimated ~8 nm with the size distribution in the range 5–18 nm. XRD patterns of as-prepared Fe_3_O_4_ and dispersed in Copaiba oil Fe_3_O_4_-CO demonstrated the same phase of magnetite or maghemite (γ-Fe_2_O_3_) which cannot be distinguished by usual XRD measurements. However, HRTEM and SAED images (Figure 30b,c) indicate the presence of magnetite based on the crystallographic plane (311), *d* = 2.532 Å.

Mössbauer spectra of Fe_3_O_4_ and Fe_3_O_4_-CO samples measured in 4096 channels at 295 and 90 K are shown in Figure 31. They demonstrated asymmetric six-line patterns. Similar spectra were measured for magnetite nanoparticles with the size of 10 nm at 295 and 78 K in [74]; however, the correct fits revealed much more spectral components than in other studies of magnetite nanoparticle. Accounting for the tetrahedral (A) and octahedral [B] sites in magnetite and contribution of the surface only, i.e., decomposition with three magnetic sextets, did not give a good result for all measured spectra. Therefore, these spectra were fitted without an *a priori* model, based their analysis of the best fit reach and keeping physical meaning of the parameters. As a result, large numbers of magnetic sextets were obtained for the best fits. Additionally, a small quadrupole doublet was found in the 295 K spectra of both samples indicating the superparamagnetic state. To compare the spectra of as-prepared and dispersed in Copaiba oil magnetite samples, the normalized spectra differences were obtained (Figure 31e,f). These normalized spectra differences show that the variations between two spectra are much more pronounced at 295 K than that at 90 K.

Further comparison was performed using histograms of the distributions of the relative areas of magnetic sextets in the order of increase in the values of *H*_eff_ for corresponding spectral components as shown in Figure 32. To explain the observed fitting results for the high velocity resolution Mössbauer spectra, the hypothesis about complex architecture of magnetite nanoparticles was considered. Within this hypothesis, there are some variations in the nanoparticles from the surface inside the internal layers/regions, the presence of the (A) and [B] sites in these regions, formation of regions with different non-stoichiometry, the presence of nanodomains, interparticle interactions, size distributions, etc., which lead to small variations in the ^57^Fe local microenvironments. Variations observed for the magnetite nanoparticles dispersed in Copaiba oil were explained as a result of the surface Fe_3_O_4_ interactions with polar molecules of Copaiba oil, namely, kaurenoic acid (C_20_H_30_O_2_), containing polar groups, as shown in Figure 33. These polar groups may interact with iron cations at the magnetite nanoparticle surface and, therefore, decrease the typical spin flip-flop re-orientation time as the magnetic moment of the nanoparticle’s surface increases, further affecting the nanoparticle’s internal layers/regions.

This may be a reason of small detectable changes in the *H*_eff_ values for the corresponding spectral components, thus changing their relative contributions in the high velocity resolution spectra of the Fe_3_O_4_–CO in comparison with those in the spectra of the Fe_3_O_4_. Therefore, the normalized differences between two spectra are so pronounced at 295 K (see Figure 31e). At 90 K, however, the iron cations spin flip-flop re-orientation time is significantly decreased due to the higher energy barrier for the re-orientation. In this case, spin re-orientation time should be approximately the same for the as-prepared Fe_3_O_4_ sample and for the surface coated with polar molecules Fe_3_O_4_–CO sample. Thus, the effect of the surface interaction involving polar molecules of Copaiba oil decreases, leading to a decrease in the differences in the hyperfine interactions of the ^57^Fe in the as-prepared Fe_3_O_4_ and Fe_3_O_4_–CO samples and much smaller differences in the normalized difference for their Mössbauer spectra (see Figure 31d). This fact is also confirmed by magnetization measurements shown in Figure 34.

The next group of iron oxide nanoparticles comprises: (i) as-prepared maghemite nanoparticles (γ-Fe_2_O_3_) and maghemite nanoparticles coated with dimercaptosuccinic acid HO_2_CCH(SH)CH(SH)CO_2_H (DMSA) (γ-Fe_2_O_3_–DMSA); (ii) maghemite/magnetite nanoparticles prepared as low pH ionic magnetic fluid (IMF) (γ-Fe_2_O_3_/Fe_3_O_4_–IMF) and (iii) maghemite/magnetite nanoparticles coated with oleic acid CH_3_(CH_2_)_7_CH=CH(CH_2_)_7_COOH (OA) and dispersed in toluene (T) as surfacted magnetic fluid (γ-Fe_2_O_3_/Fe_3_O_4_–OAT) [72,73,75]. TEM images of the as-prepared γ-Fe_2_O_3_, γ-Fe_2_O_3_/Fe_3_O_4_–IMF and γ-Fe_2_O_3_/Fe_3_O_4_–OAT nanoparticles are shown in Figure 35 with corresponding histograms of the particle size distributions. The main size of around 8 nm was determined for the γ-Fe_2_O_3_ nanoparticles, whereas the main size of around 6.5 nm was estimated for the γ-Fe_2_O_3_/Fe_3_O_4_–IMF and γ-Fe_2_O_3_/Fe_3_O_4_–OAT nanoparticles.

The Mössbauer spectra of the as-prepared γ-Fe_2_O_3_ and γ-Fe_2_O_3_–DMSA nanoparticles measured in 4096 channels at 295 and 90 K and converted into 2048-channel spectra are shown in Figure 36. These spectra demonstrate asymmetric six-line patterns that cannot be fitted well using the known physical models for maghemite nanoparticles accounting for the ^57^Fe in the (A) and [B] sites. Therefore, similar to the spectra of magnetite nanoparticles, these spectra were fitted well using the large number of magnetic sextets and one paramagnetic quadrupole doublet at both temperatures. The 295 K spectra of the as-prepared γ-Fe_2_O_3_ and γ-Fe_2_O_3_–DMSA nanoparticles are visually different and may indicate the effect of surface interactions of DMSA with the nanoparticles’ surface as well as a decrease in the interparticle interactions due to the DMSA coating layer. The large number of magnetic sextets used to fit all Mössbauer spectra cannot be explained using an exact physical model yet. However, following the previous approach for magnetite nanoparticles, it is possible to suggest the presence of the surface layer and deeper layers/regions in the investigated nanoparticles with slight variations in the local microenvironments and orientations of the iron magnetic moments as well as variations in non-stoichiometry, size distributions, etc., i.e., the presence of complexity in the architecture of these nanoparticles. Therefore, a comparison of histograms of the distributions of the relative areas of magnetic sextets in the order of increase in the magnetic hyperfine field obtained for the Mössbauer spectra of the as-prepared γ-Fe_2_O_3_ and γ-Fe_2_O_3_–DMSA nanoparticles is shown in Figure 37. These histograms for the 295 K spectra show differences probably resulting from the nanoparticles coating with DMSA that leads to redistribution of corresponding magnetic components contributions. Histograms for the 90 K spectra also show differences as a result of different effects of the slowing down of the magnetic relaxation in the as-suggested layers/regions with redistribution of their relative content.

The Mössbauer spectra of the γ-Fe_2_O_3_/Fe_3_O_4_–IMF and γ-Fe_2_O_3_/Fe_3_O_4_–OAT nanoparticles measured at 295 and 90 K with a high velocity resolution are shown in Figure 38. The 295 K Mössbauer spectra of the γ-Fe_2_O_3_/Fe_3_O_4_–IMF and γ-Fe_2_O_3_/Fe_3_O_4_–OAT show asymmetric six-line patterns with some visual differences. These spectra were better fitted using 10 magnetic sextets and 1 quadrupole doublet without physical models for magnetite and maghemite nanoparticles. Similar to the above-mentioned iron oxide nanoparticles, the large number of magnetic sextets were considered as a result of the presence of the surface layer and deeper layers/regions within nanoparticles. It was found that the relative area for the paramagnetic component was different for these samples: About 4.7 % for the spectrum of the γ-Fe_2_O_3_/Fe_3_O_4_–IMF and about 7.6% for the spectrum of the γ-Fe_2_O_3_/Fe_3_O_4_–OAT. This observation indicates that γ-Fe_2_O_3_/Fe_3_O_4_–IMF demonstrates smaller contribution to the paramagnetic component than γ-Fe_2_O_3_/Fe_3_O_4_–OAT, whereas both nanoparticles have the same average size. It is possible that the oleic acid coating decreases interparticle interactions with further decreasing of the magnetic anisotropy energy barrier. The 90 K Mössbauer spectrum of the γ-Fe_2_O_3_/Fe_3_O_4_–IMF had no paramagnetic component while 10 magnetic sextets were used for the better spectrum fit. Following the previous consideration of histograms of the distribution of the relative areas of magnetic sextets in the order of increase in the values of *H*_eff_, a comparison of these histograms for the Mössbauer spectra of the γ-Fe_2_O_3_/Fe_3_O_4_–IMF and γ-Fe_2_O_3_/Fe_3_O_4_–OAT nanoparticles measured at 295 and 90 K is shown in Figure 39. The histograms for the spectra of the γ-Fe_2_O_3_/Fe_3_O_4_–IMF nanoparticles measured at 295 and 90 K demonstrate the redistribution of magnetic sextets and their relative areas and the values of magnetic hyperfine fields in addition to the slowing down of the magnetic relaxation with temperature decrease. Furthermore, a comparison of the histograms for the Mössbauer spectra of the γ-Fe_2_O_3_/Fe_3_O_4_–IMF and γ-Fe_2_O_3_/Fe_3_O_4_–OAT nanoparticles measured at 295 K demonstrates different distributions of magnetic sextets and their parameters that may be a result of the surface coating of the γ-Fe_2_O_3_/Fe_3_O_4_ nanoparticles with oleic acid.

Comparing the results obtained for the γ-Fe_2_O_3_–DMSA and γ-Fe_2_O_3_/Fe_3_O_4_–OAT nanoparticles, the effect of surface coating with dimercaptosuccinic acid or oleic acid, as schematically shown in Figure 40, leads to some variations in the Mössbauer parameters that can be seen in the Mössbauer spectra patterns and histograms of the distributions of the relative areas of magnetic sextets in the order of increase in the values of the magnetic hyperfine fields. Despite the absence of a physical model yet to explain a large number of magnetic sextets required for the best fits, a suggestion about complex nanoparticle architecture with the presence of the surface layer and deeper layers/regions, (A) and [B] sites in spinels, variations in non-stoichiometry within the nanoparticles, interparticle interactions, size distributions, and so on, can be the reasons for the Mössbauer spectra complexity.

Recently, the study of maghemite nanoparticle surface functionalization with citrate using Mössbauer spectroscopy was started, and the first results were considered in [76]. TEM and HRTEM images of the as-prepared γ-Fe_2_O_3_ nanoparticles (FM) as well as their size distribution are shown in Figure 41. The average size of these nanoparticles is about 10.6 nm. The scheme of preparation of maghemite nanoparticles with citrate surface functionalization (FMC) is shown in Figure 42. These two samples (FM and FMC) were studied by Mössbauer spectroscopy with a high velocity resolution at room temperature. The measured spectra are shown in Figure 43. The difference between these spectra is clearly visible: The Mössbauer spectra of FM and FMC nanoparticles demonstrate relaxation character with different relaxation rates [76]. However, it was not possible to fit these spectra reliably due to the poor signal-to-noise ratios. Therefore, to fit these Mössbauer spectra similarly to the above-mentioned fits of the ferric oxide nanoparticles, the spectra of FM and FMC were converted into 1024-channel spectra and were fitted using eight magnetic sextets and one quadrupole doublet, as shown in Figure 44.

More pronounced magnetically split pattern of the Mössbauer spectrum of FM nanoparticles may be explained as a result of interparticle interactions, which lead to slowing down of the magnetic relaxation with remaining small paramagnetic doublet (~3.9%). The difference between the Mössbauer spectra of FM (Figure 43a) and as-prepared γ-Fe_2_O_3_ nanoparticles (Figure 36a) may be a result of different sample preparations and different nanoparticle sizes. The shape of the Mössbauer spectrum of FMC nanoparticles demonstrates a smaller contribution of magnetically split components and a larger part of paramagnetic doublet (~9.4%). This can be explained as a result of a decrease in interparticle interactions due to the FMC nanoparticles capping by citrate leading to a faster relaxation rate.

Following the above-mentioned comparison of the histograms of the distributions of the relative areas of magnetic sextets in the order of increase the values of *H*_eff_, these histograms were obtained for the 1024-channel Mössbauer spectra of FM and FMC samples and shown in Figure 45. The differences in these histograms and the smallest values of *H*_eff_ for FMC are clearly seen. Again, the large number of magnetic sextets in both FM and FMC Mössbauer spectra measured at room temperature can be related to the complex nanoparticle architecture, namely, surface layer and deeper layers/regions in these nanoparticles as well as to other factors considered above. The Mössbauer spectra of FM and FMC nanoparticles were additionally measured at 80 K with a low velocity resolution in 256 channels in [76]. These spectra were decomposed using three magnetic sextets only.

## 4. Nanosized Ferrites

Various nanosized spinel ferrites demonstrate interesting magnetic and electrochemical properties. Therefore, these spinel ferrites were used or developed for various applications in spintronics, optoelectronics, ferrofluid technology, drug delivery, magnetic resonance imaging, electrochemical energy devices including electrodes for Li-ion batteries and supercapacitors, information storage, microwave devices and some other applications. Some nanosized ferrites, namely, NiFe_2_O_4_, CuFe_2_O_4_ and MgFe_2_O_4_, were studied using Mössbauer spectroscopy with a high velocity resolution.

### 4.1. Nickel Ferrites

NiFe_2_O_4_ has the inverse spinel structure with 8 tetrahedral (A) and 16 octahedral [B] positions for metal cations in the unit cell. The Fe^3+^ cations occupy both [B] and (A) sites while Ni^2+^ cations occupy the [B] sites only. Firstly, two samples of nanosized NiFe_2_O_4_ particles, which were synthesized by solution combustion method using two different fuels: EDTA (C_10_H_16_N_2_O_8_) and urea (CH_4_N_2_O), and named NA and NB, respectively, were studied by Mössbauer spectroscopy with a high velocity resolution in [73,77,78,79]. Characterization of NA and NB samples by XRD showed the presence of the NiFe_2_O_4_ spinel structure with the same lattice parameter *a* and different crystallite sizes calculated by the Scherrer equation: 26 nm for NA and 46 nm for NB. SEM images of NA and NB samples showed highly agglomerated nano-meter sized particles (Figure 46). Chemical composition of both samples determined by EDS are as follows: 29.6 at. % of Fe, 13.7 at. % of Ni, 56.6 at. % of O for NA, and 29.2 at. % of Fe, 13.9 at. % of Ni, 56.9 at. % of O for NB.

TEM and HRTEM images of NiFe_2_O_4_ nanoparticles in sample NA are shown in Figure 47. The average sizes for nanoparticles were 25 nm for NA and 35 nm for NB. The isothermal magnetization measurements (Figure 48) demonstrated different values of the saturation magnetic moment and remanence at both 5 and 295 K and the coercive field at 5 K for NA and NB as a result of the different nanoparticle sizes.

The Mössbauer spectra of NiFe_2_O_4_ nanoparticles measured at room temperature with a low velocity resolution were fitted using two magnetic sextets assigned to the (A) and [B] sites, mainly [80,81,82] (see, e.g., Figure 49a). The Mössbauer spectra of NA and NB samples measured in 4096 channels at 295 K (Figure 49b,c) resemble other spectra of nickel ferrite nanoparticles. However, the fits of the high velocity resolution spectra using the model of two magnetic sextets related to the (A) and [B] sites were not good, demonstrating large misfits in the differential spectra. The latter indicates the incomplete model for the spectra fits.

A new model for the Mössbauer spectra fits was developed based on the analysis of the ^57^Fe local microenvironments in both (A) and [B] sites in NiFe_2_O_4_ crystal. Two parts of the ideal NiFe_2_O_4_ crystal with the centers in the (A) and [B] sites occupied by Fe^3+^ (Fe^3+^_A_ and Fe^3+^_B_, respectively) are shown in Figure 50. The smallest coordination sphere in which both Fe^3+^_A_ and Fe^3+^_B_ cations have the [B] sites in the local microenvironment is 3.5 Å. In these local microenvironments, there are eight [B] sites around Fe^3+^_A_ and six [B] sites around Fe^3+^_B_ that can be occupied by Ni^2+^ cations. Calculated probabilities of various numbers of Ni^2+^ cations around both Fe_A_^3+^ and Fe_B_^3+^ cations are shown in Figure 51.

Considering the most reliable probabilities of different numbers of Ni^2+^ cations around the Fe^3+^_A_ and Fe^3+^_B_ cations ≥0.05 to reveal corresponding components in the Mössbauer spectra of NA and NB samples, five reliable probabilities for both (A) and [B] sites were obtained. Therefore, the measured Mössbauer spectra of NA and NB samples were fitted using 10 magnetic sextets as shown in Figure 52 and demonstrated significantly better differential spectra showing the quality of the fits. The obtained magnetic sextets were assigned to the (A) and [B] sites based on the differences between both δ and *H*_eff_ values for the ^57^Fe in these two sites in nickel ferrites (see, e.g., [80,81,82]) and taken into account variations in the ^57^Fe local microenvironments in two sites.

Further, the relative areas of spectral components ordered from the largest values of *H*_eff_ to the smallest ones shown in histograms for the (A) and [B] sites of NiFe_2_O_4_ nanoparticles in NA and NB samples were compared with the corresponding histograms of reliable probabilities (≥0.05) of different numbers of Ni^2+^ cations in the local microenvironments of both Fe^3+^_A_ and Fe^3+^_B_ cations estimated for NA and NB samples as shown in Figure 53. Some similarities between the corresponding histograms can be seen in Figure 53.

Some differences between corresponding histograms may be a result of differences between the ideal crystal model and real crystal structure of NiFe_2_O_4_ nanoparticles. Nevertheless, it is possible to consider the association of corresponding numbers of Ni^2+^ cations in the ^57^Fe local microenvironments in both (A) and [B] sites by comparing the histograms in Figure 53. The effect of one Ni^2+^ cation adding in the ^57^Fe local microenvironments in both (A) and [B] sites on the *H*_eff_ value decrease for NiFe_2_O_4_ nanoparticles in NA and NB samples was observed and shown in Figure 54. The behavior of the *H*_eff_ vs. number of Ni^2+^ cations is similar for both samples. However, a decrease in *H*_eff_ with an increase in the number of Ni^2+^ cations for the (A) sites demonstrates nonlinear characteristics while that for the [B] sites shows linear dependence.

Another study of nickel ferrite nanoparticles by Mössbauer spectroscopy with a high velocity resolution was carried out in [82]. Polycrystalline NiFe_2_O_4_ was prepared by the conventional ceramic method, this sample was denoted N1. Then, parts of the N1 sample were ground in a planetary ball mill at room temperature for various times: 15 min (sample N2) and 30 min (sample N3). XRD showed the same lattice parameter *a* for these samples while crystallite sizes determined by the Scherrer formula were of 150 nm (N1), 28 nm (N2) and 25 nm (N3). SEM images (Figure 55) also demonstrated a decrease in the particle sizes for milled samples in agreement with XRD data. EDS measurements showed slight deviation from stoichiometry in these samples. Isothermal magnetization curves measured for N1, N2 and N3 samples at room temperature are shown in Figure 56. These curves show that the saturation magnetic moment decreases for the milled samples N2 and N3 due to a decrease in particle size and due to the appearance of small paramagnetic components. Hysteresis loops were observed for samples N2 and N3 only as well as the coercive force was found increased with increasing the milling time: −1.1 kOe (N2) and −1.3 kOe (N3).

Mössbauer spectra of NiFe_2_O_4_ samples N1, N2 and N3 measured at 295 K with a high velocity resolution and converted into the 1024-channel spectra are shown in Figure 57. The spectrum of sample N1 with the largest average particle size (150 nm) demonstrates the visual shape of two six-line patterns similar to the other NiFe_2_O_4_ Mössbauer spectra (see, e.g., [80,81,82]). However, the fit of this Mössbauer spectrum using two magnetic sextets related to the (A) and [B] sites in NiFe_2_O_4_ was not optimal (see misfits in the differential spectra in insets for the spectrum of N1 in Figure 57). These misfits indicate that the model of two magnetic sextets is not sufficient for the spectrum fit. Based on the above-mentioned study of NiFe_2_O_4_ nanoparticles with average sizes of 25 and 35 nm using the calculation of different numbers of Ni^2+^ cations in the ^57^Fe local microenvironments in both (A) and [B] sites [79], the same model of a superposition of 10 magnetic sextets was applied to fit this Mössbauer spectrum. The result of this fit demonstrates significant improvement of the differential spectrum (see Figure 57a). This model was further applied to fit the spectra of N2 and N3 samples shown in Figure 57b,c. The latter spectra were fitted using eight magnetic sextets and one quadrupole doublet.

It is possible to consider the histograms of the relative areas of sextets vs. *H*_eff_ decreases for the spectral components revealed in the Mössbauer spectrum of N1 by analogy with those shown in Figure 53a,c,e,g for other NiFe_2_O_4_ nanoparticles. These histograms are shown in Figure 58.

Histograms for magnetic sextets obtained in the Mössbauer spectrum of sample N1 are different from those shown in Figure 53a,c,e,g. This may be a result of different sample preparations, deviation from stoichiometry, larger particle size (150 vs. 25 nm or 35 nm), internal inhomogeneity leading to larger deviations from the calculations for the ideal crystal structure and some other factors. Nevertheless, the Mössbauer spectra of three samples of NiFe_2_O_4_ nanoparticles (NA, NB and N1) measured with a high velocity resolution were better fitted using a new physical model accounting for the different probabilities of various numbers of Ni^2+^ cations in the ^57^Fe local microenvironments in both (A) and [B] sites in spinel crystals.

As for the Mössbauer spectra of samples N2 and N3, these spectra show the presence of additional paramagnetic components, which increase with the milling time (see Figure 59) due to a decrease in the nanoparticles’ size. Thus, observation of a quadrupole doublet in these spectra showed the presence of small NiFe_2_O_4_ nanoparticles in the superparamagnetic state.

Magnetic sextets S1–S8 obtained in the fits of the Mössbauer spectra of N2 and N3 samples were not assigned to the (A) and [B] sites in nickel ferrite because it was not possible to analyze the nanoparticles crystal structure and Ni^2+^ cation distributions in the particles affected by milling. Moreover, sextet S8 resembles a collapsing sextet with smaller *H*_eff_ and broad line width. This sextet indicates an increase in the contribution of the fast magnetic relaxation due to decreasing size. However, revealing of sextets S1–S8 demonstrates the possibility of Mössbauer spectroscopy with a high velocity resolution to detect tiny variations in the local iron microenvironments.

### 4.2. Copper Ferrites

CuFe_2_O_4_ has the inverse spinel structure where 8 Cu^2+^ cations occupy the octahedral [B] sites while 16 Fe^3+^ cations occupy both the tetrahedral (A) and octahedral [B] positions in the unit cell; however, the inversion parameter may be smaller than one. Samples of CuFe_2_O_4_–SnO_2_ nanocomposites were studied firstly by Mössbauer spectroscopy with the spectra presentation in 512 channels [84,85]. The Mössbauer spectra of (CuFe_2_O_4_)_1–*x*_(SnO_2_)*_x_* nanocomposites with *x* = 0, 1, 5, 10 and 20 wt.% were fitted using the conventional model with two magnetic sextets assigned to the (A) and [B] sites in spinel (see, e.g., [86,87,88,89,90]) with additional magnetic and paramagnetic components resulting from the effect of SnO_2_ at *x* = 5, 10 and 20 wt.%. Further, these initial spectra were converted into the 1024-channel spectra and fitted again [91]. TEM and HRTEM images of synthesized CuFe_2_O_4_ and (CuFe_2_O_4_)_0.95_(SnO_2_)_0.05_ nanocomposites are shown in Figure 60. These images demonstrate average particle sizes of 10 and 25 nm, respectively.

The fits of the 1024-channel spectra in [91] revealed one more magnetic sextet in the spectra of CuFe_2_O_4_ (see Figure 61a) and (CuFe_2_O_4_)_0.99_(SnO_2_)_0.01_ as well as in the spectrum of (CuFe_2_O_4_)_0.80_(SnO_2_)_0.20_. Unfortunately, at that time, there was no control of the spectra fits using the differential spectra. Therefore, insertion of the differential spectra in the recently performed fits repetition clearly demonstrated misfits indicating incomplete fits (see, e.g., Figure 61b with the spectrum of CuFe_2_O_4_). Therefore, the Mössbauer spectra of (CuFe_2_O_4_)_1–*x*_(SnO_2_)*_x_* nanocomposites with *x* = 0, 1, 5, 10 and 20 wt.% converted into the 1024-channel spectra were refitted to achieve better results and appropriate differential spectra. The obtained results are shown in Figure 60. A new fit of the Mössbauer spectrum of CuFe_2_O_4_ nanoparticles (Figure 62a) revealed 10 magnetic sextets. Based on the difference between δ values for sextets assigned to the (A) and [B] sites (δ < 0.30 mm/s and δ > 0.30 mm/s, respectively, see, e.g., [86,87,88,89,90]), the obtained sextets were associated with [B] sites (6 sextets) and (A) sites (4 sextets). These sextets can also be related to some variations in the ^57^Fe local microenvironments in two sites resulting from different probabilities of the numbers of Cu^2+^ cations in these microenvironments. The Mössbauer spectrum of (CuFe_2_O_4_)_0.99_(SnO_2_)_0.01_ nanocomposite (Figure 62b) looks similar to that for CuFe_2_O_4_ nanoparticles with some small visual differences. This spectrum was fitted using the same model and 10 magnetic sextets assigned to the [B] sites (six sextets) and (A) sites (four sextets). However, the Mössbauer parameters were slightly different for these two spectra. The Mössbauer spectra of (CuFe_2_O_4_)_1–*x*_(SnO_2_)*_x_* nanocomposites with *x* = 5, 10 and 20 wt.% were different from the first two ones and showed the presence of additional paramagnetic components, which were fitted with two quadrupole doublets D1 and D2. The ^57^Fe hyperfine parameters of these doublets correspond to Fe^3+^ compounds with the total relative area in the range ~6.3–6.9%. Parameters for D1 are close: δ = 0.406 mm/s, Δ*E*_Q_ = 0.656 mm/s (*x* = 5 wt.%), δ = 0.411 mm/s, Δ*E*_Q_ = 0.670 mm/s (*x* = 10 wt.%) and δ = 0.411 mm/s, Δ*E*_Q_ = 0.624 mm/s (*x* = 20 wt.%). In contrast, parameters of D2 were different for nanocomposites with *x* = 5 and with *x* = 10 and 20 wt.% of SnO_2_. Moreover, the line widths for D2 were very broad. However, it is not possible to excavate more information from these spectra fits.

The number of magnetic sextets obtained in the fits of the Mössbauer spectra of (CuFe_2_O_4_)_1–*x*_(SnO_2_)*_x_* nanocomposites with *x* = 5, 10 and 20 wt.% were different: 10 for both *x* = 5 and 10 wt.% and 12 for *x* = 20 wt.%. It is not possible to relate these sextets to the (A) and [B] sites correctly due to the unknown effect of SnO_2_ on the crystal structure and iron local microenvironments. However, sextets S1–S7 have δ < 0.30 mm/s and S8–S10 have δ > 0.30 mm/s for *x* = 5 wt.%, sextets S1–S8 have δ < 0.30 mm/s and S9, S10 have δ > 0.30 mm/s for *x* = 10 wt.% while sextets S1–S9 have δ < 0.30 mm/s and S10–S12 have δ > 0.30 mm/s for *x* = 20 wt.%.

A comparison of histograms of the relative areas of magnetic sextets related to the (A) and [B] sites in the order of magnetic hyperfine field decrease for CuFe_2_O_4_ nanoparticles and (CuFe_2_O_4_)_0.99_(SnO_2_)_0.01_ nanocomposite is shown in Figure 63. The Mössbauer spectra of both samples were similar with small differences of parameters. Therefore, corresponding histograms for the (A) and [B] sites were also similar with some variations, which can be a result of SnO_2_ addition. Similar to the above consideration of the Mössbauer spectra of NiFe_2_O_4_ nanoparticles [79], the revealing of several magnetic sextets assigned to both (A) and [B] sites was associated with differences in the local microenvironments of ^57^Fe^3+^ in these sites (Fe^3+^_A_ and Fe^3+^_B_, respectively). These differences may be a result of different numbers of Cu^2+^ cations around the Fe^3+^_A_ and Fe^3+^_B_. To check this suggestion, the calculations of the Fe^3+^_A_ and Fe^3+^_B_ local microenvironments and the numbers of Cu^2+^ cations near (A) and [B] sites were carried out.

The formula for copper ferrite was taken as (Cu*_y_*^2+^Fe_1–*y*_^3+^)_A_[Cu_1–2*y*_^2+^Fe_2*y*_^3+^]_B_O_4_^2–^ from [92], where *y* is the inversion parameter. XRD showed the tetragonal crystal structure of CuFe_2_O_4_ nanoparticles with an increase in tetragonal distortion and an increase in SnO_2_ adding. Calculations of the minimal radius of the coordination sphere around the Fe^3+^_A_ and Fe^3+^_B_ cations containing both (A) and [B] sites in the ideal CuFe_2_O_4_ crystal showed a value of 3.7 Å. These local structures are shown in Figure 64, indicating positions of the (A) and [B] sites and oxygens.

These calculations showed that within the coordination sphere with a radius of 3.7 Å, Fe^3+^_A_ has six (A) and six [B] sites while Fe^3+^_B_ has twenty (A) and four [B] sites. Considering an inverse character of copper ferrite spinel [92], to simplify calculations, the presence of one Cu^2+^ cation in the (A) sites was accounted for. The obtained probabilities of different numbers of Cu^2+^ cations (total in the (A) and [B] sites) around the Fe^3+^_A_ and Fe^3+^_B_ are shown in Figure 65.

These histograms of the probabilities of different numbers of Cu^2+^ cations in the local microenvironments of Fe^3+^_A_ and Fe^3+^_B_ demonstrate four and five reliable different numbers of Cu^2+^, respectively. This result may confirm decomposition of the Mössbauer spectra of CuFe_2_O_4_ nanoparticles and the (CuFe_2_O_4_)_0.99_(SnO_2_)_0.01_ nanocomposite with four magnetic sextets assigned to ^57^Fe in the (A) sites and six magnetic sextets associated with ^57^Fe in the [B] sites (see Figure 62 and Figure 63).

### 4.3. Magnesium Ferrites

Magnesium ferrite MgFe_2_O_4_ is an inverse spinel where 16 Fe^3+^ and 8 Mg^2+^ cations occupy both the tetrahedral (A) and octahedral [B] sites with the general formula (Mg_1-_*_y_*Fe*_y_*)_A_[Mg*_y_*Fe_2-_*_y_*]_B_O_4_. It was shown that *y* is in the range of 0.6–0.9 for MgFe_2_O_4_ (see [93] and references therein). Three samples of MgFe_2_O_4_ nanoparticles prepared by solution combustion method using different amounts of succinic acid as the fuel (with fuel/oxidant ratios of 0.5, 1 and 1.5) and named as MFO1, MFO2 and MFO3, respectively, were studied using Mössbauer spectroscopy with a high velocity resolution in [94]. SEM images of MFO1, MFO2 and MFO3 samples are shown in Figure 66.

These SEM images show that the MgFe_2_O_4_ particles in the samples are agglomerated with a slightly distorted shape. The size of the particles is decreased with increasing the concentration of fuel in the order of MFO1 > MFO2 > MFO3. XRD patterns of MFO1, MFO2 and MFO3 samples are shown in Figure 67. XRD demonstrates the single-phase spinel composition of these samples as well as the XRD peaks agreed with the same peaks from ICDD card No. 01-088-1935 for MgFe_2_O_4_. Crystal structure parameters of the MgFe_2_O_4_ samples as well as Fe^3+^ and Mg^2+^ occupations of the (A) and [B] sites were determined by full profile Rietveld refinement. The inversion parameter was determined as *y* = 0.9 with a magnesium ferrite formula (Mg_0.1_Fe_0.9_)[Mg_0.45_Fe_0.55_]_2_O_4_.

The crystallite size calculated from XRD patterns using the Scherrer formula decreases with an increase in fuel concentration: 110 nm (MFO1), 95 nm (MFO2) and 75 nm (MFO3). Isothermal magnetization curves also demonstrate some differences for these three samples as shown in Figure 68. The results of magnetization measurements indicated that the values of coercivity decreased while remanent magnetization values increased continuously with increasing the fuel amount and decreasing the crystallite size.

Mössbauer spectra of MgFe_2_O_4_ nanoparticles (samples MFO1, MFO2 and MFO3) measured at 295 K with a high velocity resolution in 4096 channels are shown in Figure 69. These spectra are asymmetrical six-line patterns with some asymmetrical peaks, indicating the presence of a superposition of several magnetic sextets. Previously, the room temperature Mössbauer spectra of MgFe_2_O_4_ nanoparticles measured using conventional spectrometers (with a low velocity resolution) were mainly fitted using two magnetic sextets assigned to ^57^Fe in the (A) and [B] sites, respectively (see, e.g., [95,96,97]). The fit of the present spectra using two magnetic sextets only showed many large misfits at the differential spectra. Therefore, these spectra were fitted by assuming a larger number of magnetic sextets, and the results are shown in Figure 69.

Based on the ^57^Fe hyperfine parameters, these magnetic sextets were assigned to spectral components related to Fe^3+^ cations in the (A) and [B] sites. In the sample MFO1, one small magnetic sextet (A ≈ 3%) with the largest values of *H*_eff_ ≈ 505 kOe and δ = 0.489 mm/s was considered as hematite α-Fe_2_O_3_ (see for comparison, e.g., [68]), while all other spectral components were related to MgFe_2_O_4_. The results of these fits showed that the spectrum of MFO1 sample was decomposed using eight magnetic sextets associated with the (A) sites and seven magnetic sextets associated with the [B] sites in MgFe_2_O_4_, the spectrum of MFO2 was fitted using seven magnetic sextets assigned to the (A) sites and nine magnetic sextets assigned to the [B] sites, and the spectrum of MFO3 was decomposed using nine magnetic sextets related to the (A) sites and seven magnetic sextets related to the [B] sites. Similar to decomposition of the Mössbauer spectra of NiFe_2_O_4_ and CuFe_2_O_4_ nanoparticles (see Section 4.1 and Section 4.2 and [79,83]), these large numbers of magnetic sextets assigned to the (A) and [B] sites in MgFe_2_O_4_ were related to variations in the iron local microenvironments in the (A) and [B] sites (Fe^3+^_A_ and Fe^3+^_B_, respectively) in these samples. Therefore, considering the equal Mössbauer effect probabilities (*f*-factors) for ^57^Fe in these samples and sites, it is possible to evaluate the relative iron fraction in each site in the MFO1, MFO2 and MFO3 samples using the corresponding relative area of spectral components. The histograms of the *A* values for magnetic sextets related to the (A) and [B] sites (the relative iron fractions in these sites) in the order of *H*_eff_ decrease are shown in Figure 70. The differences in these histograms may reflect different iron local microenvironments in the (A) and [B] sites for MFO1, MFO2 and MFO3 samples, which were prepared with different fuel concentrations.

For the analysis of the iron local microenvironments in the (A) and [B] sites in MgFe_2_O_4_ and samples of MFO1, MFO2 and MFO3 nanoparticles, the ideal structure of the Fe^3+^_A_ and Fe^3+^_B_ was considered as shown in Figure 71. The presence of both (A) and [B] sites in the Fe^3+^_A_ and Fe^3+^_B_ local microenvironments was evaluated for the coordination sphere with a minimal radius of 3.7 Å. Therefore, further calculations were performed for the local microenvironments within this sphere. From XRD patterns Rietveld refinement (see above) the structural formula of the MgFe_2_O_4_ samples was determined as (Mg_0.1_Fe_0.9_)[Mg_0.45_Fe_0.55_]_2_O_4_ with the inversion parameter *y* ≈ 0.9. The refined structural parameters of MFO1, MFO2 and MFO3 samples are practically the same within the limits of errors. The probabilities of Fe^3+^ and Mg^2+^ cations presence in the (A) and [B] sites were determined as follows: 0.87 Fe and 0.13 Mg in (A) and 0.55 Fe and 0.45 Mg in [B] (MFO1); 0.85 Fe and 0.15 Mg in (A) and 0.57 Fe and 0.43 Mg in [B] (MFO2) and 0.86 Fe and 0.14 Mg in (A) and 0.58 Fe and 0.42 Mg in [B] (MFO3). Calculations showed that within the chosen coordination sphere of 3.7 Å, there are six (A) sites and six [B] sites around the Fe^3+^_A_ while there are four (A) sites and twenty four [B] sites around the Fe^3+^_B_. Finally, the histograms of the probabilities of different numbers of Mg^2+^ cations in the Fe^3+^_A_ and Fe^3+^_B_ local microenvironments were obtained (Figure 72).

Considering the probabilities of about 5% and higher, the five reliable different local microenvironments with various numbers of Mg^2+^ cations were determined for the Fe^3+^_A_ and six different local microenvironments with various numbers of Mg^2+^ cations were determined for the Fe^3+^_B_. However, the results of decompositions of the Mössbauer spectra of MFO1, MFO2 and MFO3 demonstrated 15 and 16 magnetic sextets with different relative areas (relative iron fractions) assigned to the (A) and [B] sites with variations in the Fe^3+^_A_ and Fe^3+^_B_ local microenvironments (see Figure 67 and Figure 68). This discrepancy may be a result of the possible effect of geometrical positions of Mg^2+^ cations with respect to the Fe^3+^_A_ and Fe^3+^_B_ cations, e.g., opposite two Mg^2+^ cations and neighboring two Mg^2+^ cations can be a reason for two different magnetic hyperfine fields on the ^57^Fe, while these calculations did not consider these positions and did not calculate their probabilities. Nevertheless, these calculations demonstrate again, such as those for NiFe_2_O_4_ and CuFe_2_O_4_ nanoparticles in Section 4.1 and Section 4.2, that there are variations in the Fe^3+^_A_ and Fe^3+^_B_ local microenvironments in MgFe_2_O_4_ that also correlate with the best fits of the MFO1, MFO2 and MFO3 Mössbauer spectra. Moreover, some variations in the Fe^3+^_A_ and Fe^3+^_B_ local microenvironments in these samples were observed that may result from different fuel concentrations used for MFO1, MFO2 and MFO3 samples preparation.

## 5. FINEMET Alloys

The nanostructured Fe-Si-B-Nb-Cu alloys prepared by annealing amorphous alloys made by the single roller method over their crystallization temperature demonstrated magnetic properties (e.g., variations in permeability) that are widely used in the industry for many applications such as saturable reactors, choke coils and transformers, etc. These alloys, called “FINEMET”, were developed in 1988 [98] and are interesting by now for investigation using Mössbauer spectroscopy as one of the most powerful techniques. Variations in the soft magnetic properties of FINEMET alloys can be achieved by inducing a variation in transversal magnetic anisotropy by stress annealing under different tensile stresses or under different transversal magnetic fields as well as by defects formation using irradiation with swift heavy ions (see, e.g., [99,100] and references therein). Therefore, the study of the iron local microenvironments in FINEMET alloys in relation to their magnetic properties, in particular, to their magnetic anisotropy and permeability, is of interest.

Mössbauer spectroscopy with a high velocity resolution was used to study FINEMET alloys with a composition of Fe_73.5_Si_15.5_Nb_3_B_7_Cu_1_ prepared under stress annealing with two different tensile stresses (named FT1 and FT2), with an external magnetic field (F1) and without a tensile and magnetic field (F2) with the following permeabilities: μ_r_ = 1350 (FT1), μ_r_ = 6000 (FT2), μ_r_ = 34,500 (F1) and μ_r_ = 190,000 (F2) [101,102]. The Mössbauer spectra of FT1 and FT2 FINEMET alloys measured in 4096 channels are shown in Figure 73. The magnetic anisotropy was characterized by measuring the ratio of the relative areas of the second and the fifth lines (*A*_2,5_) to those of the first and the sixth lines (*A*_1,6_) in sextets, determining the angle *θ* between the direction of magnetic moment and γ-ray from the formula:*A*_2,5_/*A*_1,6_ = 4 sin^2^*θ*/(3 (1 + cos^2^*θ*))(2)
with the fixed ratio of *A*_1,6_/*A*_3,4_ = 3 (*A*_3,4_ is the relative areas of the third and the fourth lines in sextets). The Mössbauer spectra of FT1 and FT2 samples were roughly decomposed using the MossWinn code [33] into three sextets corresponding to Fe having four, five and six nearest Fe neighbors in sublattice A, one sextet corresponding to Fe in sublattice B in which Fe has seven or eight nearest Fe neighbors in Fe-Si phase and three sextets assigned to Fe in the amorphous phase, similar to previous studies of the annealed FINEMET alloys [99,103]. The resulting *θ* values were ~74.4° (FT1) and ~78.5° (FT2).

A more detailed study was carried out for FINMET ribbons F1 and F2 [102]. SEM images of the ribbon surfaces are shown in Figure 74. SEM images show some differences between two ribbons: Nanocrystalline precipitates have spherical shape with a narrow size distribution around 10–12 nm in sample F2 while the precipitates have a broad size distribution between 10–80 nm in sample F1 probably due to the effect of post crystallization heat treatment under the external magnetic field.

XRD patterns for FINEMENT ribbons F1 and F2 are shown in Figure 75. These patterns were fitted with the full profile Rietveld analysis. The results show the presence of a cubic phase of Fe_3_Si/FeSi (the unit cell parameter was as follows: *a* = 5.674(3) Å for F1 ribbon and *a* = 5.672(3) Å for F2 ribbon) and amorphous phase. Using the Scherrer formula with a shape factor of 0.9, the crystallite sizes were obtained ~145 Å for sample F1 and ~155 Å for sample F2.

The Mössbauer spectra of FINEMET samples F1 and F2 were measured in 4096 channels and then converted into 2048-channel spectra to increase the signal-to-noise ratio. These spectra (see Figure 76a,b) are as complex as those shown in Figure 73 for samples FT1 and FT2 and look very similar to each other. Therefore, the normalized difference of these spectra is shown in Figure 76c which clearly refers to differences between the spectra of F1 and F2 ribbons. To fit these spectra, the well crystallized part of FINEMET was assumed to be composed mainly of an α-Fe-Si phase having the DO_3_ structure (see [104]) with an excess of iron in comparison with α-Fe_3_Si. The relative occurrence of various local iron microenvironments in α-Fe_3+*x*_Si_1-*x*_ were calculated as a function of *x*, and a corresponding theoretical Mössbauer fit model of the α-Fe_3+*x*_Si_1-*x*_ phase has been worked out and adapted to the MossWinn program (see [102] and references therein). In the fit model, the relative intensities of the predicted spectrum components of the α-Fe-Si phase of FINEMET were constrained to their theoretical values as a function of the Si concentration in α-Fe_3+*x*_Si_1-*x*_, which the latter was adjusted as a free parameter during the fit. For the development of the fit model, the scheme of the ordered DO_3_ structure of α-Fe_3_Si was used as shown in Figure 77. The presence of the following iron microenvironments was considered in the crystalline DO_3_ phase: (i) Fe atoms at the A sites having eight nearest neighbor D-site Fe atoms (A_8_); (ii) Fe atoms at the A sites having seven nearest neighbor D-site Fe atoms (A_7_); (iii) Fe atoms at the A sites having six nearest neighbor D-site Fe atoms (A_6_); (iv) Fe atoms at the A sites having five nearest neighbor D-site Fe atoms (A_5_); (v) Fe atoms at the A sites having four nearest neighbor D-site Fe atoms (A_4_) and (vi) Fe atoms at the D sites having eight nearest neighbor A-site Fe atoms (D). Besides the magnetic sextet components of the crystalline α-Fe_3+*x*_Si_1-*x*_ phase and the spectral component accounting for the amorphous phases of FINEMET, the fit of the Mössbauer spectra of samples F1 and F2 required the assumption of an additional sextet component attributed to another crystalline phase. The results of the fit are shown in Figure 78. The fit provided a Si concentration of the α-Fe-Si phase as ~21 at. % in both of the samples. The ratio of the relative areas of the second and the fifth peaks and the first and the sixth peaks were assumed to be the same for the sextets belonging to the Fe-Si phase. The resulting values of *θ* (see Equation (2)), which were obtained within this model, were ~81° (T1, μ_r_ = 34,500) and ~88° (T2, μ_r_ = 190,000). This result shows the correlation of the direction of magnetic moment and permeability in FINEMET alloys.

This new fitting model is better than the previous one [103]. However, the differential spectra in Figure 78 show some residual misfits, indicating that the fitting model is not completed yet. Therefore, the Mössbauer spectra of FINEMET ribbons FT1, FT2, F1 and F2 were fitted using the UNIVEM-MS program without any *a priori* physical model similar to fitting the complex Mössbauer spectra of iron oxide nanoparticles (see Section 3.5) to reach the best fits with a physical meaning of parameters. The results of these fits are shown in Figure 79.

These decompositions with 18 magnetic sextets demonstrate the differential spectra without significant misfits. However, there is no physical meaning of this number of spectral components yet. It is possible to consider a hypothesis that this may be a result of various local microenvironments for D_Fe_ and A_Fe_ sites and for iron in the amorphous phases considered above (see also Figure 77), which may be revealed in the spectra measured with a high velocity resolution. In this case, the magnetic sextets resulting from the latter decompositions can be assigned to ^57^Fe in the amorphous phase (*H*_eff_ is in the range 50–190 kOe), ^57^Fe in the crystalline sublattice D (*H*_eff_ is in the range 195–296 kOe) and ^57^Fe in the crystalline sublattice A (*H*_eff_ is in the range 297–340 kOe). Histograms of the relative areas of the magnetic sextets vs. *H*_eff_ increase for these fits are shown in Figure 80.

Thus, if these fits can be reliable, there are at least five different microenvironments for ^57^Fe in the amorphous phase (~26% FT1, ~25% FT2, ~28% F1 and ~29% F2), at least eight different microenvironments for ^57^Fe in the crystalline sublattice D (~41% FT1, ~44% FT2, ~48% F1 and ~45% F2) and five different microenvironments for ^57^Fe in the crystalline sublattice A (~33% FT1, ~31% FT2, ~23% F1 and ~26% F2) in the FINEMET alloys. The values of *θ* obtained within these fits are as follows: ~69° for FT1, µ_r_ = 1350, ~71° for FT2, µ_r_ = 6000, ~74° for F1, µ_r_ = 34,500 and ~76° for F2, µ_r_ = 190,000.

## 6. Conclusions

Mössbauer spectroscopy appeared to be a useful tool for the study of various iron-containing nanoparticles and nanostructured materials in both living and abiotic systems. Application of Mössbauer spectroscopy with a high velocity resolution for investigation of these materials demonstrates some advances in the case of complex spectra with overlapped spectral lines. This technique permits to decompose the complex Mössbauer spectra measured in 4096 channels and sometimes converted into the 2048- and 1024-channel spectra using more spectral components than that extracted from the Mössbauer spectra of the same samples measured in 512 or 256 channels. The necessity of these components in the fits is clearly confirmed by misfits at the differential spectra, indicating incomplete fitting models with a smaller number of components even in the case of the use of physical models for these fits. Therefore, it is possible to excavate more detailed information about the iron local microenvironments in nanosized and nanostructured materials. However, decompositions of the complex Mössbauer spectra with many components require new physical models for explanation of these results and relation to the different iron sites with variations in their local microenvironments in the materials. It was possible to demonstrate that the original results in the field of nanoscience could be obtained by using Mössbauer spectroscopy with a high velocity resolution as follows:

(i) In the case of ferritins and their pharmaceutical analogues, a new model taking into consideration five different Fe microenvironments assigned to various layers/regions with different nano-FeOOH packings at the iron core of ferritins (and its pharmaceutical analogues) was given by which the anomalous temperature dependence of Mössbauer parameters was also explained.

(ii) In the case of nickel, copper and magnesium ferrite nanoparticles, the cation distributions at the tetrahedral (A) and octahedral [B] sites in spinels were determined for different compositions, and the random distributions of Ni^2+^ (Cu^2+^ or Mg^2+^) and Fe^3+^ at the (A) and [B] sites were proven.

(iii) In the case of the FINEMET alloy with nano-crystalline phases, ensuring very extreme good soft magnetic properties, two new Fe-microenvironments in the DO_3_ structure, which were suggested but never observed with conventional Mössbauer spectroscopy, were shown.

Therefore, further development of detailed physical models for these materials as well as further thorough studies of nanomaterials using Mössbauer spectroscopy with a high velocity resolution can bring more useful information about the magnetic properties of nanosized and nanostructured materials.

## Figures and Tables

**Figure 1 nanomaterials-12-03748-f001:**
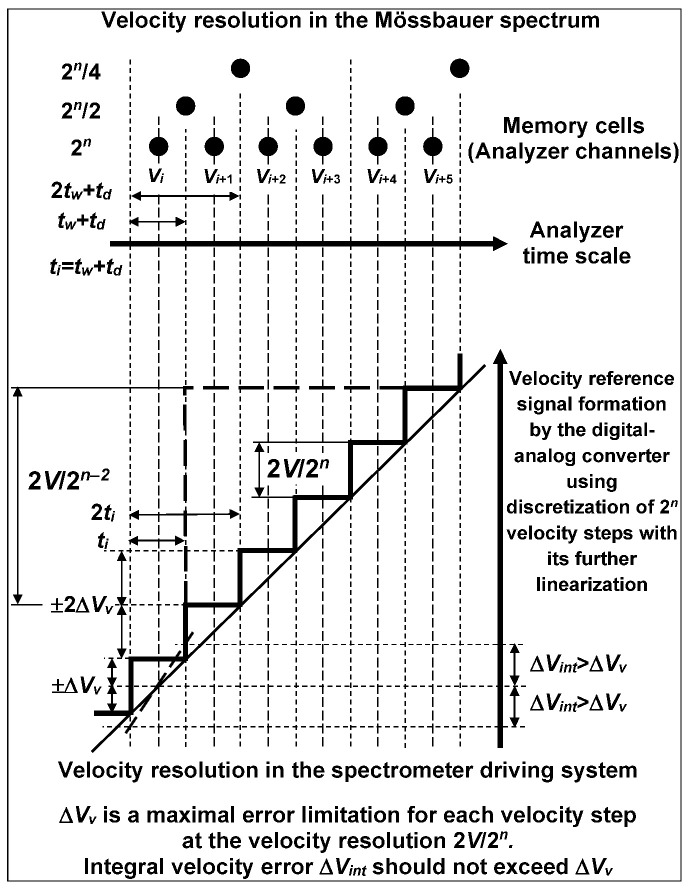
Scheme of the velocity reference signal formation in the Mössbauer spectrometer and in the spectrum. Adapted from Ref. [31]. (Reproduced from “Mössbauer spectroscopy with a high velocity resolution: Principles and applications”, M.I. Oshtrakh and V.A. Semionkin; AIP Conference Proceedings *1781*, 020019 (2016), with the permission of AIP Publishing).

**Figure 2 nanomaterials-12-03748-f002:**
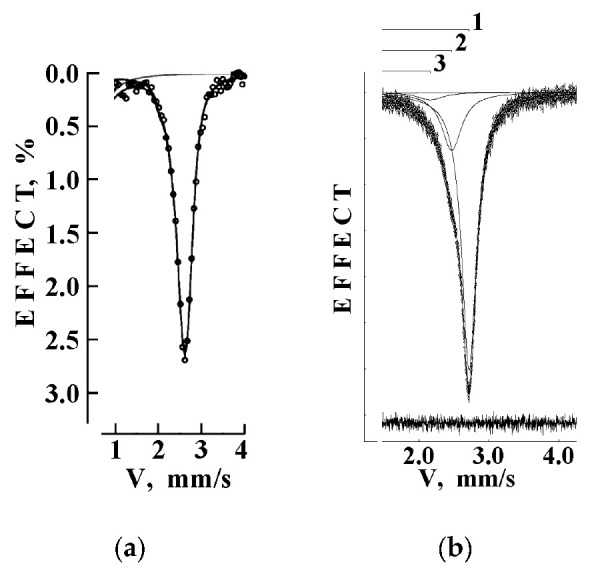
Comparison of the line shapes of the most positive peak in the Mössbauer spectra of Fe-gluconate in Ascofer^®^ measured in 128 channels (**a**) and in 4096 channels (the differential spectrum is shown on the bottom) (**b**). Adapted from Ref. [31]. (Reproduced from “Mössbauer spectroscopy with a high velocity resolution: Principles and applications”, M.I. Oshtrakh and V.A. Semionkin; AIP Conference Proceedings *1781*, 020019 (2016), with the permission of AIP Publishing).

**Figure 3 nanomaterials-12-03748-f003:**
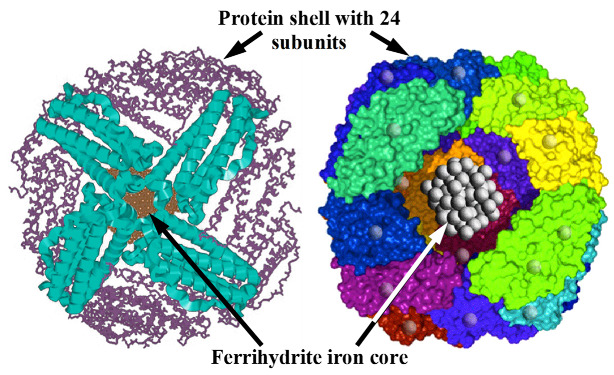
Two schematical views on the ferritin structure with indications of protein multi-subunit shell and the iron core.

**Figure 4 nanomaterials-12-03748-f004:**
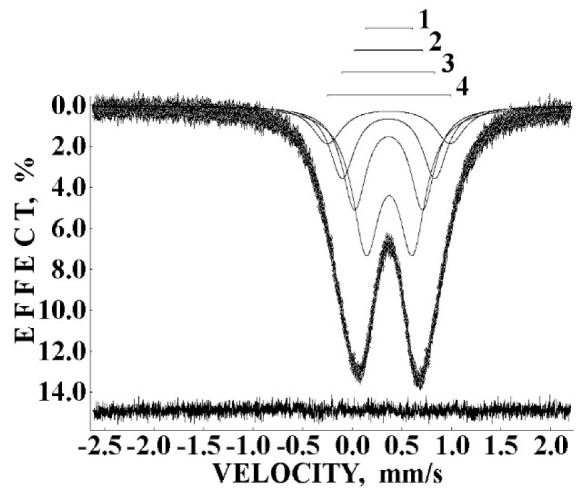
Mössbauer spectrum of human liver ferritin (200 mg) measured in 4096 channels at 295 K. Components 1–4 are the result of the best fit with free variation of parameters. The differential spectrum is shown on the bottom. Adapted from Ref. [39] with permission from Springer Nature: Hyperfine Interactions, Mössbauer spectroscopy with high velocity resolution in the study of ferritin and Imferon: preliminary results, Oshtrakh M.I., Milder O.B., Semionkin V.A., ©2008.

**Figure 5 nanomaterials-12-03748-f005:**
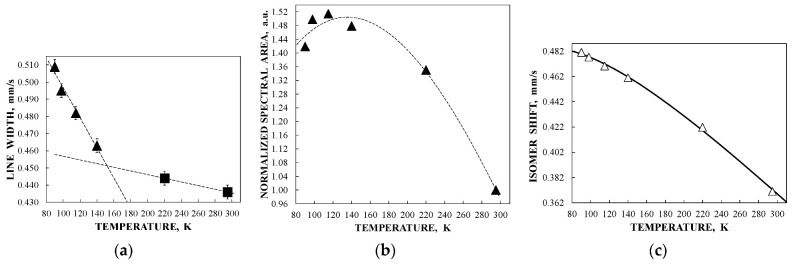
Anomalous temperature dependencies of the line width obtained within one doublet fit (two dashed lines indicate two different slopes, ■ and ▲ are the experimental data) (**a**) and normalized total spectral areas (dashed line is polynomial approximation for the experimental values ▲) (**b**); and estimation of Θ_M_ for ferritin from temperature dependence of isomer shifts (△) obtained within one doublet fits (solid line is the fit using Equation (1)) (**c**). Adapted from Ref. [45].

**Figure 6 nanomaterials-12-03748-f006:**
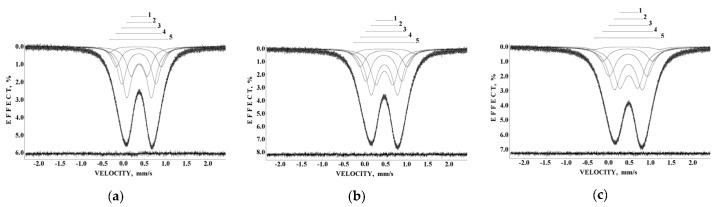
Mössbauer spectra of human liver ferritin (100 mg) measured in 4096 channels at 295 K (**a**), 140 K (**b**) and 90 K (**c**). The indicated five components (1–5) are the results of the best fits within the new heterogeneous model for the iron core with equal line width for five components. The differential spectra are shown on the bottom. Adapted from Ref. [49] with permission from Elsevier: Spectrochimica Acta, Part A: Molecular and Biomolecular Spectroscopy, Vol. 172, Oshtrakh M.I., Alenkina I.V., Klencsár Z., Kuzmann E., Semionkin V.A., “Different ^57^Fe microenvironments in the nanosized iron cores in human liver ferritin and its pharmaceutical analogues on the basis of temperature dependent Mössbauer spectroscopy”, 14–24, ©2017.

**Figure 7 nanomaterials-12-03748-f007:**
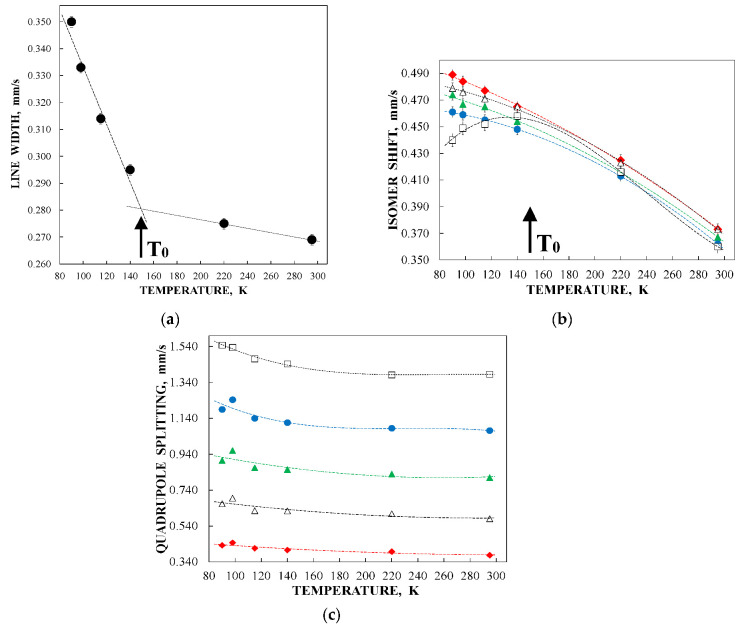
Temperature dependencies of: (**a**) the line width (● is the line width for components 1–5, dashed lines indicate two different slopes before and after critical temperature *T*_0_), (**b**) isomer shift and (**c**) quadrupole splitting for components 1–5 revealed in the ferritin Mössbauer spectra (♦ is component 1, △ is component 2, ▲ is component 3, ● is component 4 and ☐ is component 5; dashed lines are polynomial approximations). Adapted from Ref. [49] with permission from Elsevier: Spectrochimica Acta, Part A: Molecular and Biomolecular Spectroscopy, Vol. 172, Oshtrakh M.I., Alenkina I.V., Klencsár Z., Kuzmann E., Semionkin V.A., “Different ^57^Fe microenvironments in the nanosized iron cores in human liver ferritin and its pharmaceutical analogues on the basis of temperature dependent Mössbauer spectroscopy”, 14–24, ©2017.

**Figure 8 nanomaterials-12-03748-f008:**
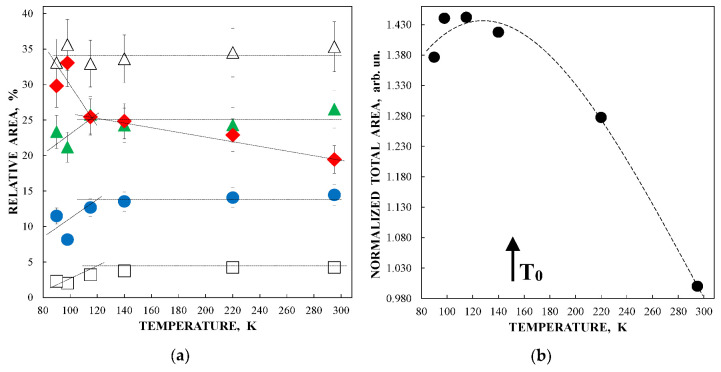
Temperature dependencies of: (**a**) the relative areas for components 1–5 revealed in the ferritin Mössbauer spectra (♦ is component 1, △ is component 2, ▲ is component 3, ● is component 4 and ☐ is component 5; dashed lines indicate different slopes above and below critical temperature) and (**b**) normalized total spectrum areas (●, dashed line indicates polynomial approximation, arrow indicates critical temperature *T*_0_). Adapted from Ref. [49] with permission from Elsevier: Spectrochimica Acta, Part A: Molecular and Biomolecular Spectroscopy, Vol. 172, Oshtrakh M.I., Alenkina I.V., Klencsár Z., Kuzmann E., Semionkin V.A., “Different ^57^Fe microenvironments in the nanosized iron cores in human liver ferritin and its pharmaceutical analogues on the basis of temperature dependent Mössbauer spectroscopy”, 14–24, ©2017.

**Figure 9 nanomaterials-12-03748-f009:**
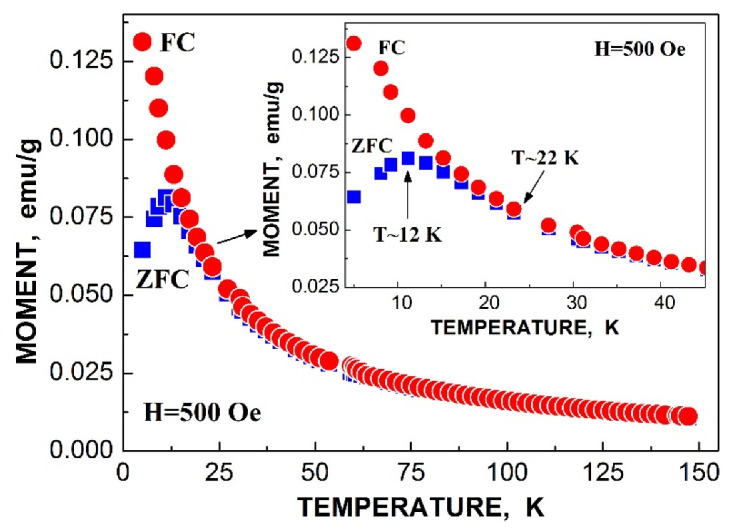
Zero-field-cooled (ZFC) and field-cooled (FC) curves for human liver ferritin iron cores. Adapted from Ref. [51] with permission from Elsevier: Journal of Inorganic Biochemistry, Vol. 213, Alenkina I.V., Kovacs Kis V., Felner I., Kuzmann E., Klencsár Z., Oshtrakh M.I., “Structural and magnetic study of the iron cores in iron(III)-polymaltose pharmaceutical ferritin analogue Ferrifol^®^.”, 1112020, ©2020.

**Figure 10 nanomaterials-12-03748-f010:**
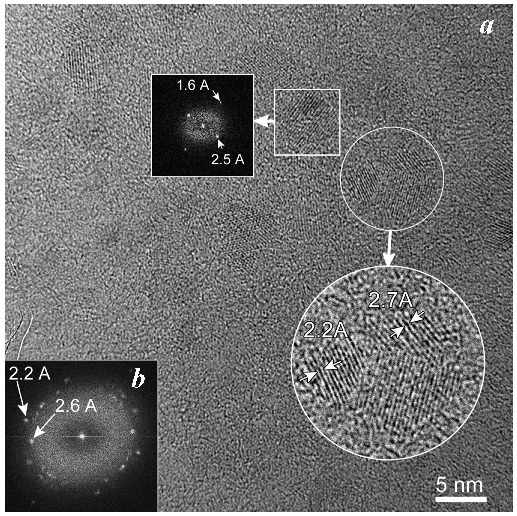
High resolution transmission electron microscopy image of ferritin (**a**) and the corresponding Fourier transform of 50 × 50 nm area (**b**). Some iron cores are enlarged and are shown in the insets. The iron cores in this image constitute ordered domains with lattice periodicities between 2.0 and 2.7 Å. Adapted from Ref. [51] with permission from Elsevier: Journal of Inorganic Biochemistry, Vol. 213, Alenkina I.V., Kovacs Kis V., Felner I., Kuzmann E., Klencsár Z., Oshtrakh M.I., “Structural and magnetic study of the iron cores in iron(III)-polymaltose pharmaceutical ferritin analogue Ferrifol^®^.”, 1112020, ©2020.

**Figure 11 nanomaterials-12-03748-f011:**
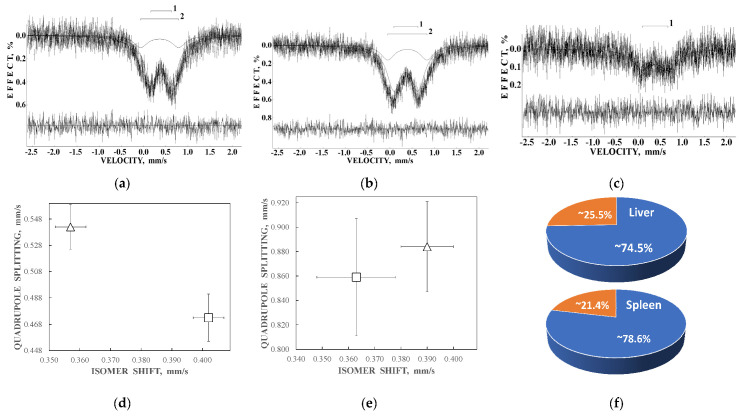
Mössbauer spectra of normal chicken liver (**a**) and spleen (**b**) fitted within the simple heterogeneous model for the ferritin-like iron cores using two doublets 1 and 2, and lymphoid lymphoma chicken spleen (**c**) roughly fitted using one quadrupole doublet. The differential spectra are shown on the bottom (adapted from Ref. [53] with permission from Elsevier: Journal of Molecular Structure, Vol. 924–926, Oshtrakh M.I., Semionkin V.A., Milder O.B., Novikov E.G., “Mössbauer spectroscopy with high velocity resolution: new possibilities in biomedical research”, 20–26, ©2009); plots of the Mössbauer hyperfine parameters for doublet 1 (**d**) and doublet 2 (**e**), where ☐ is normal spleen, △ is normal liver; and comparison of the relative areas of doublet 1 (■) and doublet 2 (■) in the Mössbauer spectra of normal chicken liver and spleen (**f**).

**Figure 12 nanomaterials-12-03748-f012:**
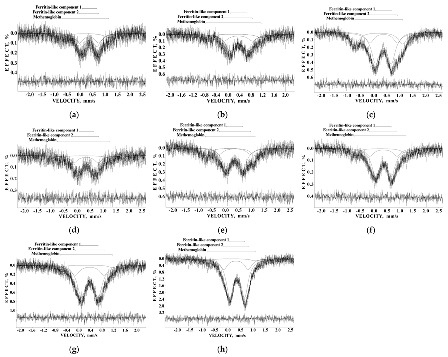
Mössbauer spectra of spleen tissues from normal human “1” (**a**) and patients with mantle cell lymphoma “N” (**b**) and “G” (**c**), marginal zone lymphoma “De” (**d**), “Do” (**e**) and “P” (**f**), acute myeloid leukemia “Ch” (**g**), and primary myelofibrosis “B” (**h**) fitted within the simple heterogeneous model for the ferritin-like iron cores (doublets 1 and 2) and component 3 for residual methemoglobin. The differential spectra are shown on the bottom. Adapted from Ref. [59] with permission from Springer Nature: Cell Biochemistry and Biophysics, “The iron state in spleen and liver tissues from patients with hematological malignancies studied using magnetization measurements and Mössbauer spectroscopy”, Alenkina I.V., Vinogradov A.V., Felner I., Konstantinova T.S., Kuzmann E., Semionkin V.A., Oshtrakh M.I., ©2019.

**Figure 13 nanomaterials-12-03748-f013:**
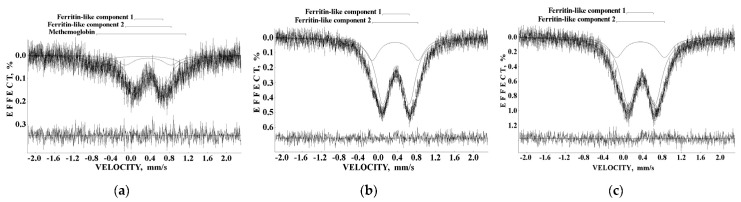
Mössbauer spectra of liver tissues from normal human “2” (**a**) and patients with mantle cell lymphoma “N” (**b**) and acute myeloid leukemia “Ch” (**c**) fitted within the simple heterogeneous model for the ferritin-like iron cores (doublets 1 and 2) and component 3 for residual methemoglobin. The differential spectra are shown on the bottom. Adapted from Ref. [59] with permission from Springer Nature: Cell Biochemistry and Biophysics, “The iron state in spleen and liver tissues from patients with hematological malignancies studied using magnetization measurements and Mössbauer spectroscopy”, Alenkina I.V., Vinogradov A.V., Felner I., Konstantinova T.S., Kuzmann E., Semionkin V.A., Oshtrakh M.I., ©2019.

**Figure 14 nanomaterials-12-03748-f014:**
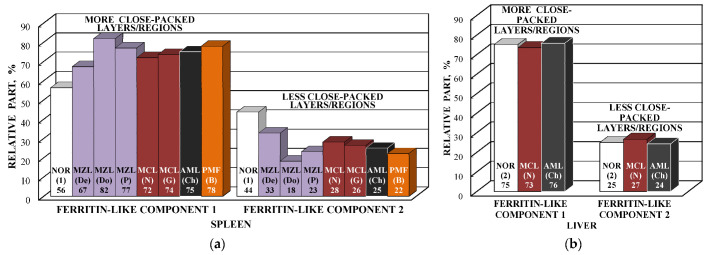
Rough estimation of more and less close-packed layers/regions in the nanosized ferritin-like iron cores in spleen (**a**) and liver (**b**) tissues from healthy human and patients with hematological malignancies based on the relative areas of ferritin-like components 1 and 2 in the corresponding Mössbauer spectra. Adapted from Ref. [59] with permission from Springer Nature: Cell Biochemistry and Biophysics, “The iron state in spleen and liver tissues from patients with hematological malignancies studied using magnetization measurements and Mössbauer spectroscopy”, Alenkina I.V., Vinogradov A.V., Felner I., Konstantinova T.S., Kuzmann E., Semionkin V.A., Oshtrakh M.I., ©2019.

**Figure 15 nanomaterials-12-03748-f015:**
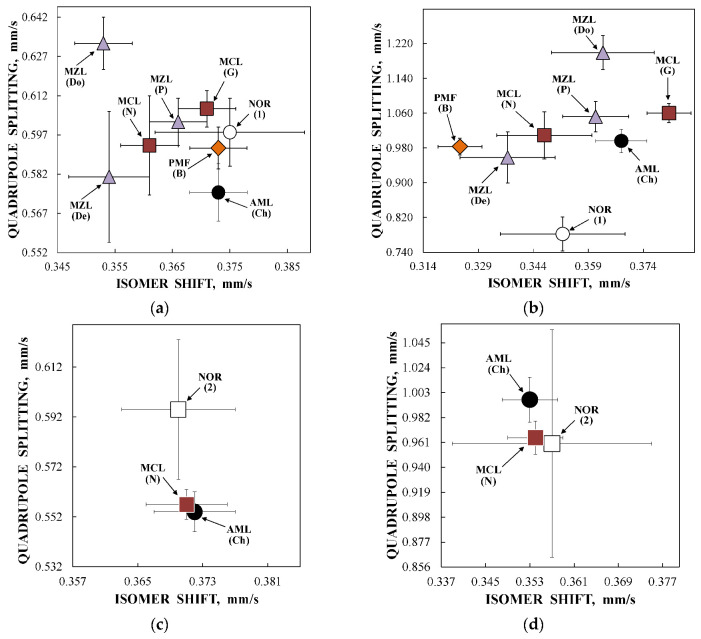
Plots of the ^57^Fe hyperfine parameters for ferritin-like components 1 (**a**,**c**) and 2 (**b**,**d**) in the Mössbauer spectra of spleen (**a**,**b**) and liver (**c**,**d**) tissues from healthy human and patients with hematological malignancies. Adapted from Ref. [59] with permission from Springer Nature: Cell Biochemistry and Biophysics, “The iron state in spleen and liver tissues from patients with hematological malignancies studied using magnetization measurements and Mössbauer spectroscopy”, Alenkina I.V., Vinogradov A.V., Felner I., Konstantinova T.S., Kuzmann E., Semionkin V.A., Oshtrakh M.I., ©2019.

**Figure 16 nanomaterials-12-03748-f016:**
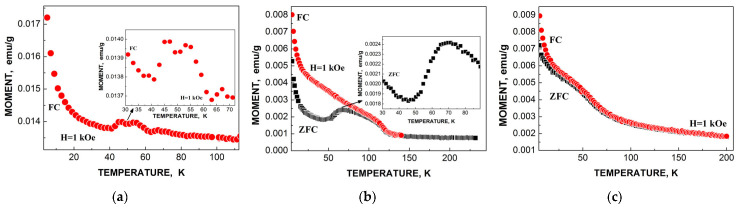
Selected zero-field-cooled (ZFC) and field-cooled (FC) curves of spleen tissues from normal human “1” (**a**) and patients with mantle cell lymphoma “G” (**b**), marginal zone lymphoma “P” (**c**) and “Do” (**d**), acute myeloid leukemia “Ch” (**e**) and primary myelofibrosis “B” (**f**). Adapted from Ref. [59] with permission from Springer Nature: Cell Biochemistry and Biophysics, “The iron state in spleen and liver tissues from patients with hematological malignancies studied using magnetization measurements and Mössbauer spectroscopy”, Alenkina I.V., Vinogradov A.V., Felner I., Konstantinova T.S., Kuzmann E., Semionkin V.A., Oshtrakh M.I., ©2019.

**Figure 17 nanomaterials-12-03748-f017:**
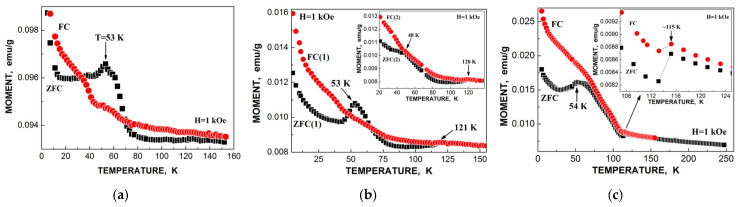
Selected zero-field-cooled (ZFC) and field-cooled (FC) curves of liver tissues from normal human “2” measured 7 months after the first run (**a**) and patients with mantle cell lymphoma “N” with the first run marked (1) and inset with the second run marked (2) (**b**) and acute myeloid leukemia “Ch” (**c**). Adapted from Ref. [59] with permission from Springer Nature: Cell Biochemistry and Biophysics, “The iron state in spleen and liver tissues from patients with hematological malignancies studied using magnetization measurements and Mössbauer spectroscopy”, Alenkina I.V., Vinogradov A.V., Felner I., Konstantinova T.S., Kuzmann E., Semionkin V.A., Oshtrakh M.I., ©2019; and from Ref. [60] with permission from Elsevier: Journal of Magnetism and Magnetic Materials, Vol. 399, Felner I., Alenkina I.V., Vinogradov A.V., Oshtrakh M.I., “Peculiar magnetic observations in pathological human liver”, 118–122, ©2016.

**Figure 18 nanomaterials-12-03748-f018:**
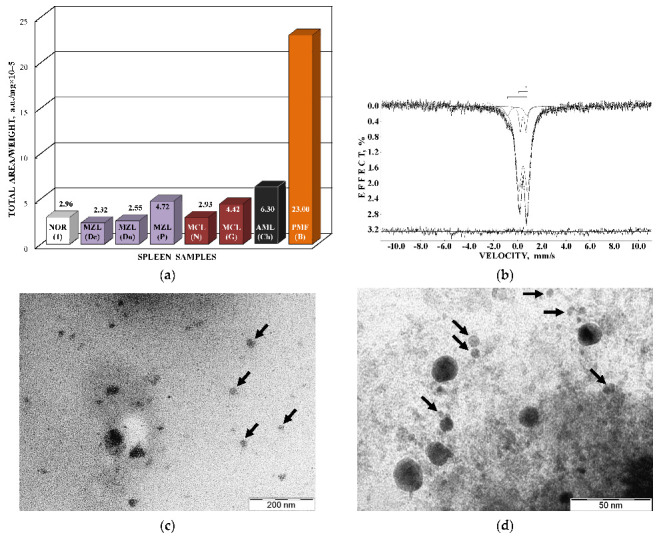
Plot of the total areas of ferritin-like components 1 and 2 in the Mössbauer spectra of spleen tissues from healthy human and patients with hematological malignancies normalized by the sample weight (**a**), the Mössbauer spectrum of spleen tissue from patient “B” with primary myelofibrosis measured at 90 K and fitted using three quadrupole doublets, the differential spectrum is shown on the bottom (**b**) and transmission electron microscopy images of spleen tissues from healthy human “1” (**c**) and patient “B” with primary myelofibrosis (**d**), possible ferritin iron cores are indicated by arrows in (**c**,**d**). Adapted from Ref. [55] with permission from Springer Nature: Biometals, “Mössbauer spectroscopy of the iron cores in human liver ferritin, ferritin in normal human spleen and ferritin in spleen from patient with primary myelofibrosis: preliminary results of comparative analysis”, Oshtrakh M.I., Alenkina I.V., Vinogradov A.V., Konstantinova T.S., Kuzmann E., Semionkin V.A., ©2013; from Ref. [59] with permission from Springer Nature: Cell Biochemistry and Biophysics, “The iron state in spleen and liver tissues from patients with hematological malignancies studied using magnetization measurements and Mössbauer spectroscopy”, Alenkina I.V., Vinogradov A.V., Felner I., Konstantinova T.S., Kuzmann E., Semionkin V.A., Oshtrakh M.I., ©2019; and from Ref. [60] with permission from Elsevier: Journal of Magnetism and Magnetic Materials, Vol. 399, Felner I., Alenkina I.V., Vinogradov A.V., Oshtrakh M.I., “Peculiar magnetic observations in pathological human liver”, 118–122, ©2016.

**Figure 19 nanomaterials-12-03748-f019:**
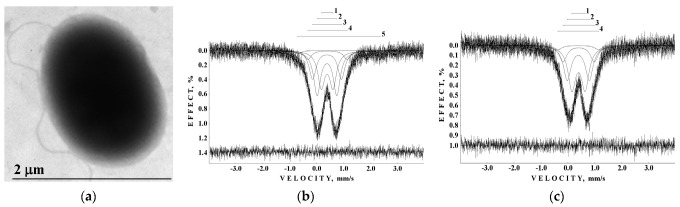
Transmission electron microscopy image of the rhizobacterium *Azospirillum brasilense* (strain Sp245) (**a**) and Mössbauer spectra of bacteria samples 1 (**b**) and 2 (**c**). Indicated components 1–4 are the results of the best fits within the new heterogeneous model for the iron core with equal line width for four ferritin-like components. Component 5 in the spectrum **b** corresponds to ferrous compound. The differential spectra are shown on the bottom. Adapted from Ref. [64] with permission from Elsevier: Journal of Molecular Structure, Vol. 1073, Alenkina I.V., Oshtrakh M.I., Tugarova A.V., Biró B., Semionkin V.A., Kamnev A.A., “Study of the rhizobacterium *Azospirillum brasilense* Sp245 using Mössbauer spectroscopy with a high velocity resolution: implication for the analysis of ferritin-like iron cores”, 181–186, ©2014.

**Figure 20 nanomaterials-12-03748-f020:**
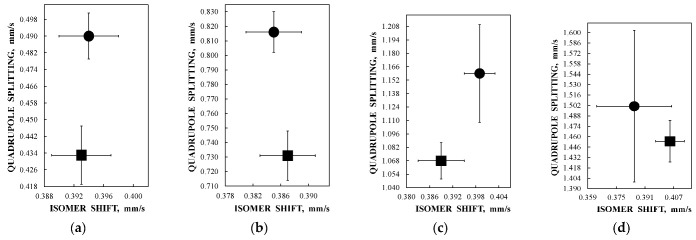
Plots of quadrupole splitting vs. isomer shift for quadrupole doublets 1 (**a**), 2 (**b**), 3 (**c**) and 4 (**d**) revealed in the Mössbauer spectra of the rhizobacterium *Azospirillum brasilense* (strain Sp245) samples 1 (■) and 2 (●) within the new heterogeneous model for the iron core. Data were taken from Ref. [64].

**Figure 21 nanomaterials-12-03748-f021:**
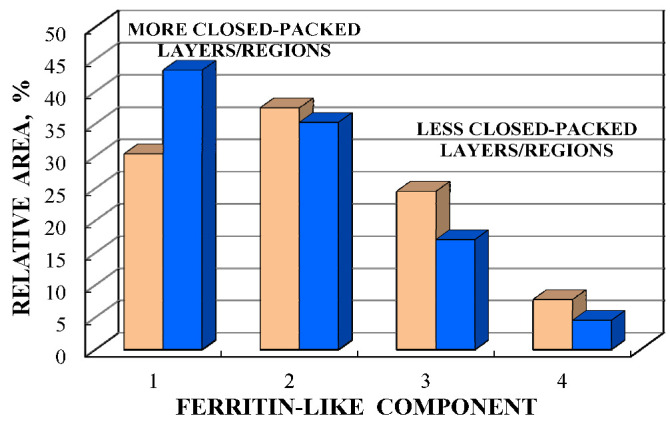
A comparison of the relative areas of the ferritin-like components 1–4 in the Mössbauer spectra of the rhizobacterium *Azospirillum brasilense* (strain Sp245) samples 1 (■) and 2 (■). Data for sample 1 were recalculated accounting for the non-ferritin-like component 5. Adapted from Ref. [64] with permission from Elsevier: Journal of Molecular Structure, Vol. 1073, Alenkina I.V., Oshtrakh M.I., Tugarova A.V., Biró B., Semionkin V.A., Kamnev A.A., “Study of the rhizobacterium *Azospirillum brasilense* Sp245 using Mössbauer spectroscopy with a high velocity resolution: Implication for the analysis of ferritin-like iron cores”, 181–186, ©2014.

**Figure 22 nanomaterials-12-03748-f022:**
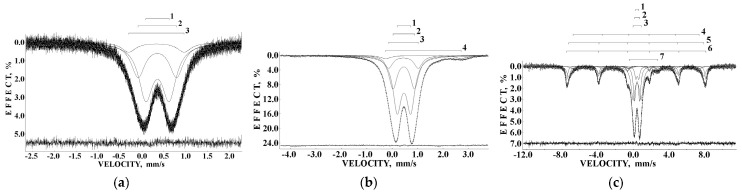
Mössbauer spectra of lyophilized Imferon at 295 K (**a**), initial Imferon in frozen solution at 87 K (**b**) and Imferon in frozen solution measured 10 years after sample (**b**) at 90 K (**c**). Indicated components in the spectra (**a**,**b**) are the results of the best fits within the new heterogeneous model for the iron core with equal line width for three components 1–3 assigned to the iron cores and residual FeCl_2_ (4) while the spectrum (**c**) was fitted with 7 component using an old heterogeneous iron core model with free variation of parameters (all parameters are given in Table 1). The differential spectra are shown on the bottom. (**b**,**c**) were adapted from Ref. [41] with permission from Elsevier: Journal of Molecular Structure, Vol. 993, Oshtrakh M.I., Alenkina I.V., Dubiel S.M., Semionkin V.A., “Structural variations of the iron cores in human liver ferritin and its pharmaceutically important models: a comparative study using Mössbauer spectroscopy with a high velocity resolution”, 287–291, ©2011.

**Figure 23 nanomaterials-12-03748-f023:**
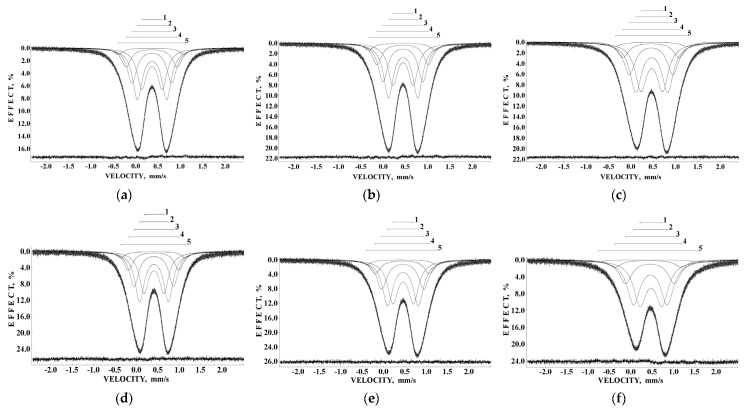
Selected Mössbauer spectra of Ferrum Lek (**a**–**c**), Maltofer^®^ (**d**–**f**) and Ferrifol (**g**–**i**) measured with a high velocity resolution in 4096 channels at different temperatures: 295 K (**a**), 170 K (**b**), 105 K (**c**), 220 K (**d**), 125 K (**e**), 90 K (**f**), 295 K (**g**), 140 K (**h**) and 98 K (**i**). These spectra were fitted within the new heterogeneous model for the iron cores (quadrupole doublets 1–5). The differential spectra are shown on the bottom. Adapted from Ref. [49] with permission from Elsevier: Spectrochimica Acta, Part A: Molecular and Biomolecular Spectroscopy, Vol. 172, Oshtrakh M.I., Alenkina I.V., Klencsár Z., Kuzmann E., Semionkin V.A., “Different ^57^Fe microenvironments in the nanosized iron cores in human liver ferritin and its pharmaceutical analogues on the basis of temperature dependent Mössbauer spectroscopy”, 14–24, ©2017; and from Ref. [51] with permission from Elsevier: Journal of Inorganic Biochemistry, Vol. 213, Alenkina I.V., Kovacs Kis V., Felner I., Kuzmann E., Klencsár Z., Oshtrakh M.I., “Structural and magnetic study of the iron cores in iron(III)-polymaltose pharmaceutical ferritin analogue Ferrifol^®^.”, 1112020, ©2020.

**Figure 24 nanomaterials-12-03748-f024:**
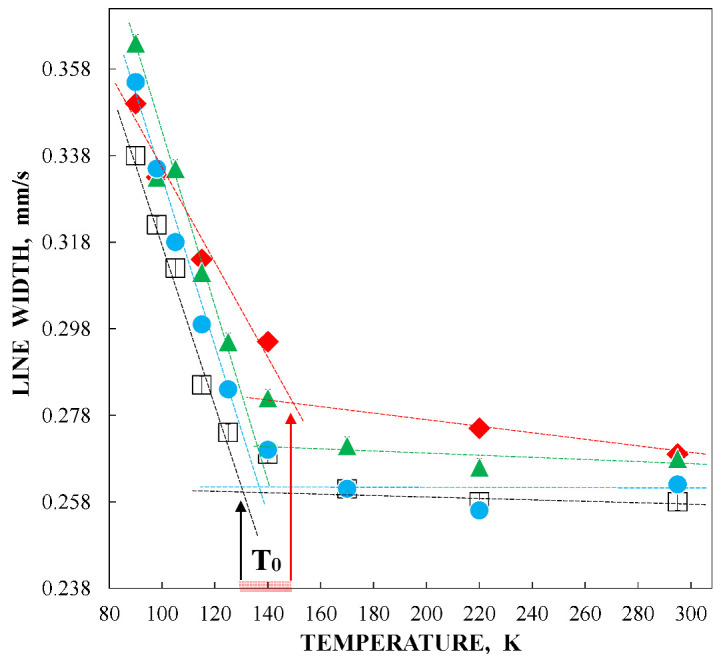
Unusual temperature dependencies of the line widths for the spectral components 1–5 obtained within the new heterogeneous iron core model used for fitting the Mössbauer spectra of ferritin (♦) and their analogues Ferrum Lek (☐), Maltofer^®^ (▲) and Ferrifol (●). Dashed lines indicate different slopes of the line broadening upper and below so-called critical temperature *T*_0_ (arrows indicate the range of *T*_0_ for the nanosized iron cores in ferritin and its analogues). Adapted from Ref. [50] with permission from Springer Nature: Hyperfine Interactions, “Unusual temperature dependencies of Mössbauer parameters of the nanosized iron cores in ferritin and its pharmaceutical analogues”, Alenkina I.V., Kuzmann E., Felner I., Kis V.K., Oshtrakh M.I., ©2021.

**Figure 25 nanomaterials-12-03748-f025:**
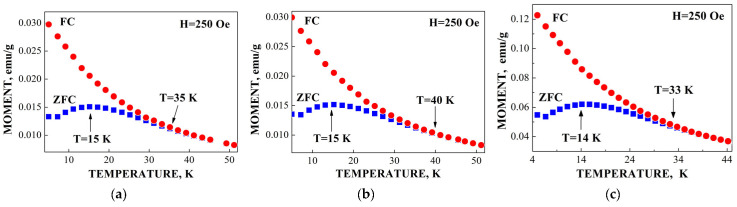
Zero-field-cooled (ZFC) and field-cooled (FC) curves for the iron cores in Ferrum Lek (**a**), Maltofer^®^ (**b**) and Ferrifol (**c**). Indicated temperatures are the blocking temperatures *T*_B_ for the largest iron cores (~35, ~40 and ~33 K) and the mean blocking temperatures (~15–14 K). Adapted from Ref. [50] with permission from Springer Nature: Hyperfine Interactions, “Unusual temperature dependencies of Mössbauer parameters of the nanosized iron cores in ferritin and its pharmaceutical analogues”, Alenkina I.V., Kuzmann E., Felner I., Kis V.K., Oshtrakh M.I., ©2021.

**Figure 26 nanomaterials-12-03748-f026:**
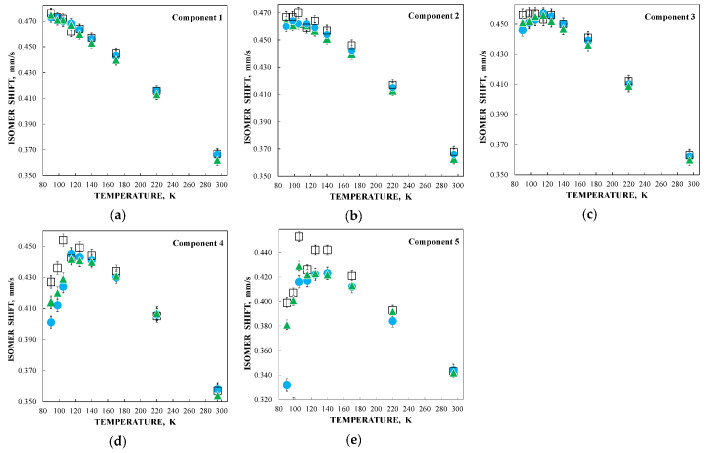
Unusual temperature dependencies of isomer shift for spectral components 1 (**a**), 2 (**b**), 3 (**c**), 4 (**d**) and 5 (**e**) obtained within the new heterogeneous iron core model used for fitting the Mössbauer spectra of Ferrum Lek (☐), Maltofer^®^ (▲) and Ferrifol (●). Adapted from Ref. [50] with permission from Springer Nature: Hyperfine Interactions, “Unusual temperature dependencies of Mössbauer parameters of the nanosized iron cores in ferritin and its pharmaceutical analogues”, Alenkina I.V., Kuzmann E., Felner I., Kis V.K., Oshtrakh M.I., ©2021.

**Figure 27 nanomaterials-12-03748-f027:**
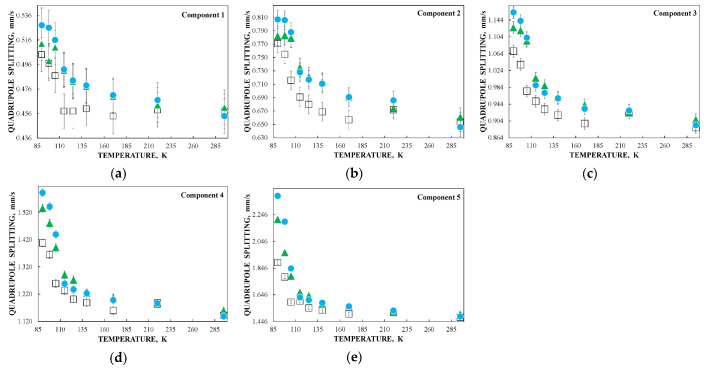
Unusual temperature dependencies of quadrupole splitting for spectral components 1 (**a**), 2 (**b**), 3 (**c**), 4 (**d**) and 5 (**e**) obtained within the new heterogeneous iron core model used for fitting the Mössbauer spectra of Ferrum Lek (☐), Maltofer^®^ (▲) and Ferrifol (●). Data were taken from Refs. [49,50,51].

**Figure 28 nanomaterials-12-03748-f028:**
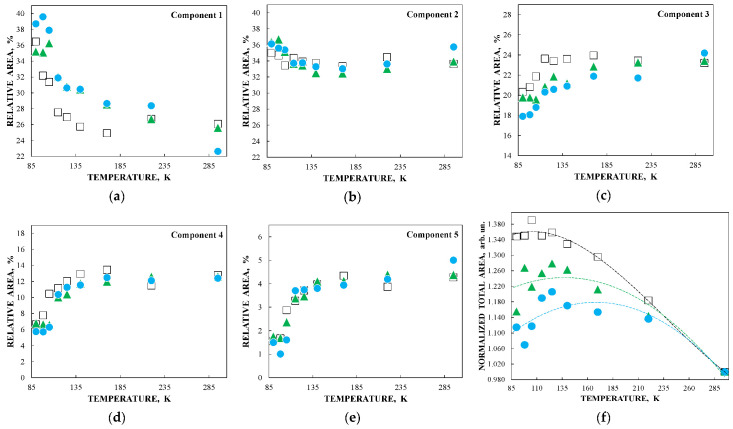
Unusual temperature dependencies of the relative areas for spectral components 1 (**a**), 2 (**b**), 3 (**c**), 4 (**d**) and 5 (**e**) and normalized total spectrum areas (**f**) obtained within the new heterogeneous iron core model used for fitting the Mössbauer spectra of Ferrum Lek (☐), Maltofer^®^ (▲) and Ferrifol^®^ (●). Data were taken from Refs. [49,50,51].

**Figure 29 nanomaterials-12-03748-f029:**
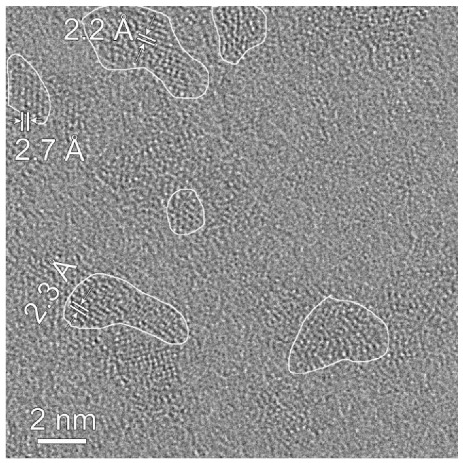
High resolution transmission electron microscopy image of Ferrifol^®^. The iron cores in this image constitute ordered domains with lattice periodicities between 2.2 and 2.7 Å. Adapted from Ref. [51] with permission from Elsevier: Journal of Inorganic Biochemistry, Vol. 213, Alenkina I.V., Kovacs Kis V., Felner I., Kuzmann E., Klencsár Z., Oshtrakh M.I. “Structural and magnetic study of the iron cores in iron(III)-polymaltose pharmaceutical ferritin analogue Ferrifol^®^.”, 1112020, ©2020.

**Figure 30 nanomaterials-12-03748-f030:**
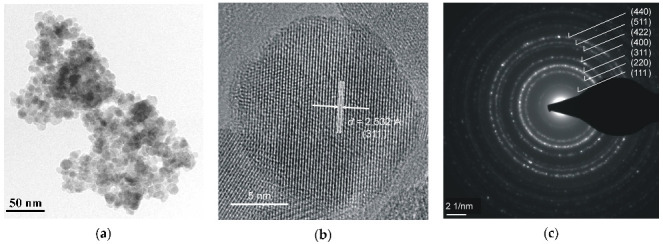
TEM (**a**) and HRTEM (**b**) images of the as-prepared Fe_3_O_4_ nanoparticles and SAED (selective area electron diffraction) pattern of the nanoparticles (**c**). Adapted from Ref. [70] with permission from Elsevier: Spectrochimica Acta, Part A: Molecular and Biomolecular Spectroscopy, Vol. 100, Oshtrakh M.I., Šepelák V., Rodriguez A.F.R., Semionkin V.A., Ushakov M.V., Santos J.G., Silveira L.B., Marmolejo E.M., De Souza Parise M., Morais P.C., “Comparative study of iron oxide nanoparticles as-prepared and dispersed in Copaiba oil using Mössbauer spectroscopy with low and high velocity resolution”, 94–100, ©2013.

**Figure 31 nanomaterials-12-03748-f031:**
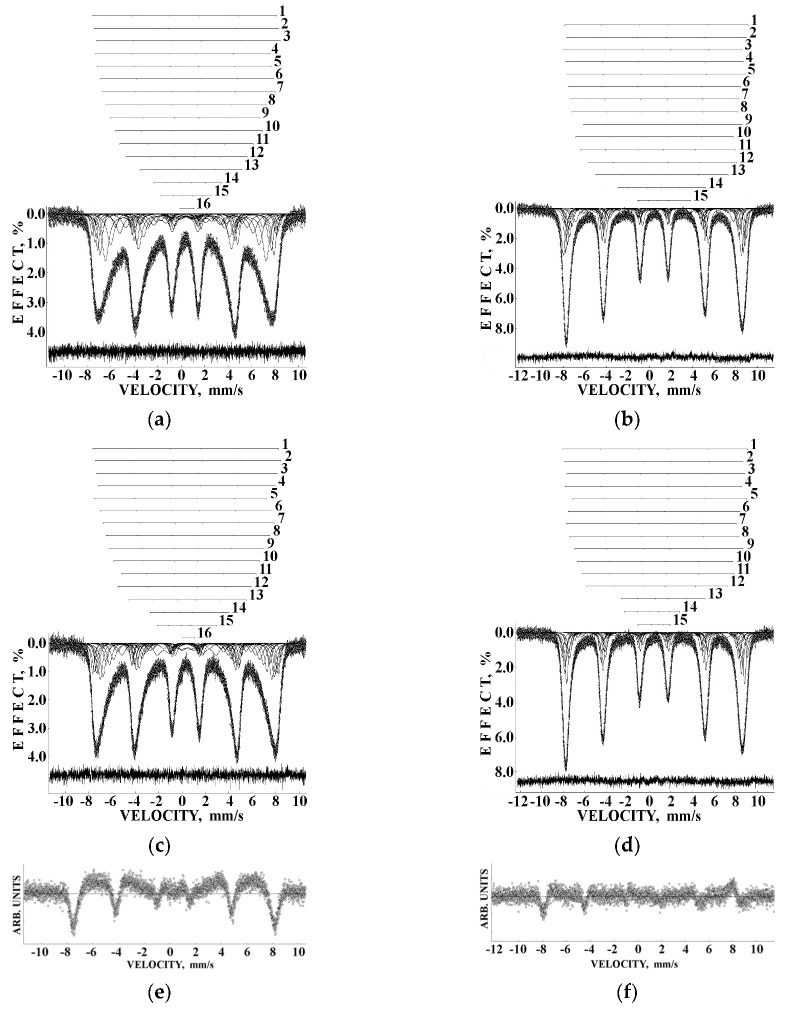
Mössbauer spectra of the as-prepared Fe_3_O_4_ nanoparticles (**a**,**b**) and those dispersed in Copaiba oil Fe_3_O_4_-CO (**c**,**d**) measured in 4096 channels at 295 K (**a**,**c**) and 90 K (**b**,**d**), and normalized differences between the Mössbauer spectra of the as-prepared Fe_3_O_4_ and dispersed in Copaiba oil Fe_3_O_4_–CO measured at 295 K (**e**) and 90 K (**f**). Indicated components 1–16 (**a**,**c**) and 1–15 (**b**,**d**) are the results of the best fits. The differential spectra are shown on the bottom (**a**–**d**). Adapted from Ref. [72] with permission from Elsevier: Spectrochimica Acta, Part A: Molecular and Biomolecular Spectroscopy, Vol. 152, Oshtrakh M.I., Ushakov M.V., Šepelák V., Semionkin V.A., Morais P.C., “Study of iron oxide nanoparticles using Mössbauer spectroscopy with a high velocity resolution”, 666–679, ©2016.

**Figure 32 nanomaterials-12-03748-f032:**
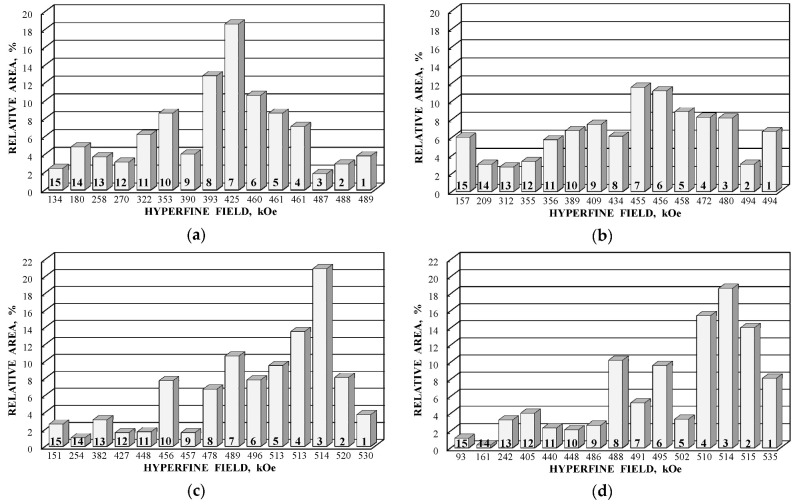
Histograms of the distributions of the relative areas of magnetic sextets in the order of *H*_eff_ increase for corresponding spectral components of the as-prepared Fe_3_O_4_ nanoparticles (**a**,**c**) and those dispersed in Copaiba oil Fe_3_O_4_-CO (**b**,**d**) at 295 K (**a**,**b**) and 90 K (**c**,**d**) (the numbers of sextets are the same as indicated in Figure 31). Adapted from Ref. [72] with permission from Elsevier: Spectrochimica Acta, Part A: Molecular and Biomolecular Spectroscopy, Vol. 152, Oshtrakh M.I., Ushakov M.V., Šepelák V., Semionkin V.A., Morais P.C., “Study of iron oxide nanoparticles using Mössbauer spectroscopy with a high velocity resolution”, 666–679, ©2016.

**Figure 33 nanomaterials-12-03748-f033:**
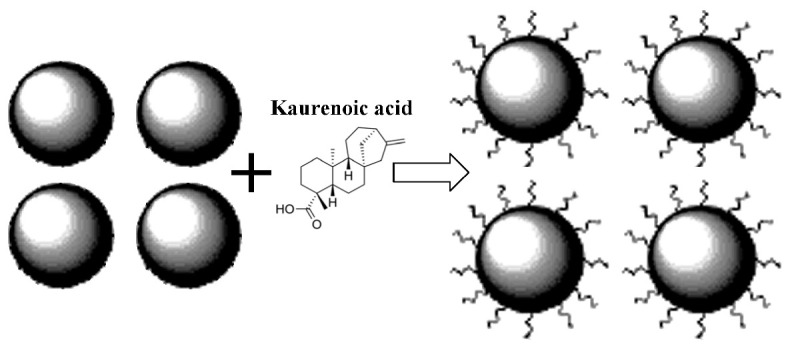
Scheme of the interaction between the as-prepared magnetite nanoparticles and kaurenoic acid (containing polar groups) after dispersion in Copaiba oil. Adapted from Ref. [72] with permission from Elsevier: Spectrochimica Acta, Part A: Molecular and Biomolecular Spectroscopy, Vol. 152, Oshtrakh M.I., Ushakov M.V., Šepelák V., Semionkin V.A., Morais P.C., “Study of iron oxide nanoparticles using Mössbauer spectroscopy with a high velocity resolution”, 666–679, ©2016.

**Figure 34 nanomaterials-12-03748-f034:**
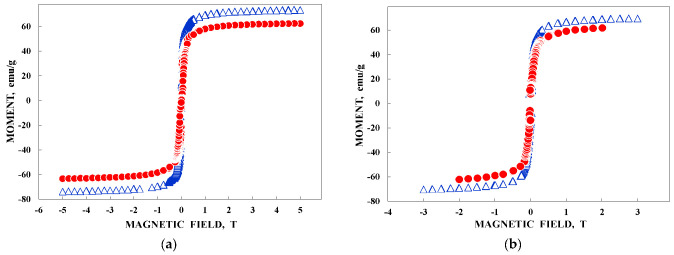
A decrease in the differences in magnetization for the as-prepared Fe_3_O_4_ (●) and dispersed in Copaiba oil Fe_3_O_4_-CO (Δ) samples with temperature decrease from 294 K (**a**) down to 8 K (**b**). Adapted from Ref. [71] with permission from Springer Nature: Hyperfine Interactions, “Magnetite nanoparticles as-prepared and dispersed in Copaiba oil: Study using magnetic measurements and Mössbauer spectroscopy”, Oshtrakh M.I., Ushakov M.V., Semenova A.S., Kellerman D.G., Šepelák V., Rodriguez A.F.R., Semionkin V.A., Morais P.C., ©2013.

**Figure 35 nanomaterials-12-03748-f035:**
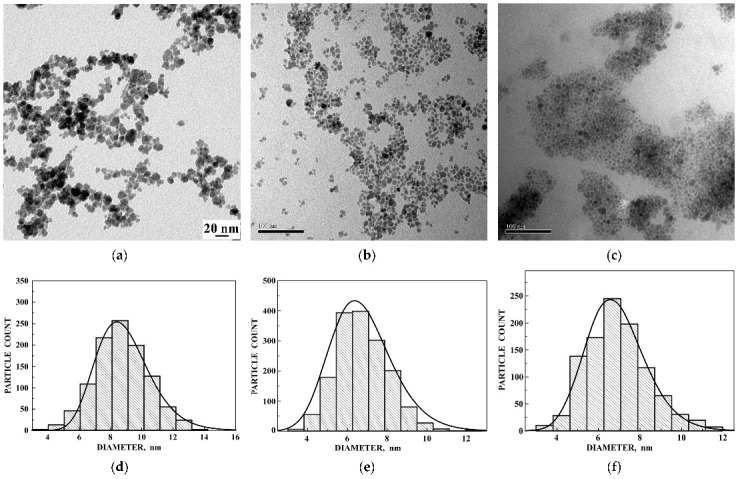
TEM images of the as-prepared γ-Fe_2_O_3_ (**a**), γ-Fe_2_O_3_/Fe_3_O_4_–IMF (**b**) and γ-Fe_2_O_3_/Fe_3_O_4_–OAT (**c**) nanoparticles with evaluations of particle size distributions for the as-prepared γ-Fe_2_O_3_ (**d**), γ-Fe_2_O_3_/Fe_3_O_4_–IMF (**e**) and γ-Fe_2_O_3_/Fe_3_O_4_–OAT (**f**). Adapted from Ref. [72] with permission from Elsevier: Spectrochimica Acta, Part A: Molecular and Biomolecular Spectroscopy, Vol. 152, Oshtrakh M.I., Ushakov M.V., Šepelák V., Semionkin V.A., Morais P.C., “Study of iron oxide nanoparticles using Mössbauer spectroscopy with a high velocity resolution”, 666–679, ©2016.

**Figure 36 nanomaterials-12-03748-f036:**
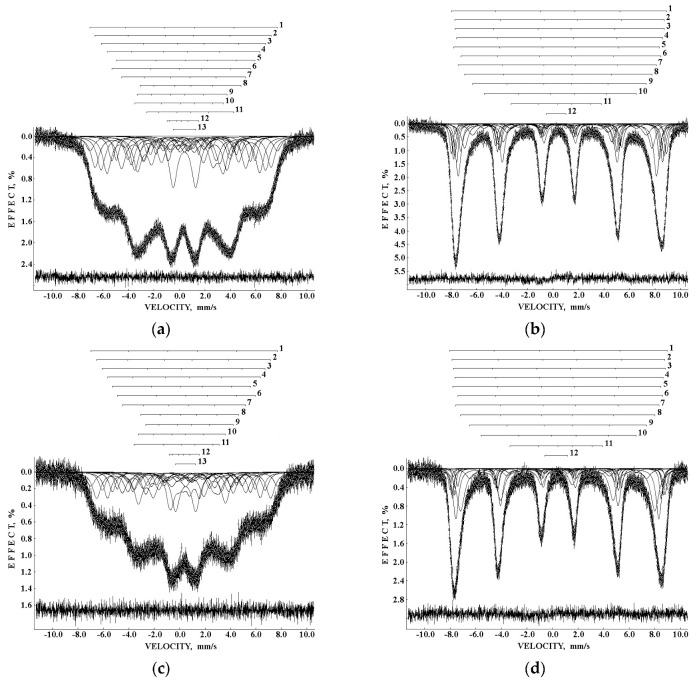
Mössbauer spectra of the as-prepared γ-Fe_2_O_3_ (**a**,**b**) and γ-Fe_2_O_3_–DMSA (**c**,**d**) nanoparticles measured at 295 K (**a**,**c**) and 90 K (**b**,**d**) and converted into 2048-channel spectra. Indicated components 1–13 (**a**,**c**) and 1–12 (**b**,**d**) are the results of the best fits. The differential spectra are shown on the bottom. Adapted from Ref. [72] with permission from Elsevier: Spectrochimica Acta, Part A: Molecular and Biomolecular Spectroscopy, Vol. 152, Oshtrakh M.I., Ushakov M.V., Šepelák V., Semionkin V.A., Morais P.C., “Study of iron oxide nanoparticles using Mössbauer spectroscopy with a high velocity resolution”, 666–679, ©2016.

**Figure 37 nanomaterials-12-03748-f037:**
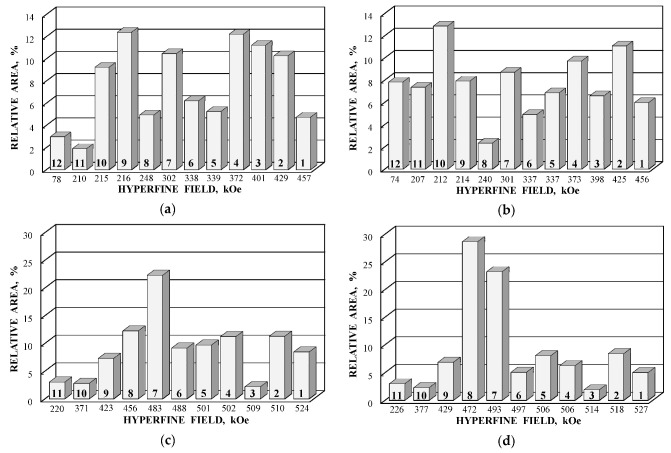
Histograms of the distributions of the relative areas of magnetic sextets in the order of increase in the *H*_eff_ values for corresponding spectral components of the as-prepared γ-Fe_2_O_3_ (**a**,**c**) and γ-Fe_2_O_3_–DMSA (**b**,**d**) samples at 295 K (**a**,**b**) and 90 K (**c**,**d**) (the numbers of sextets are the same as indicated in Figure 36). Adapted from Ref. [72] with permission from Elsevier: Spectrochimica Acta, Part A: Molecular and Biomolecular Spectroscopy, Vol. 152, Oshtrakh M.I., Ushakov M.V., Šepelák V., Semionkin V.A., Morais P.C., “Study of iron oxide nanoparticles using Mössbauer spectroscopy with a high velocity resolution”, 666–679, ©2016.

**Figure 38 nanomaterials-12-03748-f038:**
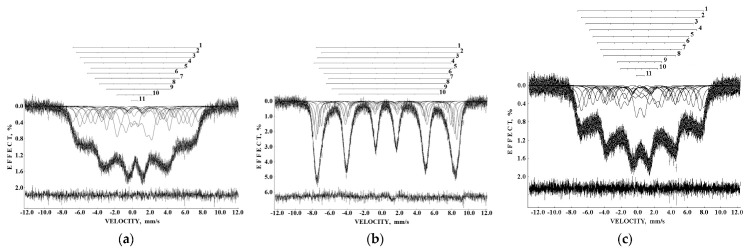
Mössbauer spectra of the γ-Fe_2_O_3_/Fe_3_O_4_–IMF (**a**,**b**) and γ-Fe_2_O_3_/Fe_3_O_4_–OAT (**c**) nanoparticles measured at 295 K (**a**,**c**) and 90 K (**b**) in 4096 channels and converted into 2048-channel spectra for (**a**,**b**). Indicated components 1–11 (**a**,**c**) and 1–10 (**b**) are the results of the best fits. The differential spectra are shown on the bottom. Adapted from Ref. [72] with permission from Elsevier: Spectrochimica Acta, Part A: Molecular and Biomolecular Spectroscopy, Vol. 152, Oshtrakh M.I., Ushakov M.V., Šepelák V., Semionkin V.A., Morais P.C., “Study of iron oxide nanoparticles using Mössbauer spectroscopy with a high velocity resolution”, 666–679, ©2016.

**Figure 39 nanomaterials-12-03748-f039:**
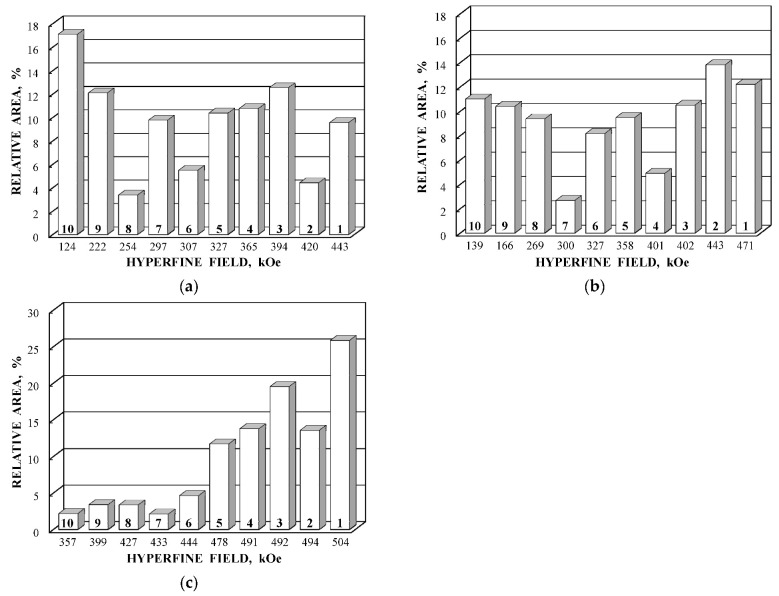
Histograms of the distributions of the relative areas of magnetic sextets in the order of increase in the *H*_eff_ values for corresponding spectral components of the γ-Fe_2_O_3_/Fe_3_O_4_–IMF (**a**, **b**) and γ-Fe_2_O_3_/Fe_3_O_4_–OAT (**c**) nanoparticles measured at 295 K (**a**,**c**) and 90 K (**b**) (the numbers of sextets are the same as indicated in Figure 38). Adapted from Ref. [72] with permission from Elsevier: Spectrochimica Acta, Part A: Molecular and Biomolecular Spectroscopy, Vol. 152, Oshtrakh M.I., Ushakov M.V., Šepelák V., Semionkin V.A., Morais P.C., “Study of iron oxide nanoparticles using Mössbauer spectroscopy with a high velocity resolution”, 666–679, ©2016.

**Figure 40 nanomaterials-12-03748-f040:**
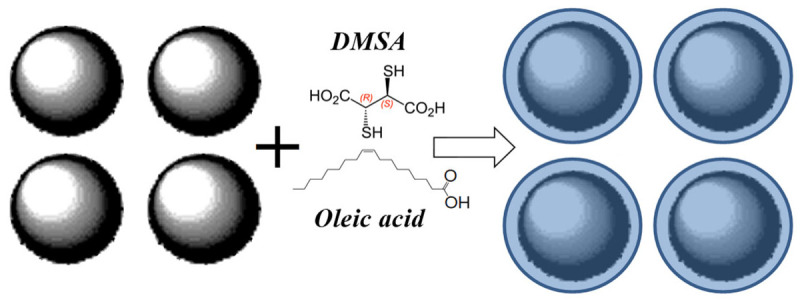
Scheme of the as-prepared γ-Fe_2_O_3_ or γ-Fe_2_O_3_/Fe_3_O_4_ nanoparticles coating with DMSA or oleic acid. Adapted from Ref. [72] with permission from Elsevier: Spectrochimica Acta, Part A: Molecular and Biomolecular Spectroscopy, Vol. 152, Oshtrakh M.I., Ushakov M.V., Šepelák V., Semionkin V.A., Morais P.C., “Study of iron oxide nanoparticles using Mössbauer spectroscopy with a high velocity resolution”, 666–679, ©2016.

**Figure 41 nanomaterials-12-03748-f041:**
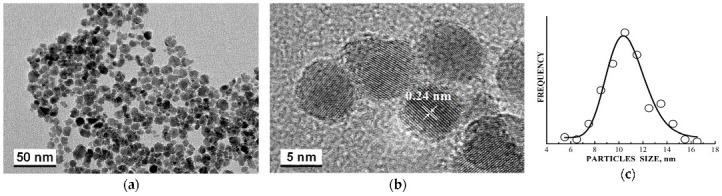
TEM and HRTEM images of FM as-prepared γ-Fe_2_O_3_ nanoparticles (**a**,**b**) and particle size distribution (**c**). Adapted from Ref. [76] with permission from Springer Nature: Hyperfine Interactions, “Effect of iron oxide nanoparticles functionalization by citrate analyzed using Mössbauer spectroscopy”, Ushakov M.V., Sousa M.H., Morais P.C., Kuzmann E., Semionkin V.A., Oshtrakh M.I., ©2020.

**Figure 42 nanomaterials-12-03748-f042:**
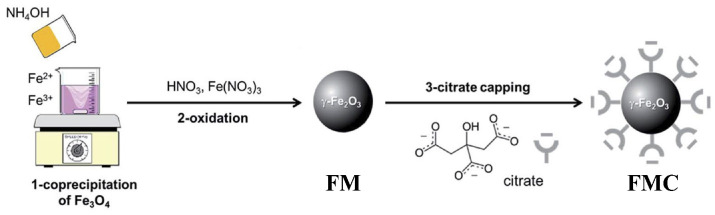
Scheme of the preparation of γ-Fe_2_O_3_ nanoparticles (FM) with further surface functionalization with citrate (FMC). Adapted from Ref. [76] with permission from Springer Nature: Hyperfine Interactions, “Effect of iron oxide nanoparticles functionalization by citrate analyzed using Mössbauer spectroscopy”, Ushakov M.V., Sousa M.H., Morais P.C., Kuzmann E., Semionkin V.A., Oshtrakh M.I., ©2020.

**Figure 43 nanomaterials-12-03748-f043:**
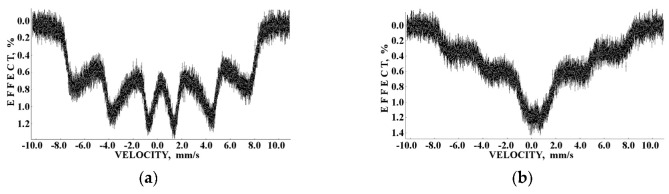
Mössbauer spectra of the FM (**a**) and FMC (**b**) nanoparticles measured at 295 K in 4096 channels. Adapted from Ref. [76] with permission from Springer Nature: Hyperfine Interactions, “Effect of iron oxide nanoparticles functionalization by citrate analyzed using Mössbauer spectroscopy”, Ushakov M.V., Sousa M.H., Morais P.C., Kuzmann E., Semionkin V.A., Oshtrakh M.I., ©2020.

**Figure 44 nanomaterials-12-03748-f044:**
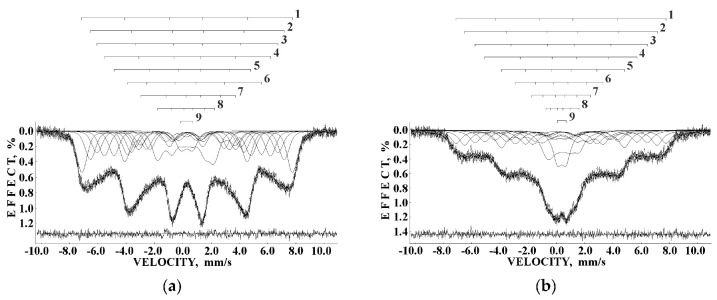
Mössbauer spectra of the FM (**a**) and FMC (**b**) nanoparticles measured at 295 K in 4096 channels and converted into 1024-channel spectra. Indicated components 1–9 are the results of the best fits. The differential spectra are shown on the bottom.

**Figure 45 nanomaterials-12-03748-f045:**
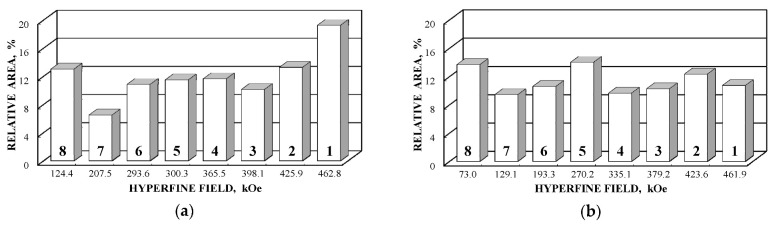
Histograms of the distributions of the relative areas of magnetic sextets in the order of increase in the *H*_eff_ values for corresponding spectral components revealed from the Mössbauer spectra of the FM (**a**) and FMC (**b**) nanoparticles measured at 295 K (the numbers of sextets are the same as indicated in Figure 44).

**Figure 46 nanomaterials-12-03748-f046:**
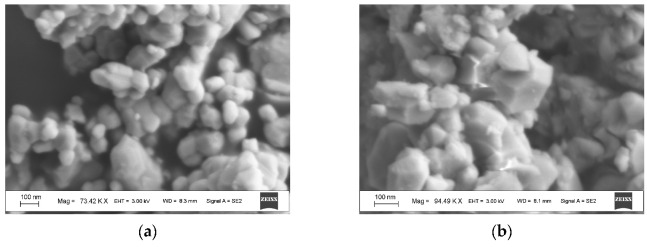
Scanning electron microscopy images of nickel ferrite nanoparticles: NA (**a**) and NB (**b**). Adapted from Ref. [79] with permission from Elsevier: Materials Chemistry and Physics, Vol. 202, Ushakov M.V., Senthilkumar B., Kalai Selvan R., Felner I., Oshtrakh M.I., “Mössbauer spectroscopy of NiFe_2_O_4_ nanoparticles: the effect of Ni^2+^ in the Fe^3+^ local microenvironment in both tetrahedral and octahedral sites”, 159–168, ©2017.

**Figure 47 nanomaterials-12-03748-f047:**
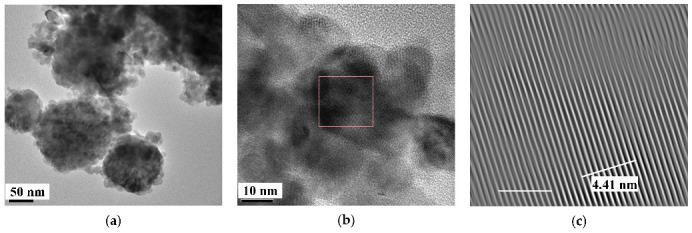
TEM (**a**) and HRTEM (**b**,**c**) images of NiFe_2_O_4_ nanoparticles in sample NA. An enlarged area marked in (**b**) is shown in (**c**). Adapted from Ref. [79] with permission from Elsevier: Materials Chemistry and Physics, Vol. 202, Ushakov M.V., Senthilkumar B., Kalai Selvan R., Felner I., Oshtrakh M.I., “Mössbauer spectroscopy of NiFe_2_O_4_ nanoparticles: the effect of Ni^2+^ in the Fe^3+^ local microenvironment in both tetrahedral and octahedral sites”, 159–168, ©2017.

**Figure 48 nanomaterials-12-03748-f048:**
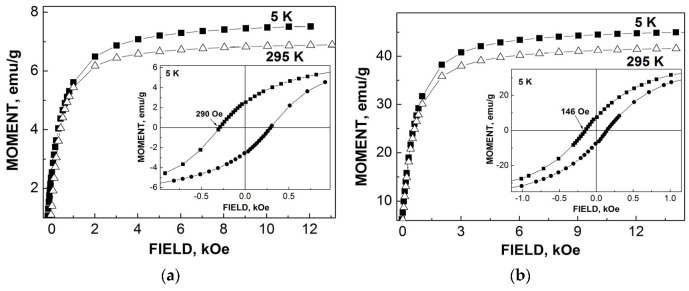
Isothermal magnetization curves for NA (**a**) and NB (**b**) nanoparticles at 5 and 295 K. Insets show the hysteresis loops at 5 K. Adapted from Ref. [79] with permission from Elsevier: Materials Chemistry and Physics, Vol. 202, Ushakov M.V., Senthilkumar B., Kalai Selvan R., Felner I., Oshtrakh M.I., “Mössbauer spectroscopy of NiFe_2_O_4_ nanoparticles: the effect of Ni^2+^ in the Fe^3+^ local microenvironment in both tetrahedral and octahedral sites”, 159–168, ©2017.

**Figure 49 nanomaterials-12-03748-f049:**
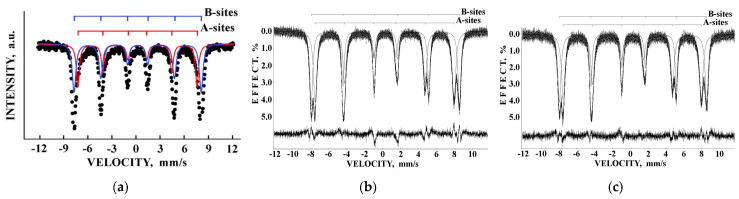
Room temperature Mössbauer spectra of NiFe_2_O_4_ nanoparticles measured with a low velocity resolution (**a**) and with a high velocity resolution: NA (**b**) and NB (**c**), adapted from Ref. [79] with permission from Elsevier: Materials Chemistry and Physics, Vol. 202, Ushakov M.V., Senthilkumar B., Kalai Selvan R., Felner I., Oshtrakh M.I., “Mössbauer spectroscopy of NiFe_2_O_4_ nanoparticles: the effect of Ni^2+^ in the Fe^3+^ local microenvironment in both tetrahedral and octahedral sites”, 159–168, ©2017. Indicated fits were performed using two magnetic sextets associated with ^57^Fe in the (A) and [B] sites in spinels. The differential spectra are shown on the bottom.

**Figure 50 nanomaterials-12-03748-f050:**
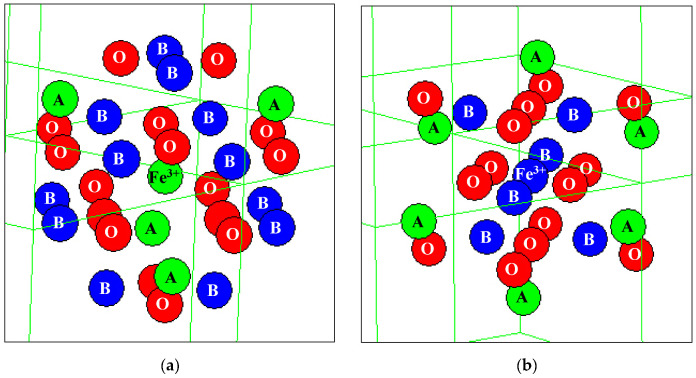
Local microenvironments of Fe^3+^ cation in the (A) (**a**) and [B] (**b**) sites of the ideal NiFe_2_O_4_ crystal: O (red circles) are O^2–^ ions, A (green circles) are tetrahedral sites occupied by Fe^3+^ cations and B (blue circles) are octahedral sites occupied by both Fe^3+^ and Ni^2+^ cations. Adapted from Ref. [79] with permission from Elsevier: Materials Chemistry and Physics, Vol. 202, Ushakov M.V., Senthilkumar B., Kalai Selvan R., Felner I., Oshtrakh M.I., “Mössbauer spectroscopy of NiFe_2_O_4_ nanoparticles: the effect of Ni^2+^ in the Fe^3+^ local microenvironment in both tetrahedral and octahedral sites”, 159–168, ©2017.

**Figure 51 nanomaterials-12-03748-f051:**
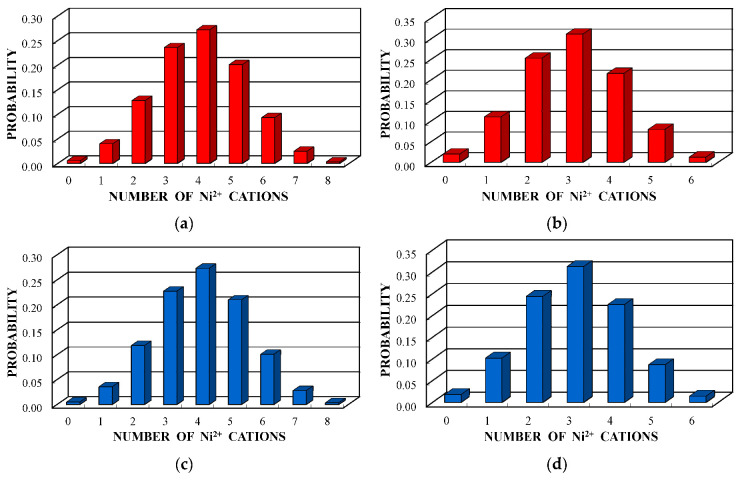
Histograms of the probabilities of different numbers of Ni^2+^ cations in the local microenvironments of Fe^3+^_A_ (**a**,**c**) and Fe^3+^_B_ (**b**,**d**) cations within the coordination sphere of 3.5 Å in the ideal NiFe_2_O_4_ crystal with chemical compositions corresponding to NA (**a**,**b**) and NB (**c**,**d**) samples. Adapted from Ref. [79] with permission from Elsevier: Materials Chemistry and Physics, Vol. 202, Ushakov M.V., Senthilkumar B., Kalai Selvan R., Felner I., Oshtrakh M.I., “Mössbauer spectroscopy of NiFe_2_O_4_ nanoparticles: the effect of Ni^2+^ in the Fe^3+^ local microenvironment in both tetrahedral and octahedral sites”, 159–168, ©2017.

**Figure 52 nanomaterials-12-03748-f052:**
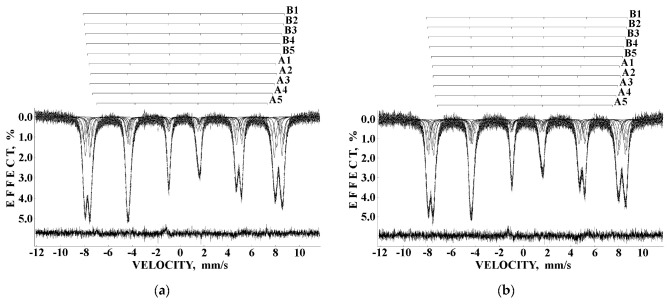
Room temperature Mössbauer spectra of NiFe_2_O_4_ nanoparticles measured with a high velocity resolution: NA (**a**) and NB (**b**). Indicated components A1–A5 and B1–B5 related to the (A) and [B] sites, respectively, are the results of the best fits. The differential spectra are shown on the bottom. Adapted from Ref. [79] with permission from Elsevier: Materials Chemistry and Physics, Vol. 202, Ushakov M.V., Senthilkumar B., Kalai Selvan R., Felner I., Oshtrakh M.I., “Mössbauer spectroscopy of NiFe_2_O_4_ nanoparticles: the effect of Ni^2+^ in the Fe^3+^ local microenvironment in both tetrahedral and octahedral sites”, 159–168, ©2017.

**Figure 53 nanomaterials-12-03748-f053:**
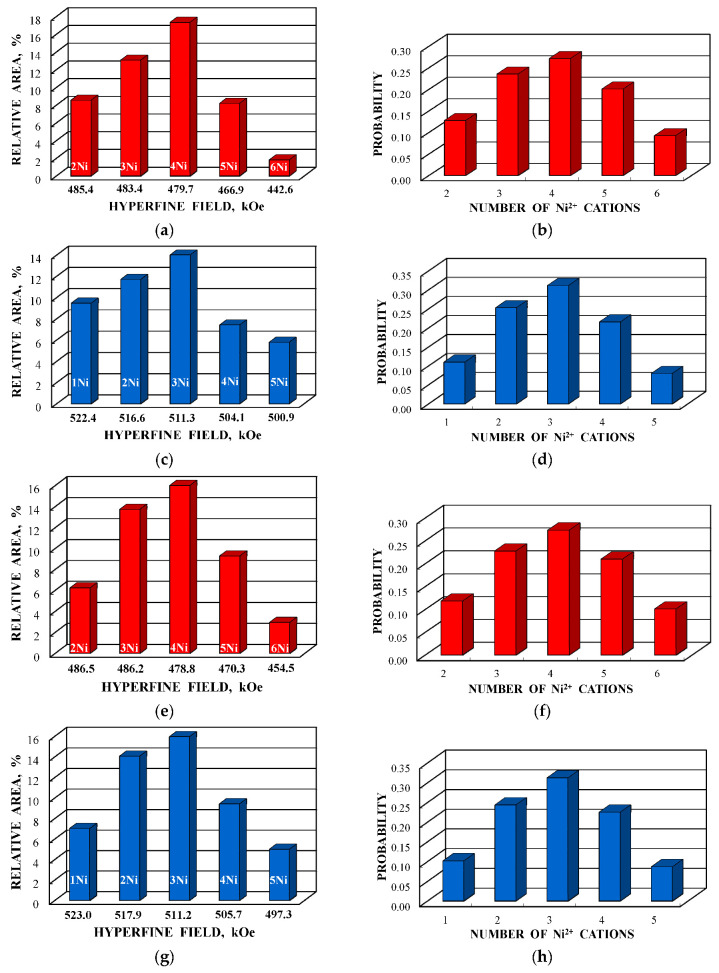
Comparison of histograms of the relative areas of spectral components assigned to the (A) (**a**,**e**) and [B] (**c**,**g**) sites in the Mössbauer spectra of NA (**a**,**c**) and NB (**e**,**g**) samples with *H*_eff_ decrease and corresponding histograms of the reliable probabilities (≥0.5) of different numbers of Ni^2+^ cations in the local microenvironments of Fe^3+^_A_ (**b**,**f**) and Fe^3+^_B_ (**d**,**h**) cations within the coordination sphere of 3.5 Å in the ideal NiFe_2_O_4_ crystal with chemical compositions corresponding to NA (**b**,**d**) and NB (**f**,**g**) samples. Adapted from Ref. [79] with permission from Elsevier: Materials Chemistry and Physics, Vol. 202, Ushakov M.V., Senthilkumar B., Kalai Selvan R., Felner I., Oshtrakh M.I., “Mössbauer spectroscopy of NiFe_2_O_4_ nanoparticles: the effect of Ni^2+^ in the Fe^3+^ local microenvironment in both tetrahedral and octahedral sites”, 159–168, ©2017.

**Figure 54 nanomaterials-12-03748-f054:**
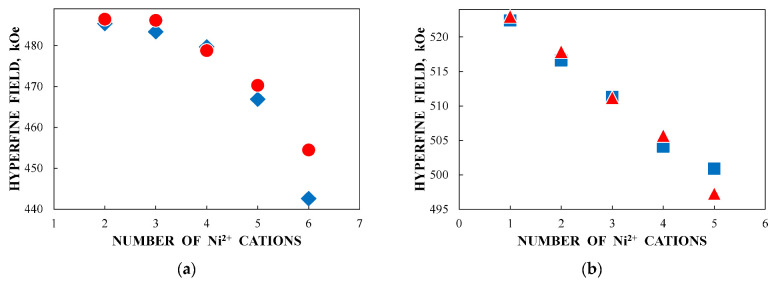
Dependencies of the hyperfine field at the ^57^Fe nuclei vs. number of Ni^2+^ cation in the Fe^2+^ local microenvironments in both tetrahedral (**a**) and octahedral (**b**) sites of NiFe_2_O_4_ nanoparticles in NA (♦) and NB (●) samples. Adapted from Ref. [79] with permission from Elsevier: Materials Chemistry and Physics, Vol. 202, Ushakov M.V., Senthilkumar B., Kalai Selvan R., Felner I., Oshtrakh M.I., “Mössbauer spectroscopy of NiFe_2_O_4_ nanoparticles: the effect of Ni^2+^ in the Fe^3+^ local microenvironment in both tetrahedral and octahedral sites”, 159–168, ©2017.

**Figure 55 nanomaterials-12-03748-f055:**
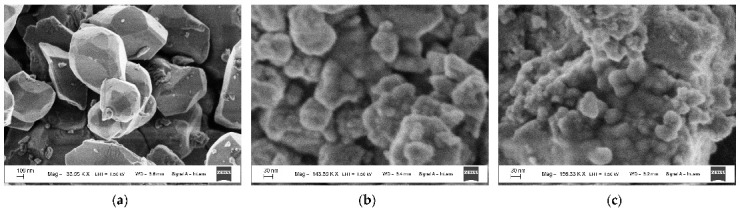
SEM images of NiFe_2_O_4_ nanoparticles in samples N1 (**a**), N2 (**b**) and N3 (**c**). Adapted from Ref. [83] with permission from Springer Nature: Hyperfine Interactions, “The milling effect on nickel ferrite particles studied using magnetization measurements and Mössbauer spectroscopy”, Ushakov M.V., Oshtrakh M.I., Chukin A.V., Šepelák V., Felner I., Semionkin V.A., ©2018.

**Figure 56 nanomaterials-12-03748-f056:**
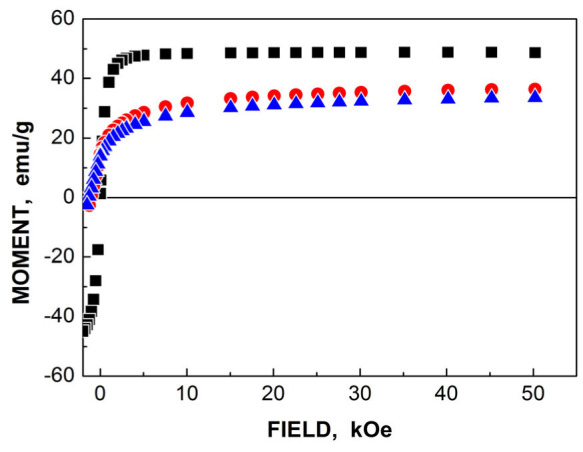
Isothermal magnetization curves measured at 295 K for NiFe_2_O_4_ samples: N1 (■), N2 (●) and N3 (▲). Adapted from Ref. [83] with permission from Springer Nature: Hyperfine Interactions, “The milling effect on nickel ferrite particles studied using magnetization measurements and Mössbauer spectroscopy”, Ushakov M.V., Oshtrakh M.I., Chukin A.V., Šepelák V., Felner I., Semionkin V.A., ©2018.

**Figure 57 nanomaterials-12-03748-f057:**
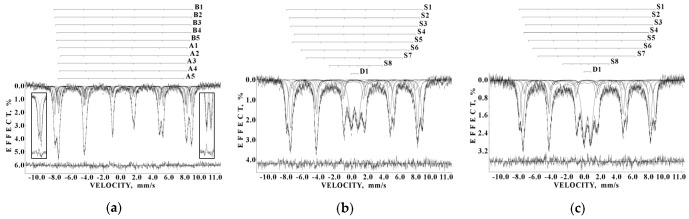
Mössbauer spectra of NiFe_2_O_4_ samples N1 (**a**), N2 (**b**) and N3 (**c**) measured at 295 K in 4096 channels and converted into the 1024-channel spectra. Indicated components are the results of the best fits (A1–A5 and B1–B5 are the magnetic sextets related to the (A) and [B] sites with different ^57^Fe local microenvironments, respectively; S1–S8 are the magnetic sextets, D1 is a paramagnetic doublet). The differential spectra are shown on the bottom. Inserts in the N1 spectrum (**a**) demonstrate misfits for the outer sextet peaks for the fit using the conventional model with two magnetic sextets assigned to the (A) and [B] sites only. Adapted from Ref. [83] with permission from Springer Nature: Hyperfine Interactions, “The milling effect on nickel ferrite particles studied using magnetization measurements and Mössbauer spectroscopy”, Ushakov M.V., Oshtrakh M.I., Chukin A.V., Šepelák V., Felner I., Semionkin V.A., ©2018.

**Figure 58 nanomaterials-12-03748-f058:**
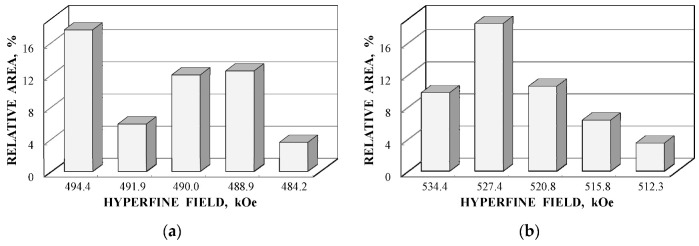
Histograms of the relative area variation for the magnetic sextets with decrease in *H*_eff_ at the ^57^Fe nuclei resulting from the different numbers of Ni^2+^ cation in the Fe^3+^ local microenvironments in both tetrahedral (**a**) and octahedral (**b**) sites of NiFe_2_O_4_ nanoparticles in sample N1.

**Figure 59 nanomaterials-12-03748-f059:**
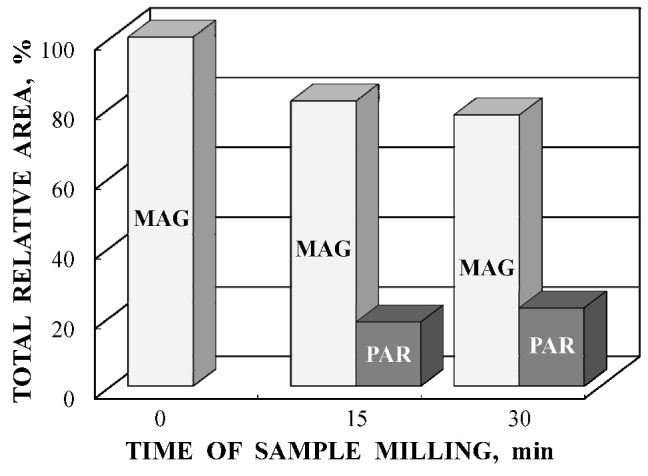
The change in the total relative area of magnetic (MAG) and paramagnetic (PAR) components in the Mössbauer spectra of NiFe_2_O_4_ samples N1, N2 and N2 with increase in milling time. Adapted from Ref. [83] with permission from Springer Nature: Hyperfine Interactions, “The milling effect on nickel ferrite particles studied using magnetization measurements and Mössbauer spectroscopy”, Ushakov M.V., Oshtrakh M.I., Chukin A.V., Šepelák V., Felner I., Semionkin V.A., ©2018.

**Figure 60 nanomaterials-12-03748-f060:**
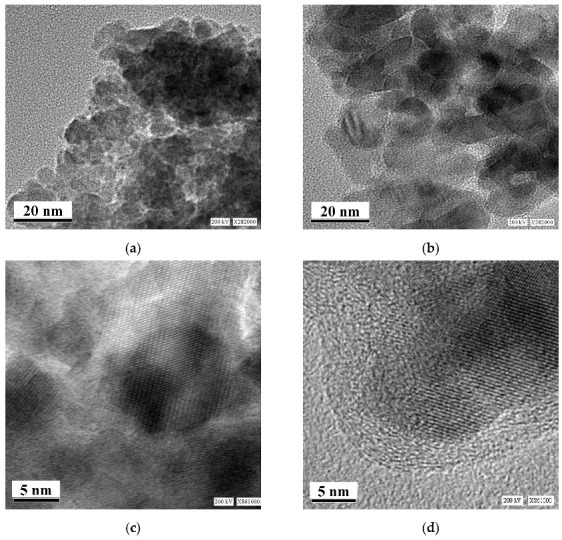
TEM (**a**,**b**) and HRTEM (**c**,**d**) images of CuFe_2_O_4_ (**a**,**c**) and (CuFe_2_O_4_)_0.95_(SnO_2_)_0.05_ (**b**,**d**). Adapted from Ref. [91] with permission from Springer Nature: Hyperfine Interactions, “Study of CuFe_2_O_4_–SnO_2_ nanocomposites by Mössbauer spectroscopy with high velocity resolution”, Oshtrakh M.I., Kalai Selvan R., Augustin C.O., Semionkin V.A., ©2008.

**Figure 61 nanomaterials-12-03748-f061:**
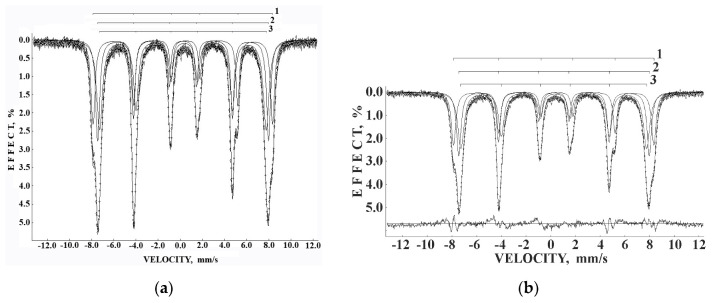
Mössbauer spectrum of CuFe_2_O_4_ measured with a high velocity resolution and converted into the 1024-channel spectrum and fitted using three magnetic sextets without the differential spectrum, adapted from Ref. [91] with permission from Springer Nature: Hyperfine Interactions, “Study of CuFe_2_O_4_–SnO_2_ nanocomposites by Mössbauer spectroscopy with high velocity resolution”, Oshtrakh M.I., Kalai Selvan R., Augustin C.O., Semionkin V.A., ©2008. (**a**) and the same fit with the differential spectrum on the bottom showing large misfits (**b**).

**Figure 62 nanomaterials-12-03748-f062:**
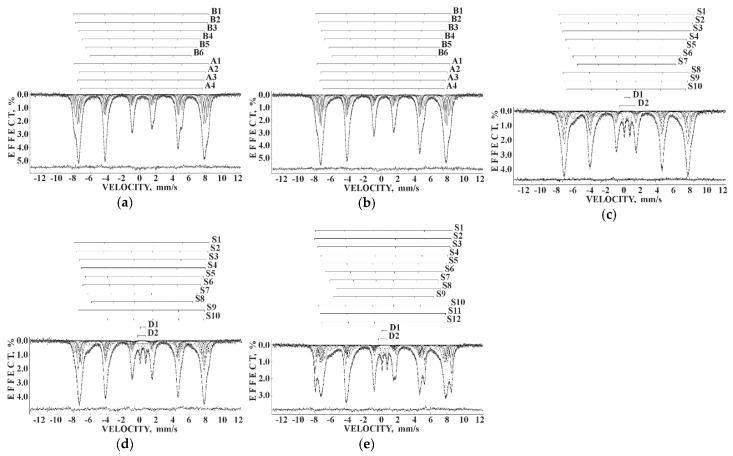
Refitted 1024-channel Mössbauer spectra of (CuFe_2_O_4_)_1–*x*_(SnO_2_)*_x_* nanocomposites with *x* = 0 wt.% (**a**), *x* = 1 wt.% (**b**), *x* = 5 wt.% (**c**), *x* = 10 wt.% (**d**) and *x* = 20 wt.% (**e**). Indicated components are the results of the better fits (A1–A4 and B1–B6 are the magnetic sextets related to the (A) and [B] sites with different ^57^Fe local microenvironments, respectively; S1–S12 are the magnetic sextets, D1 and D2 are the paramagnetic doublets). The differential spectra are shown on the bottom.

**Figure 63 nanomaterials-12-03748-f063:**
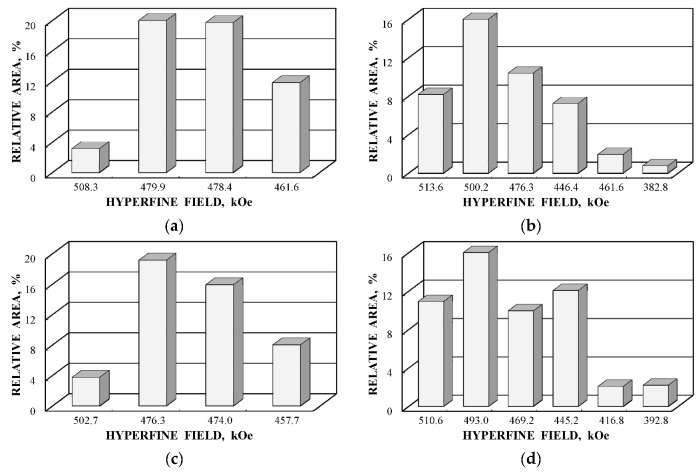
Histograms of the relative areas of variation for the magnetic sextets with a decrease in *H*_eff_ at the ^57^Fe nuclei resulting from the different numbers of Cu^2+^ cation in the Fe^2+^ local microenvironments in both tetrahedral (**a**,**c**) and octahedral (**b**,**d**) sites in CuFe_2_O_4_ nanoparticles (**a**,**b**) and the (CuFe_2_O_4_)_0.99_(SnO_2_)_0.01_ nanocomposite (**c**,**d**).

**Figure 64 nanomaterials-12-03748-f064:**
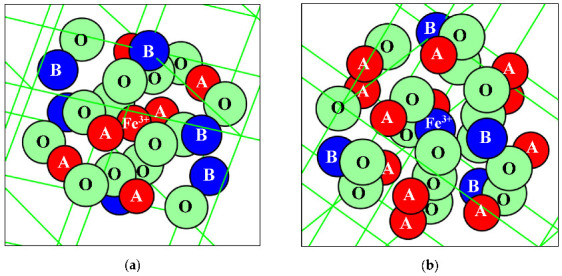
Local microenvironments of Fe^3+^ cation in the (A) (**a**) and [B] (**b**) sites of the ideal CuFe_2_O_4_ crystal: O (green circles) are O^2–^ ions, A (red circles) are tetrahedral sites occupied by Fe^3+^ cations, and B (blue circles) are octahedral sites occupied by both Fe^3+^ and Cu^2+^ cations.

**Figure 65 nanomaterials-12-03748-f065:**
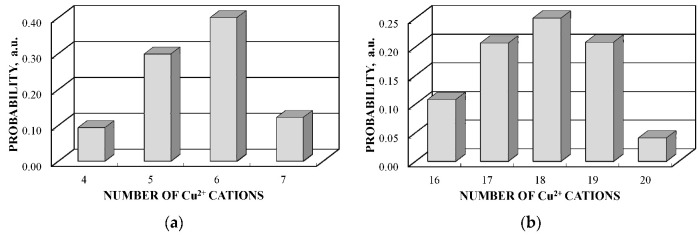
Histograms of the probabilities of different numbers of Cu^2+^ cations (total in the (A) and [B] sites) in the local microenvironments of Fe^3+^_A_ (**a**) and Fe^3+^_B_ (**b**) cations within the coordination sphere of 3.7 Å in the ideal CuFe_2_O_4_ crystal with one Cu^2+^ cation in the (A) sites.

**Figure 66 nanomaterials-12-03748-f066:**
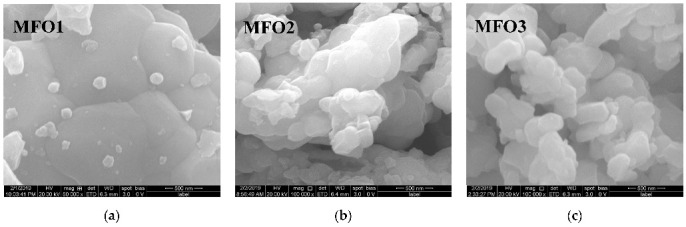
SEM images of MgFe_2_O_4_ particles in samples MFO1 (**a**), MFO2 (**b**) and MFO3 (**c**). Adapted from Ref. [94] with permission from Elsevier: Journal of Alloys and Compounds, Vol. 912, Ushakov M.V., Nithya V.D., Rajeesh Kumar N., Arunkumar S., Chukin A.V., Kalai Selvan R., Oshtrakh M.I., “X-ray diffraction, magnetic measurements and Mössbauer spectroscopy of MgFe_2_O_4_ nanoparticles”, 165125, ©2022.

**Figure 67 nanomaterials-12-03748-f067:**
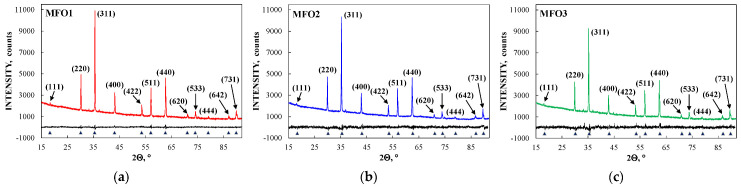
XRD patterns of MgFe_2_O_4_ nanoparticles in samples MFO1 (**a**), MFO2 (**b**) and MFO3 (**c**). Miller’s indices and results of Rietveld refinement of MgFe_2_O_4_ structure for the MFO1, MFO2 and MFO3 samples are shown. The differences between measured and calculated profiles are shown below the XRD patterns. Symbols ▲ indicate the positions of diffraction maxima in MgFe_2_O_4_ structure from ICDD card No. 01-088-1935. Adapted from Ref. [94] with permission from Elsevier: Journal of Alloys and Compounds, Vol. 912, Ushakov M.V., Nithya V.D., Rajeesh Kumar N., Arunkumar S., Chukin A.V., Kalai Selvan R., Oshtrakh M.I., “X-ray diffraction, magnetic measurements and Mössbauer spectroscopy of MgFe_2_O_4_ nanoparticles”, 165125, ©2022.

**Figure 68 nanomaterials-12-03748-f068:**
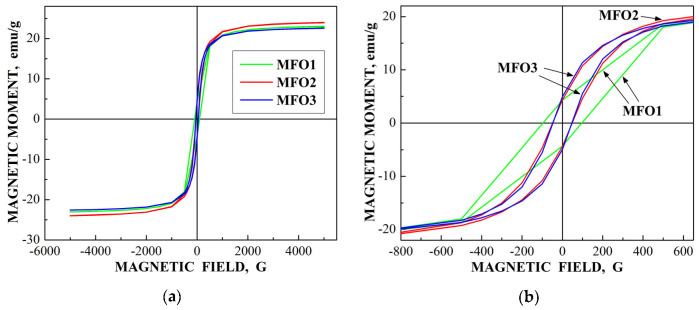
295 K isothermal magnetization curves of MFO1, MFO2 and MFO3 nanosized particles (**a**) and enlarged part of these curves with the hysteresis loops (**b**). Adapted from Ref. [94] with permission from Elsevier: Journal of Alloys and Compounds, Vol. 912, Ushakov M.V., Nithya V.D., Rajeesh Kumar N., Arunkumar S., Chukin A.V., Kalai Selvan R., Oshtrakh M.I., “X-ray diffraction, magnetic measurements and Mössbauer spectroscopy of MgFe_2_O_4_ nanoparticles”, 165125, ©2022.

**Figure 69 nanomaterials-12-03748-f069:**
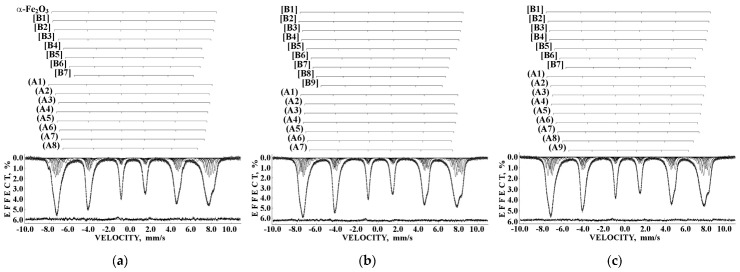
Room temperature Mössbauer spectra of MgFe_2_O_4_ nanoparticles in samples MFO1 (**a**), MFO2 (**b**) and MFO3 (**c**) measured with a high velocity resolution. Indicated components are the results of the best fits. The differential spectra are shown on the bottom. Adapted from Ref. [94] with permission from Elsevier: Journal of Alloys and Compounds, Vol. 912, Ushakov M.V., Nithya V.D., Rajeesh Kumar N., Arunkumar S., Chukin A.V., Kalai Selvan R., Oshtrakh M.I., “X-ray diffraction, magnetic measurements and Mössbauer spectroscopy of MgFe_2_O_4_ nanoparticles”, 165125, ©2022.

**Figure 70 nanomaterials-12-03748-f070:**
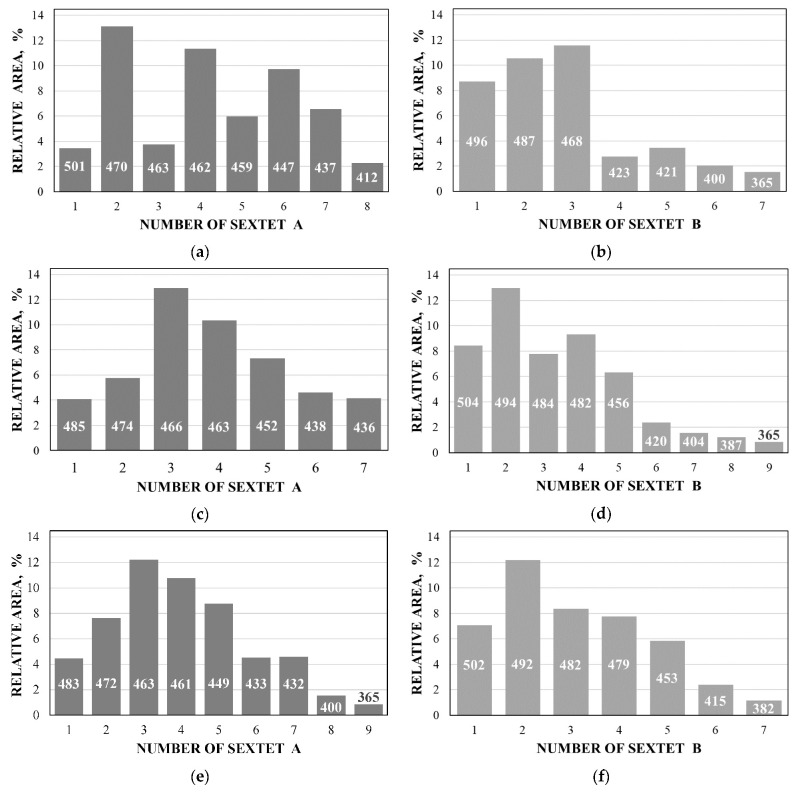
Histograms of the relative areas of magnetic sextets (relative iron fractions) for the corresponding iron local microenvironments in MFO1 (**a**,**b**), MFO2 (**c**,**d**) and MFO3 (**e**,**f**) resulting from the different numbers of Mg^2+^ cation in the Fe^3+^ local microenvironments in both tetrahedral (**a**,**c**,**e**) and octahedral (**b**,**d**,**f**) sites in MgFe_2_O_4_ nanoparticles. Numbers of sextets are the same as numbers of spectral components in Figure 69. The values of *H*_eff_ in kOe are shown for each sextet. Adapted from Ref. [94] with permission from Elsevier: Journal of Alloys and Compounds, Vol. 912, Ushakov M.V., Nithya V.D., Rajeesh Kumar N., Arunkumar S., Chukin A.V., Kalai Selvan R., Oshtrakh M.I., “X-ray diffraction, magnetic measurements and Mössbauer spectroscopy of MgFe_2_O_4_ nanoparticles”, 165125, ©2022.

**Figure 71 nanomaterials-12-03748-f071:**
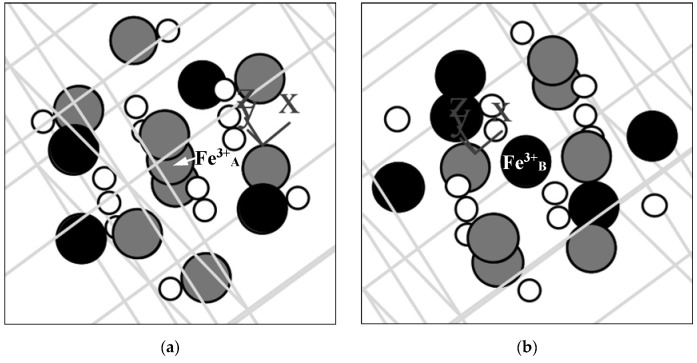
Local microenvironments of Fe^3+^ cation in the (A) (**a**) and [B] (**b**) sites of the ideal MgFe_2_O_4_ crystal: ● are the (A) sites, ● are the [B] sites, ○ are O^2–^ ions. Adapted from Ref. [94] with permission from Elsevier: Journal of Alloys and Compounds, Vol. 912, Ushakov M.V., Nithya V.D., Rajeesh Kumar N., Arunkumar S., Chukin A.V., Kalai Selvan R., Oshtrakh M.I., “X-ray diffraction, magnetic measurements and Mössbauer spectroscopy of MgFe_2_O_4_ nanoparticles”, 165125, ©2022.

**Figure 72 nanomaterials-12-03748-f072:**
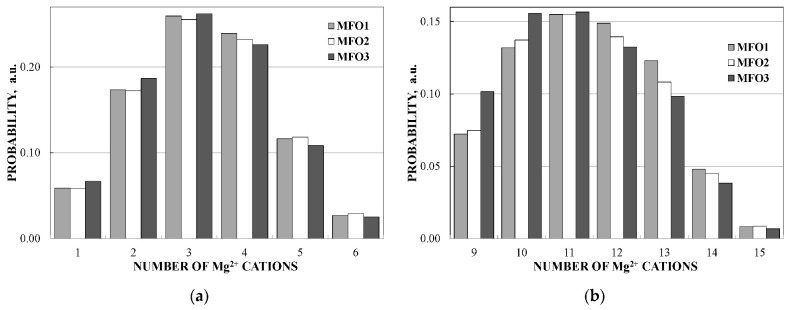
Histograms of the probabilities of different numbers of Mg^2+^ cations in the Fe^3+^_A_ (**a**) and Fe^3+^_B_ (**b**) local microenvironments within the coordination sphere of 3.7 Å in the ideal MgFe_2_O_4_ crystal with Fe and Mg compositions for MFO1, MFO2 and MFO3 samples. Adapted from Ref. [94] with permission from Elsevier: Journal of Alloys and Compounds, Vol. 912, Ushakov M.V., Nithya V.D., Rajeesh Kumar N., Arunkumar S., Chukin A.V., Kalai Selvan R., Oshtrakh M.I., “X-ray diffraction, magnetic measurements and Mössbauer spectroscopy of MgFe_2_O_4_ nanoparticles”, 165125, ©2022.

**Figure 73 nanomaterials-12-03748-f073:**
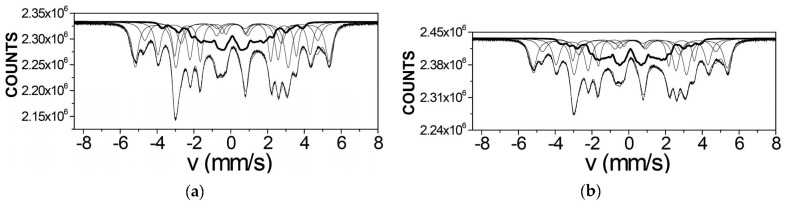
Room temperature Mössbauer spectra of FINEMET samples FT1 with permeability µ_r_ = 1350 (**a**) and FT2 with permeability µ_r_ = 6000 (**b**). Adapted from Ref. [101] with permission from Springer Nature: Hyperfine Interactions, “Mössbauer study of FINEMET with different permeability”, Kuzmann E., Stichleutner S., Sápi, A., Klencsár Z., Oshtrakh M.I., Semionkin V.A., Kubuki S., Homonnay Z., Varga L.K., ©2013.

**Figure 74 nanomaterials-12-03748-f074:**
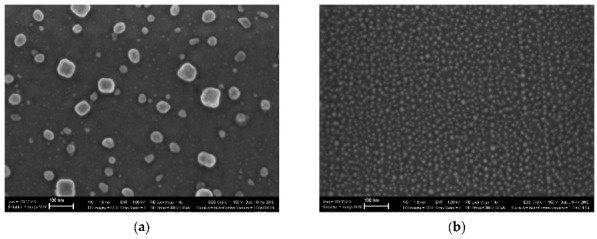
SEM images of FINEMET ribbons with different permeability: F1, µ_r_ = 34,500 (**a**) and F2, µ_r_ = 190,000 (**b**). Adapted from Ref. [102] with permission from Elsevier: Materials Chemistry and Physics, Vol. 180, Oshtrakh M.I., Klencsar Z., Semionkin V.A., Kuzmann E., Homonnay Z., Varga L.K., “Annealed FINEMET ribbons: Structure and magnetic anisotropy as revealed by the high velocity resolution Mössbauer spectroscopy”, 66–74, ©2016.

**Figure 75 nanomaterials-12-03748-f075:**
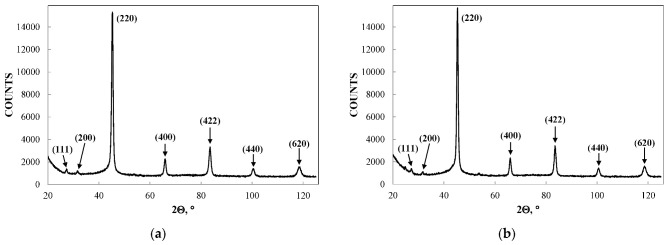
XRD patterns of FINEMET ribbons with different permeabilities: F1, µ_r_ = 34,500 (**a**) and F2, µ_r_ = 190,000 (**b**) with indicated Miller indices. Adapted from Ref. [102] with permission from Elsevier: Materials Chemistry and Physics, Vol. 180, Oshtrakh M.I., Klencsar Z., Semionkin V.A., Kuzmann E., Homonnay Z., Varga L.K., “Annealed FINEMET ribbons: Structure and magnetic anisotropy as revealed by the high velocity resolution Mössbauer spectroscopy”, 66–74, ©2016.

**Figure 76 nanomaterials-12-03748-f076:**
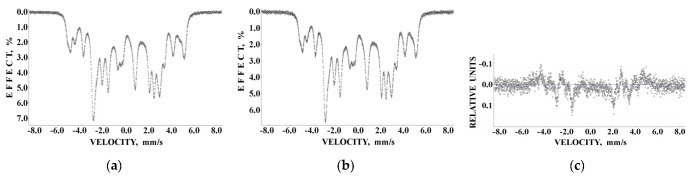
Room temperature Mössbauer spectra of FINEMET ribbons with different permeabilities: F1, µ_r_ = 34,500 (**a**) and F2, µ_r_ = 190,000 (**b**) and their normalized differences (**c**). The Mössbauer spectra were measured in 4096 channels and converted into 2048-channel spectra. Adapted from Ref. [102] with permission from Elsevier: Materials Chemistry and Physics, Vol. 180, Oshtrakh M.I., Klencsar Z., Semionkin V.A., Kuzmann E., Homonnay Z., Varga L.K., “Annealed FINEMET ribbons: Structure and magnetic anisotropy as revealed by the high velocity resolution Mössbauer spectroscopy”, 66–74, ©2016.

**Figure 77 nanomaterials-12-03748-f077:**
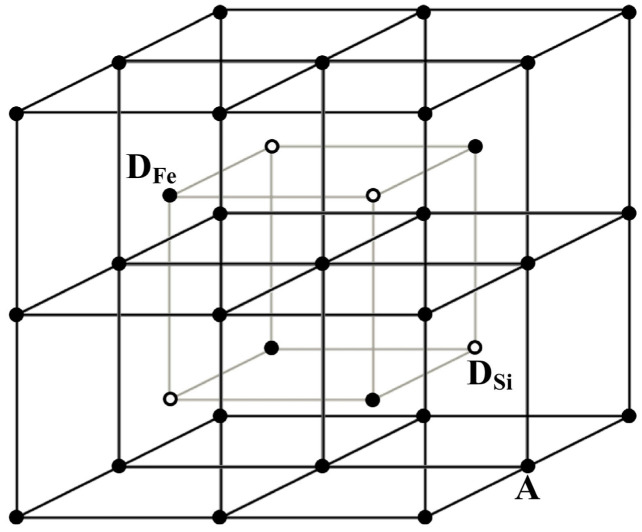
Scheme of α-Fe_3_Si with the ordered DO_3_ structure. Black circles are iron atoms at the A and D_Fe_ sites, while open circles are Si atoms at the D_Si_ sites. Black lines connect atoms at the A sites, while light grey lines connect the D-site atoms. Adapted from Ref. [102] with permission from Elsevier: Materials Chemistry and Physics, Vol. 180, Oshtrakh M.I., Klencsar Z., Semionkin V.A., Kuzmann E., Homonnay Z., Varga L.K., “Annealed FINEMET ribbons: Structure and magnetic anisotropy as revealed by the high velocity resolution Mössbauer spectroscopy”, 66–74, ©2016.

**Figure 78 nanomaterials-12-03748-f078:**
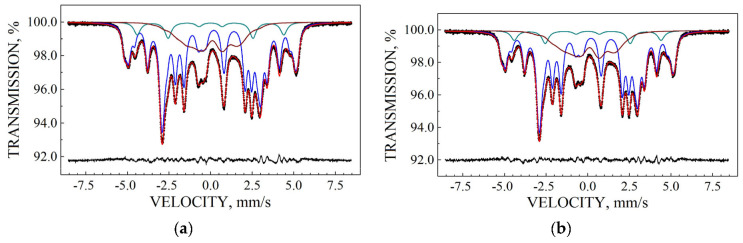
Room temperature Mössbauer spectra of FINEMET ribbons with different permeabilities: F1, µ_r_ = 34,500 (**a**) and F2, µ_r_ = 190,000 (**b**). The Mössbauer spectra were measured in 4096 channels and converted into 2048-channel spectra. The spectra were fitted by the model described in detail in [100]. The differential spectra are shown on the bottom. Adapted from Ref. [102] with permission from Elsevier: Materials Chemistry and Physics, Vol. 180, Oshtrakh M.I., Klencsar Z., Semionkin V.A., Kuzmann E., Homonnay Z., Varga L.K., “Annealed FINEMET ribbons: Structure and magnetic anisotropy as revealed by the high velocity resolution Mössbauer spectroscopy”, 66–74, ©2016.

**Figure 79 nanomaterials-12-03748-f079:**
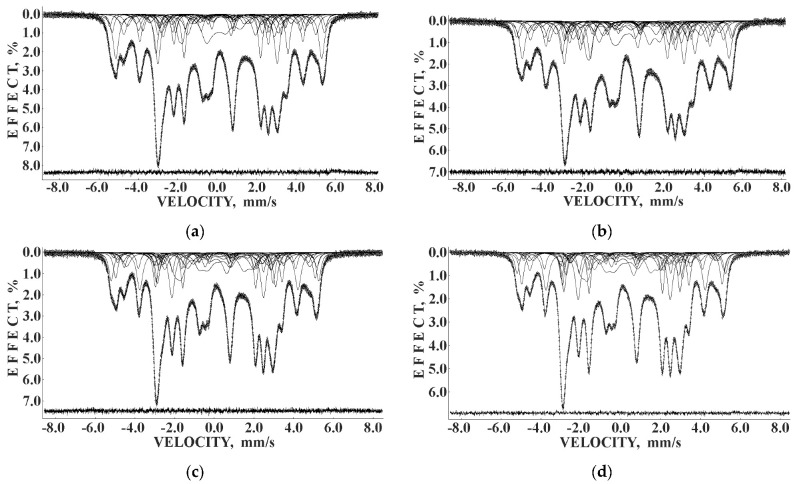
Room temperature Mössbauer spectra of FINEMET ribbons with different permeabilities: FT1, µ_r_ = 1350 (**a**), F2, µ_r_ = 6000 (**b**), F1, µ_r_ = 34,500 (**c**) and F2, µ_r_ = 190,000 (**d**). The Mössbauer spectra were measured in 4096 channels while the spectrum of F2 was converted into 2048-channel spectrum. These spectra were fitted using 18 magnetic sextets with the same ratio *A*_2,5_/*A*_1,6_. The differential spectra are shown on the bottom.

**Figure 80 nanomaterials-12-03748-f080:**
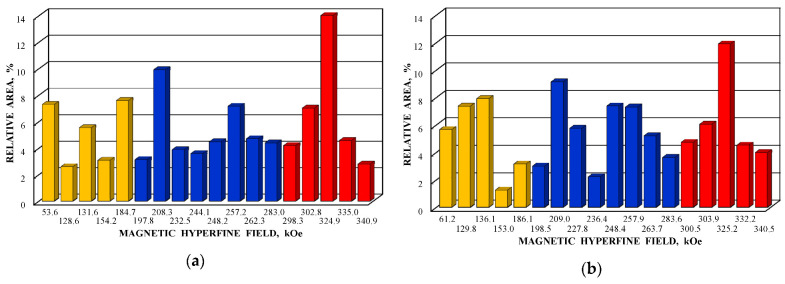
Histograms of the relative areas of the magnetic sextets revealed from the decomposition of the Mössbauer spectra of FINEMET ribbons with different permeabilities: FT1, µ_r_ = 1350 (**a**), FT2, µ_r_ = 6000 (**b**), F1, µ_r_ = 34,500 (**c**) and F2, µ_r_ = 190,000 (**d**) using 18 magnetic sextets vs. *H*_eff_ increase. ■ are magnetic sextets associated with different iron local microenvironments in the amorphous phase, ■ are magnetic sextets assigned to different iron local microenvironments in the crystalline sublattice D and ■ are magnetic sextets related to different iron local microenvironments in the crystalline sublattice A.

**Table 1 nanomaterials-12-03748-t001:** Mössbauer parameters for the spectral components of Imferon samples shown in Figure 22.

Sample	*T*, K	Γ, mm/s	δ, mm/s	Δ*E*_Q_ (2ε), mm/s	*H*_eff_, kOe	*A*, %	Spectral Component
Imferon	295	0.390 ± 0.009	0.356 ± 0.003	0.534 ± 0.015	–	55.22	1 (FeOOH)
(lyophilized)		0.390 ± 0.009	0.352 ± 0.003	0.880 ± 0.030	–	34.89	2 (FeOOH)
		0.390 ± 0.009	0.331 ± 0.009	1.268 ± 0.043	–	9.90	3 (FeOOH)
Imferon	87	0.396 ± 0.032	0.465 ± 0.016	0.530 ± 0.016	–	51.84	1 (FeOOH)
(frozen solution)		0.396 ± 0.032	0.457 ± 0.016	0.844 ± 0.016	–	31.59	2 (FeOOH)
		0.396 ± 0.032	0.444 ± 0.016	1.160 ± 0.016	–	12.43	3 (FeOOH)
		0.543 ± 0.032	1.228 ± 0.016	2.966 ± 0.018	–	4.13	4 (FeCl_2_)
Imferon	90	0.298 ± 0.024	0.452 ± 0.012	0.330 ± 0.012	–	6.14	1 (FeOOH)
(frozen solution)		0.323 ± 0.024	0.424 ± 0.012	0.580 ± 0.012	–	19.01	2 (FeOOH)
		0.537 ± 0.024	0.427 ± 0.012	0.889 ± 0.012	–	30.53	3 (FeOOH)
		0.694 ± 0.073	0.473 ± 0.017	−0.329 ± 0.035	440.0 ± 2.2	7.06	4 (FeOOH)
		0.489 ± 0.028	0.449 ± 0.012	−0.220 ± 0.012	472.0 ± 0.6	19.26	5 (FeOOH)
		0.292 ± 0.024	0.471 ± 0.012	−0.257 ± 0.012	485.2 ± 0.5	14.56	6 (FeOOH)
		0.705 ± 0.080	1.148 ± 0.027	3.199 ± 0.051	–	3.44	7 (FeCl_2_)

## Data Availability

All data are available from this paper and cited papers as well as from the corresponding author.

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
