# Peer review of "Mössbauer Spectroscopy with a High Velocity Resolution in the Studies of Nanomaterials"

_nanomaterials, 2022, doi:10.3390/nano12213748_

Round 1

Reviewer 1 Report

This is an intweresting and well-rwitten review about Mössbauer spectrsoscopy and its application to the analysis of Fe-57 containing nanomaterials. The chosen velocity resolution is higher compared to most other Mössbauer spectrometers, so that the only resolution limit is given by the natural line width. In this way it is possible to fit Mössbauer data with multiple components, e.g. up to 16 -18 magnetic sextets as shown in Figs.31. or 80.

The various examples are well described and the corresponding figures and images are of high quality.

Her the question arises if fitting with a certain continuous probability distribution of a microenviroment of magetic fields would be more appropriate. It is clear, that continuous probability distributions can only be resolved with up to about 16 values. Hoverver, the approach to analyze and fit Mössbauer spectra would be quite different.

In the conclusion the authors indicate the need of new physical models for decomposition of complex Mössbauer spectra. The authors may comment on the feasibility or usefulness of decomposition using probability distribution functions.

Author Response

We are very grateful to Reviewer 1 for his/her positive evaluation of our manuscript and useful comments and remarks.

Comments by Reviewer 1.

Reviewer 1: “Her the question arises if fitting with a certain continuous probability distribution of a microenviroment of magetic fields would be more appropriate. It is clear, that continuous probability distributions can only be resolved with up to about 16 values. Hoverver, the approach to analyze and fit Mössbauer spectra would be quite different.

M.O.: We have checked the spectra fitting using the model-independent distribution functions for magnetically split spectra as well as for paramagnetic two-peak spectra and found that the distribution functions can confirm a complicity of our Mössbauer spectra. However, the numbers of components obtained by the fit of the distribution functions by Gaussian/Lorentzian lines were usually smaller than the number of components in the case of model-dependent individual spectral components which can be directly related to the corresponding Fe local microenvironments.

Reviewer 1: “In the conclusion the authors indicate the need of new physical models for decomposition of complex Mössbauer spectra. The authors may comment on the feasibility or usefulness of decomposition using probability distribution functions.

M.O.: These physical models cannot be related to the spectra fits using the model-independent distribution functions because as we have mentioned above, the number of individual components in the model-dependent fits appeared to be larger than in the case of distribution function fit. If this is a real possibility to feel the different Fe local microenvironments by the high velocity resolution Mössbauer spectroscopy, e.g., in ferrites, maybe the large number of spectral components in the case of nanoparticles requires to consider a complex nanoparticle architecture. Therefore, details require new physical models to explain these results (deviation from stoichiometry and variations in non-stoichiometry, different nano-domains, layers/regions, different sizes, etc.).

Reviewer 2 Report

The manuscript by Alenkina et al is a comprehensive review of the use of Mössbauer spectroscopy to study nanomaterials and was an enjoyable and informative read. The review is well organized and describes a huge volume of research, containing an excellent description of the equipment and experimental methods used in this type of studies.

Considering that the review is of interest for a wide variety of readers which are not familiar with Mössbauer spectroscopy, the authors should include comments regarding:

-       Signal to noise relationship with different velocity reference signal discretizations;

-       Why not covering Liquid-He temperature range? Why are most acquisitions done between 90K and room temperature? Is it possible to gain further information using temperatures down to 1.8 K?

-       Samples use natural abundance of 57Fe. Why is it important and how acquisition time is influenced?

-       Why when using biological materials and cells all iron is considered as ferritin iron? What about the considerable percentage due to heme and FeS iron cofactors?

Regarding figures, I would like to stress that the authors should improve global quality. Also:

-       Figure 2 would be best if one could have the comparison between 128, 512, 1024 and 4096 channels. This could be done using the 4096-channel spectrum and adding channels to reproduce the lower resolution spectra. 

-       Figures such as Fig. 33 and 40 do not add much to the understanding of the phenomena they try to explain.

-       There should also be consistency in. the use of color or greyscale in figures.

-       3D-bar graphics (such as Fig 14) are less legible and should be replotted as 2D-bar graphics with error bars in percentage.

Author Response

We are very grateful to Reviewer 2 for his/her positive evaluation of our manuscript and useful comments and remarks.

Comments by Reviewer 2.

Reviewer 2: “Considering that the review is of interest for a wide variety of readers which are not familiar with Mössbauer spectroscopy, the authors should include comments regarding:

-  Signal to noise relationship with different velocity reference signal discretizations;

M.O.: There is no signal-to-noise ratio relationship with discretization of the velocity reference signal while intensity in the spectrum increases twice in the case of the spectrum conversion from 4096 to 2048 channels. This leads to increase of the signal-to-noise ratio in 21/2. The same value of the signal-to-noise ratio can be reached in the 4096-channel spectrum by increasing the measurement time twice. We have added a short explanation in the revised text.

-  Why not covering Liquid-He temperature range? Why are most acquisitions done between 90K and room temperature? Is it possible to gain further information using temperatures down to 1.8 K?

M.O.: Our Mössbauer spectrometric complex with a high velocity resolution and saw-tooth velocity reference signal is equipped with a liquid nitrogen temperature variable cryostat with moving absorber only. There is not any chance to find a liquid helium cryostat with moving absorber by now. Therefore, all other measurements at lower temperatures were carried out using spectrometers with a low velocity resolution, triangular velocity reference signal and moving source. However, these results were not considered here.

-  Samples use natural abundance of 57Fe. Why is it important and how acquisition time is influenced?

M.O.: It is not possible to enrich human ferritin with 57Fe as well as it is not possible to enrich commercial pharmaceutical products. However, ferritin and its analogues as well as other nanosized materials have the iron content which is enough for measurements. The only tissues samples have a very small iron content while bacterial cells were grown with 57Fe (we have added this fact into the revised text). An acquisition time depends on the iron content and the absorption effect in the sample, the source activity and the efficiency of the g-rays registration system. The spectrometers’ stability is very high to keep the same velocity per channel for different velocity ranges for two years as it was shown in our methodological paper (see Ref. [29]).

-  Why when using biological materials and cells all iron is considered as ferritin iron? What about the considerable percentage due to heme and FeS iron cofactors?

M.O.: There is not any other iron-containing proteins in quantities which can be detected by Mössbauer spectroscopy except some residual methemoglobin content in the spleen and liver tissues, otherwise, we were able to detect these biomolecules spectral components because their Mössbauer hyperfine parameters are different from those for ferritin (ferritin-like molecules).

Reviewer 2: “Regarding figures, I would like to stress that the authors should improve global quality. Also:

-  Figure 2 would be best if one could have the comparison between 128, 512, 1024 and 4096 channels. This could be done using the 4096-channel spectrum and adding channels to reproduce the lower resolution spectra.

M.O.: Yes, thank you! However, this figure has already been published and adapted for this review from Ref. [31]. These are two spectra measured in different laboratories with different velocity resolutions, therefore, there are not the high velocity resolution spectra converted into 512, 1024, etc. channels. However, the required comparison has already been published using the ferritin Mössbauer spectrum converted from 4096 channels to lower channel numbers (see Refs. [31, 39]).

Reviewer 2: “-  Figures such as Fig. 33 and 40 do not add much to the understanding of the phenomena they try to explain.

M.O.: The Reviewer pointed out above that “Considering that the review is of interest for a wide variety of readers …”, therefore, we decided to use some schemes including those shown in Figs. 33 and 40 to demonstrate the readers surface modification of nanoparticles. These schemes have already been published in Ref. [74] and adapted for this review manuscript.

-  There should also be consistency in. the use of color or greyscale in figures.

M.O.: Which kind of consistency should be considered? This is a review; therefore, we have used adapted figures and some new ones. In this case, some published figures were in grey while the other figures were in color. These figures do not contradict each other.

-  3D-bar graphics (such as Fig 14) are less legible and should be replotted as 2D-bar graphics with error bars in percentage.

M.O.: Fig. 14 was adapted from Ref. [61], therefore, we were able only to change some colors keeping the 3D-bar graphics to adapt this figure like other adapted figures. The relative parts of two components were calculated from the relative areas without methemoglobin content. The relative error for A was already given in section 2.

Reviewer 3 Report

This manuscript presents a comprehensive review of application of Mössbauer spectrocopy with a high velocity resolution as a unique probe in analysis of diverse iron-containing nanoparticles and nanostructured materials in both abiotic and biotic systems. Applications in several topics are summarized including ferritins and its pharmaceutical analogues, nickel, copper and magnesium ferrite nanoparticles, and FINEMET alloy with nano-crystalline phases. The advantages of high velocity resolution Mössbauer spectroscopy for the excavation of more information from complicated Mössbauer spectra are discussed in detail. I believe that the theme of this paper is suitable for nanomaterials and it would appeal broad attention in this area. However, the following issues should be settled before publish:

1. The Mössbauer spectra of γ-Fe2O3 synthesized by different methods seem to be different, such as Figure 36, Figure 43, the authors are suggested to give a detailed explanation of what causes their differences, and whether there is a standard Mössbauer spectrum for this pure phase of γ-Fe2O3。

2. Part of the manuscript is not fully presented. For example, since the authors summarize the use of high velocity resolution Mössbauer spectroscopy in nanosized spinel ferrites, spinel nanosized ferrites other than NiFe2O4, CuFe2O4 and MgFe2O4, if there is any, should be discussed as well.

3. The authors are suggested to give a more detailed discussion on the remained challenges and required efforts for further development of the high velocity resolution Mössbauer technique.

4. Some literature reports involving high velocity resolution Mössbauer spectroscopy in nanoparticles and nanostructured materials are not discussed in this manuscript. For examples, Applied Catalysis B: Environmental, 2022, 300, 120720,Chem Catalysis 2021, 1, 1215, are suggested.

5. Please check through the manuscript carefully. Some formatting errors, such as spaces in formulas in line 100 and line 102, Page 3.

Author Response

We are very grateful to Reviewer 3 for his/her positive evaluation of our manuscript and useful comments and remarks.

Comments by Reviewer 3.

Reviewer 3: “1. The Mössbauer spectra of γ-Fe2O3 synthesized by different methods seem to be different, such as Figure 36, Figure 43, the authors are suggested to give a detailed explanation of what causes their differences, and whether there is a standard Mössbauer spectrum for this pure phase of γ-Fe2O3.”

M.O.: The reason of differences is related at least to different methods of synthesis and different sizes of nanoparticles as shown in Fig. 35d (~8 nm) and in Fig. 41c (~10.6 nm). We have added an explanation into the revised text. There is not the standard Mössbauer spectrum of pure γFe2O3 nanoparticles.

Reviewer 3: “2. Part of the manuscript is not fully presented. For example, since the authors summarize the use of high velocity resolution Mössbauer spectroscopy in nanosized spinel ferrites, spinel nanosized ferrites other than NiFe2O4, CuFe2O4 and MgFe2O4, if there is 4 any, should be discussed as well.”

M.O.: We did not study other ferrites by Mössbauer spectroscopy with a high velocity resolution yet. Therefore, the only results obtained using our technique were discussed.

Reviewer 3: “3. The authors are suggested to give a more detailed discussion on the remained challenges and required efforts for further development of the high velocity resolution Mössbauer technique.”

M.O.: It would be great; however, these are methodological and technical problems mainly related to development of new types of Mössbauer spectrometers with a high velocity resolution taking into account our long experience. This is a special topic which is not interesting for the readers who are not familiar with Mössbauer spectrometer velocity driving system and γ-ray
registration system as well as automatization of spectrometer.

Reviewer 3: “4. Some literature reports involving high velocity resolution Mössbauer spectroscopy in nanoparticles and nanostructured materials are not discussed in this manuscript. For examples, Applied Catalysis B: Environmental, 2022, 300, 120720,Chem Catalysis 2021, 1, 1215, are suggested.”

M.O.: This is not correct. For example, in the first reference mentioned by Reviewer 3 the authors wrote: “All spectra were recorded in 1024 channels with a velocity scale of approximately ± 10 mm/s. The spectrometer operated in constant acceleration mode with a triangular reference signal”. This means that the authors measured two mirror spectra with registration of each in 512
channels with total 1024 channels. The folded spectra are in 512 channels.
By the way, this review discusses our own results because we do not know any the other published high velocity resolution Mössbauer spectra, unfortunately.

Reviewer 3: “5. Please check through the manuscript carefully. Some formatting errors, such as spaces in formulas in line 100 and line 102, Page 3.”

M.O.: Thank you very much. These spaces were removed.